# Longitudinal single-cell profiling of chemotherapy response in acute myeloid leukemia

Matteo Maria Naldini [1,2], Gabriele Casirati [3,7], Matteo Barcella[1,4,7], Paola Maria Vittoria Rancoita[5], Andrea Cosentino[1,2], Carolina Caserta[1,2], Francesca Pavesi[3], Erika Zonari[1], Giacomo Desantis [1], Diego Gilioli[1,2], Matteo Giovanni Carrabba [3], Luca Vago [3], Massimo Bernardi [3], Raffaella Di Micco [1], Clelia Di Serio[5], Ivan Merelli[4], Monica Volpin [1], Eugenio Montini [1], Fabio Ciceri[3] & Bernhard Gentner [1,3,6] ✉

Acute myeloid leukemia may be characterized by a fraction of leukemia stem cells (LSCs) that sustain disease propagation eventually leading to relapse. Yet, the contribution of LSCs to early therapy resistance and AML regeneration remains controversial. We prospectively identify LSCs in AML patients and xenografts by single-cell RNA sequencing coupled with functional validation by a microRNA-126 reporter enriching for LSCs. Through nucleophosmin 1 (*NPM1*) mutation calling or chromosomal monosomy detection in single-cell transcriptomes, we discriminate LSCs from regenerating hematopoiesis, and assess their longitudinal response to chemotherapy. Chemotherapy induced a generalized inflammatory and senescence-associated response. Moreover, we observe heterogeneity within progenitor AML cells, some of which proliferate and differentiate with expression of oxidative-phosphorylation (OxPhos) signatures, while others are OxPhos (low) miR-126 (high) and display enforced stemness and quiescence features. miR-126 (high) LSCs are enriched at diagnosis in chemotherapy-refractory AML and at relapse, and their transcriptional signature robustly stratifies patients for survival in large AML cohorts.

The treatment of patients with acute myeloid leukemia (AML) remains a challenge, with 5-year survival rates around 30% (SEER Cancer stat facts)[1]. Treatment failure may occur early (primary refractory disease or early relapse, typically defined as a remission duration of less than 6 months), or years after the end of treatment, with different biological and clinical implications[2]. Early treatment failure is thought to be due to resistance in disease subclones with high leukemia-regenerating capacity surpassing the reconstitution kinetics of non-malignant

hematopoiesis. Relapse after the achievement of complete remission most likely arises from minimal residual disease (MRD) persisting in a dormant state or in sanctuary sites, where leukemia-regenerating and immune-evasion capacities may gradually evolve over time[3,4]. Evidence from retrospective, paired diagnosis-relapse sequencing studies supports the notion that therapy-resistant cells are often pre-existing at diagnosis, but their prospective identification remains a challenge[5,6]. It has been shown that, in about a third of AML cases, the late relapse

[1]San Raffaele Telethon Institute for Gene Therapy (SR-TIGET), IRCCS San Raffaele Scientific Institute, Milan, Italy. [2]Vita-Salute San Raffaele University, Milan, Italy. [3]Hematology and Bone Marrow Transplantation Unit, IRCCS San Raffaele Hospital, Milan, Italy. [4]National Research Council, Institute for Biomedical Technologies, Segrate, Italy. [5]University Center for Statistics in the Biomedical Sciences (CUSSB), Vita-Salute San Raffaele University, Milan, Italy. [6]Ludwig Institute for Cancer Research and Department of Oncology, University of Lausanne (UNIL) and Lausanne University Hospital (CHUV), Lausanne, Switzerland. [7]These authors contributed equally: Gabriele Casirati, Matteo Barcella. ✉e-mail: gentner.bernhard@hsr.it

clones could be traced back to rare populations of leukemia stem cells (LSCs), which were hardly detectable in the leukemia bulk sample at diagnosis[7]. Less often, clonally unrelated leukemia may independently arise on a substrate of pre-leukemic hematopoietic stem cells (HSCs)[8–12].

Knowledge on the cell biology and population dynamics of relapse-promoting cells during chemotherapy is incomplete. The classical LSC model, whereby chemotherapy-resistant LSCs accumulate during successive cycles of chemotherapy, has recently been challenged. It has been suggested that LSCs are efficiently depleted by cytarabine in vivo[13–15] and undergo a rapid regenerative response in a limited time window following chemotherapy leading to increasingly aggressive leukemia re-growth and presenting a therapeutic vulnerability for the targeting of therapy-resistant AML[14]. Additionally, a reversible state of cellular senescence has also been described to actively contribute to disease persistence by reprogramming differentiated residual tumor cells towards an enhanced stemness state with increased in vivo propagating capacity upon escape from senescence-induced proliferative block[16]. The interchange between primitive and differentiated compartments would argue for substantial plasticity within AML cells rather than a hierarchical LSC model characterized by unidirectional differentiation. A direct demonstration of LSC plasticity in response to chemotherapy is still pending and requires single-cell analysis combined with stringent LSC identification methods and clonal tracking.

We hypothesized that longitudinal studies of LSCs before and early after chemotherapy with single-cell resolution would shed further light on the cell population changes and underlying gene expression profiles occurring during the treatment of AML patients. We have previously generated microRNA (miRNA) profiles in functionally validated human AML LSC fractions and developed a score composed of 4 miRNAs that independently predicted patient outcome[17]. As part of this score, miR-126 was tightly coupled with LSC and HSC function, preserving their quiescent state and promoting chemotherapy resistance[17–20]. Moreover, ectopic expression of miR-126 in murine HSC may induce a miR-126 addicted acute leukemia, underlining its oncogenic potential[21]. Lentiviral reporter vectors allow live monitoring of miRNA activity in single cells[22] and may serve to identify LSCs when made responsive to LSC-specific miRNAs[17].

Focusing on *NPM1*-mutated (*NPM1*[mut]) AML as a model[23], we here show that the miR-126[high] cell subpopulation, revealed by an optimized lentiviral miR-126 reporter, enriches for LSCs in xenografts, allowing dynamic LSC monitoring before and after chemotherapy and generation of a transcriptional LSC signature in prospectively isolated LSCs. This signature was highly predictive of patient outcome and identified a circumscribed miR-126[high] LSC subpopulation in single-cell RNA sequencing (scRNAseq) datasets, including a second subgroup of AMLs characterized by chromosome 7 deletion and myelodysplasia-related changes (del(7) AML). Single-cell analysis of longitudinal patient samples and xenograft chemotherapy models revealed differential responses across AML blast clusters and provided sufficient resolution to detect miR-126[high] LSCs with an inflammatory, senescence-like and enhanced stemness profile persisting upon induction chemotherapy.

## Results

### Induction chemotherapy shapes the transcriptional landscape of AML

To obtain an unbiased view on intra-tumor heterogeneity (ITH) of AML at diagnosis and following induction chemotherapy, we performed droplet-based single-cell RNA sequencing (scRNAseq) on leukemia-enriched BM cells from patients with AML (Fig. 1A and Supplementary Fig. 1A). For PT06, peripheral blood instead of BM was available at diagnosis. The study cohort included 10 patients with *NPM1*[mut] AML

(PT01, 02, 13: refractory disease; PT08, 09, 10, 15: early relapse at 3-6 months following complete remission (CR): PT06, 07, 12: long-term CR beyond 34 months) and 3 patients with del(7) AML (PT11, PT17: refractory disease; PT18: early relapse). Clinical characteristics and post-sequencing cell recovery parameters are reported in Supplementary Fig. 1B and Supplementary Data 1. To unambiguously distinguish malignant from non-leukemic cells, a particularly critical aspect in post-chemotherapy samples where LSCs may hide within regenerating hematopoiesis by sharing surface markers with non-malignant HSC, we determined the *NPM1* mutational status or expression of the module score of genes located on chromosome 7 in single cells, respectively (see Methods). Both alterations unambiguously mark leukemia cells and gave us confidence in excluding bias from HSC contamination. Applying cluster-based assignment rules, the majority of cells could be classified (Supplementary Fig. 1C and Supplementary Data 1). We transplanted diagnosis samples from PT01 and PT06, predicted to contain, respectively, a low and high frequency of residual non-malignant progenitors, into sub-lethally irradiated NSG mice. The sample from PT01 resulted in a progressively expanding leukemia graft, while the sample from PT06 generated a multi-lineage graft functionally confirming the presence of non-leukemic SCID-repopulating cells (SRCs), validating our scRNAseq-based classification algorithm (Supplementary Fig. 1D).

To investigate intra-tumor heterogeneity, we merged all 42,398 *NPM1*[mut] AML blasts into a single dataset, ran a harmonization process to account for patient-driven variability (Supplementary Fig. 1E)[24] and performed UMAP dimensionality reduction[25]. Unsupervised clustering (resolution = 0.6) identified 15 distinct clusters (cl.) distributed across the AML landscape (Fig. 1B). Marker gene analysis (Supplementary Fig. 1F, Supplementary Data 2) distinguished early progenitors (cl. 0, 11, 14) expressing several LSC-associated genes such as *CD34*, *CD99*, *HOPX* and *EGFL7*, the host gene for miR-126; myeloid progenitors (cl. 1, 3, 13); erythroid-like precursors (cl. 7, 9); actively cycling blasts (cl. 2, 10); differentiating myelo-/monoblasts (cl. 5, 6, 4, 8) and a microcluster with NK-like characteristics (cl. 12). Leukemic cells from individual patients at diagnosis distributed differently within our *NPM1*[mut] AML landscape, some predominantly mapping to the immature-like (PT01, 02, 06, 13) or the monocyte-like areas (PT07), and others showing a broader representation across the whole landscape (PT08, 09, 10, 12, 15) (Fig. 1C). Pseudotime analysis traced a trajectory from early progenitors to the monocyte-like cells, passing through the actively cycling cluster, supporting a continuum of cell differentiation linked to cell cycle entry (Fig. 1D).

Likewise, 33,308 del(7) AML cells were projected on a 2D UMAP landscape, distinguishing a prominent early progenitor cell compartment (cl. 0, 1, 5, 9), lymphoid-like clusters (cl. 2, 7), erythroid-like clusters (cl. 3, 8), a cycling compartment (cl. 6), differentiated myeloid-like cells (cl. 4) and a microcluster of NK-like cells (cl. 11) (Fig. 1E, Supplementary Fig. 1G and Supplementary Data 2).

Next, we analyzed AML cells at day 14 or day 30 after induction chemotherapy. In the *NPM1*[mut] cohort, we obtained 849 leukemic cells from 8 patients at day 14, and 469 cells from 7 patients at day 30. Not surprisingly, most cells were obtained from patients with refractory disease (PT01, PT13; $n = 1105$), while few leukemic cells could be recovered from patients who obtained morphologic CR but later relapsed (PT08, PT09, PT10; $n = 171$) or remained in CR (PT06, PT07, PT12; $n = 42$) until latest follow-up (Supplementary Data 1). Post-chemotherapy leukemic cells prominently mapped to the progenitor clusters, with enrichment in the erythroid-like clusters and accumulation of cells bridging the early and myeloid immature-like progenitor area on day 30 (Fig. 2A, B). Next, to investigate coordinated transcriptional responses to chemotherapy, we performed gene set enrichment analysis (GSEA) for hallmark molecular database signatures on post-chemotherapy and diagnosis cells co-clustering within our AML landscape (Fig. 2C). Leukemia sampled on day 14 was

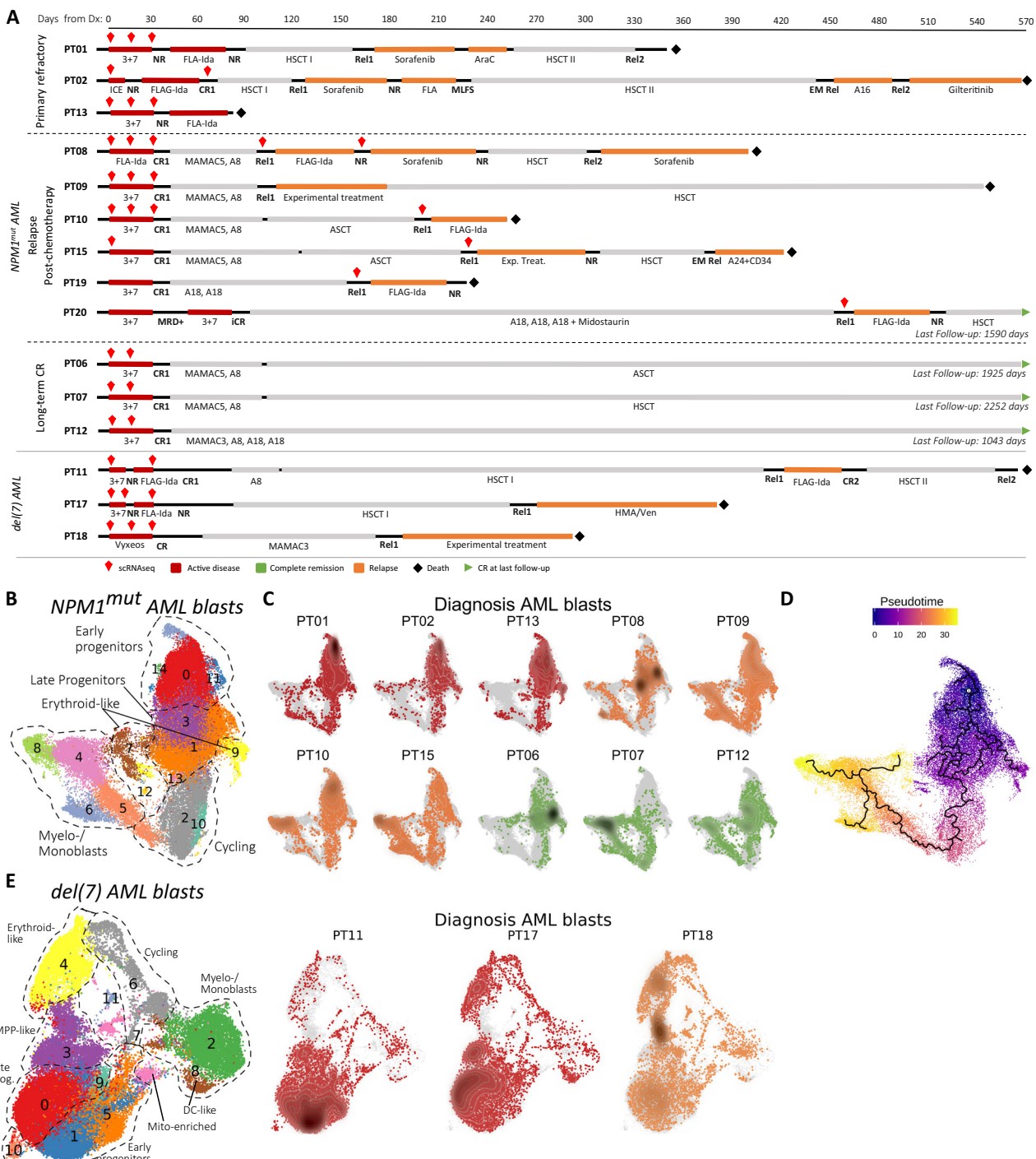

**Fig. 1 | Transcriptional landscape of human AML at single-cell resolution.**
**A** Swimmer plots detailing the patients' clinical course and timepoints selected for scRNAseq (red diamonds). Top horizontal axis: days from diagnosis. CR1, first complete remission. Rel1/2, first/second relapse; EM Rel, extramedullary relapse; NR, no response; MLFS, morphologic leukemia-free state; FLA-Ida, Fludarabine 50 mg/m² D1-5, Cytarabine 1000 mg/m² q12h D1-5, Idarubicin 10 mg/m² D1-3, FLAG-Ida, as before, with granulocyte colony stimulating factor 30MU D0-5; 3 + 7, Cytarabine 100–200 mg/m² D1-7 and Daunorubicin 60 mg/m² D1-3; ICE, Idarubicin 12 mg/m² D1-3, Cytarabine 100 mg/m² D1-7, Etoposide 100 mg/m² D1-5; MAMAC(3) 5; m-Amsacrine 100 mg/m² D1-(3)5, Cytarabine 1000 mg/m² q12h D1-(3)5; A: Cytarabine monotherapy, with the number indicating the cumulative dose in g/m² (e.g. A16, Cytarabine 16 g/m²); HMA: hypomethylating agent; HSCT I/II, first/second allogeneic hematopoietic stem cell transplantation; ASCT, autologous stem cell transplantation. **B** All single-cell RNA sequencing plots are based on uniform

manifold approximation and projection (UMAP) embeddings; x-axis: UMAP_1, y-axis UMAP_2. UMAP plot of *NPM1*$^{mut}$ AML cells colored by unsupervised clustering (resolution 0.6) and labeled according to annotated cell phenotypes. $n = 42,398$ cells. **C** UMAP density plots of each patient's AML blasts at diagnosis within the *NPM1*$^{mut}$ AML landscape. Cells are colored according to the patient's treatment response: dark red for primary refractory, orange for relapse after CR, green for persistent CR. **D** Pseudotime trajectory (black lines) and pseudotime (color gradient) for AML cells at diagnosis projected on UMAP embedding. Trajectory root is set from the centroid of *CD34* transcript expressing cells (white dot). **E** Left: UMAP plot of del(7) AML cells colored by unsupervised clustering (resolution 0.6) and labeled according to annotated cell phenotypes. Right: UMAP density plots displaying the distribution of each patient's AML cells at diagnosis within the del(7) AML landscape. Cells are colored according to the patient's treatment response: dark red for primary refractory and orange for relapse after CR. n = 33,308 cells.

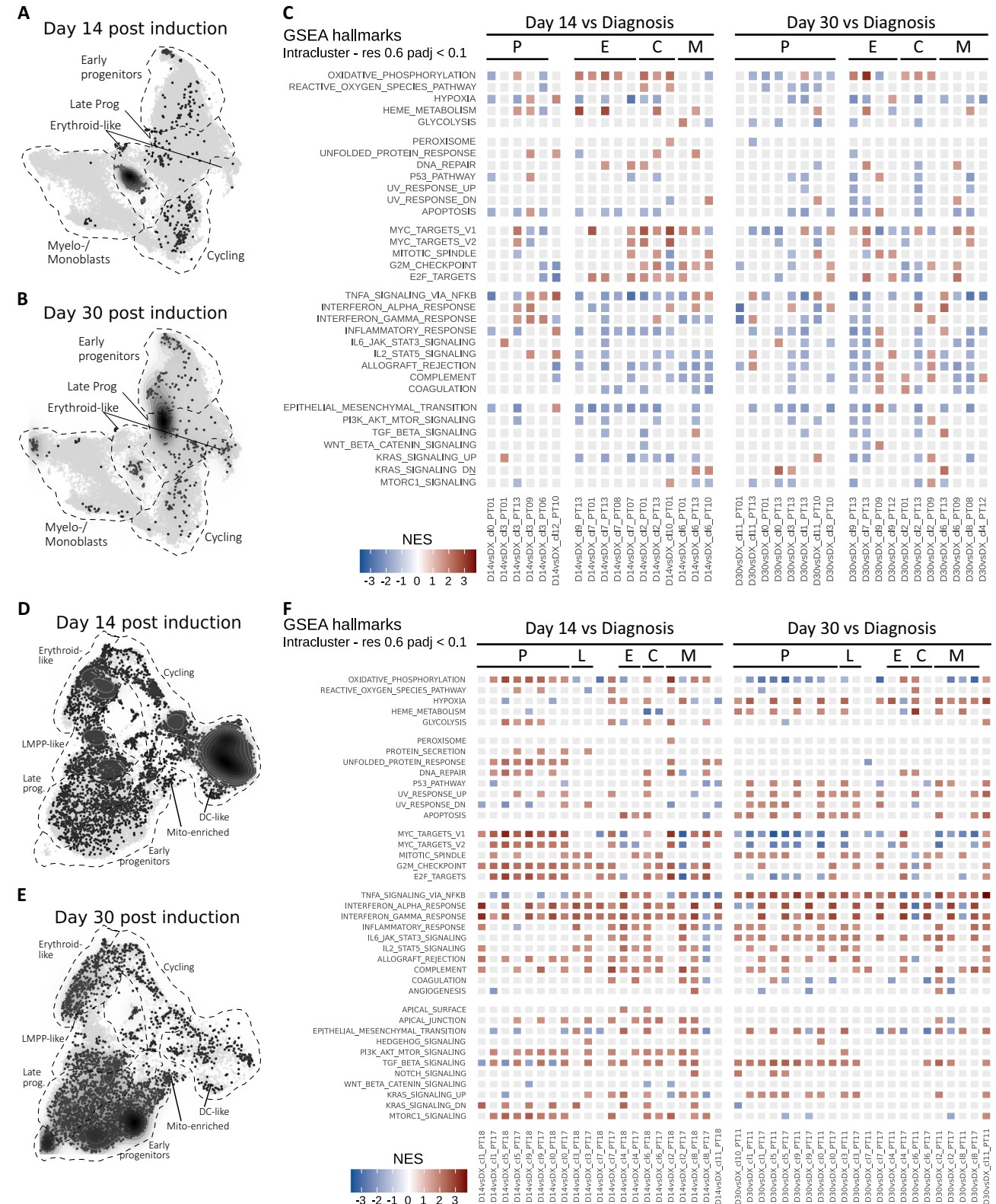

characterized by upregulation of MYC and E2F targets, as well as oxidative-phosphorylation (OxPhos) hallmark signatures as compared to diagnosis in all but the early progenitor clusters, consistent with an early proliferative response. On the contrary, early progenitors showed a depletion of OxPhos and cell cycle-related hallmarks, but enrichment for inflammatory hallmarks (Fig. 2C). On day 30, residual progenitors maintained low OxPhos & cell cycle hallmarks, along with depletion of glycolysis-related hallmarks consistent with the

persistence of a metabolically inactive state, while the inflammatory signature attenuated.

In the del(7) cohort, we obtained 8970 cells at day 14 and 3930 cells at day 30, from 2 patients each. Differently from the *NPM1*^mut cases, del(7) AML accumulated high numbers of differentiated myeloid cells on day 14, which disappeared on day 30 (Fig. 2D, E). At the same time, progenitor cells persisted, with the accumulation of clusters 10, 1, and 5 by day 30 (Fig. 2D, E). GSEA highlighted a prominent proliferative

**Fig. 2 | Single-cell RNA sequencing of paired diagnosis/post-chemotherapy samples reveals an early proliferative response alongside with persisting quiescent progenitor blasts. A** UMAP density plot showing distribution of sequenced *NPM1*^mut AML blasts at day 14 early post-chemotherapy (D14). $n = 849$ cells. **B** UMAP density plot showing distribution of sequenced *NPM1*^mut AML blasts at day 30 post-chemotherapy (D30). $n = 469$ cells. **C** Tile plot of normalized enrichment scores (NES) from gene set enrichment analysis (GSEA) of hallmark terms on differentially expressed genes (DEG) between D14 or D30 and diagnosis (DX) blasts of *NPM1*^mut AML patients within the indicated clusters (cl., see Fig. 1B for unsupervised clustering). Hallmarks (rows) are grouped by semantic similarity. Columns represent single-cluster, single-patient comparisons between the specified timepoints and are grouped by cluster phenotype (P, progenitor blasts; E, erythroid-like; C, cycling; M, myelo/monoblasts) and timepoint (Day 14; Day 30). Non-significant enrichment results are plotted in light grey (Benjamini−Hochberg adjusted *p*-value > 0.1). **D** UMAP density plot showing distribution of sequenced del(7) AML blasts at day 14 early post-chemotherapy. $n = 8970$ cells. **E** UMAP density plot showing distribution of sequenced del(7) AML blasts at day 30 post-chemotherapy. $n = 3930$. **F** Tile plot of normalized enrichment scores (NES) from GSEA of hallmark terms on DEG between D14 or D30 and diagnosis (DX) del(7) AML blasts within the indicated clusters (cl., see Fig. 1E for unsupervised clustering). P, progenitor blasts; L, LMPP-like blasts; E, erythroid-like; C, cycling; M, myelo/monoblasts), timepoint (day 14; day 30). Non-significant enrichment results are plotted in light grey (Benjamini−Hochberg adjusted *p*-value > 0.1).

response on day 14 with upregulation of MYC, E2F targets, cell cycle checkpoints and OxPhos across all progenitor clusters, followed by a depletion of these signatures on day 30 (Fig. 2F). Notably, blasts from del(7) patients showed a persistently elevated inflammatory expression profile (Fig. 2F).

Taken together, longitudinal post-chemotherapy assessment highlighted divergent (for *NPM1*^mut AML) or time-dependent (for del(7) AML) transcriptional responses in AML cell subpopulations characterized by (1) early proliferation within a regenerating, Ox-Phos^high cell compartment and (2) cell cycle quiescence coupled to inflammatory hallmarks within the early progenitor cell compartment.

### Tracking LSCs after chemotherapy in xenograft models by microRNA-126 activity

Next, we made use of a previously described microRNA reporter (Fig. 3A), which allows prospective, flow cytometry-based identification of LSC-enriched AML subpopulations in xenograft models, based on high miR-126 activity[17,22]. We validated this approach on 4 *NPM1*^mut primary AML samples (Supplementary Fig. 2A−C) and confirmed by limiting-dilution serial transplantation that leukemia-initiating cell (LIC) frequency was enriched in the GFP^low/miR-126^high subfraction in all patients, indicating that single parameter selection for miR-126 activity allows up to 20-fold LSC enrichment within *NPM1*^mut AML from a given patient (Fig. 3B and Supplementary Fig. 3A). miR-126 activity strongly correlated with LIC frequency across AML fractions from different patients (Spearman rho 0.93, p-value 0.00013, Fig. 3C), indicating that miR-126 is associated with LSC potential in a quantitative, and not just qualitative fashion. Differently from surface markers with unknown function, miR-126 is directly linked to LSC biology by regulating cell cycle progression through downregulation of targets converging on PI3K/AKT/MTOR and CDK3, thought to act on S/G2/M transit and G0 exit, respectively[17]. Cell cycle analysis on 3 representative diseases confirmed depletion of miR-126^high cells in the S/G2/M phase and enrichment in G0 (PT01) or G1 (PT03, PT16) (Supplementary Fig. 3B). We next hypothesized that the miR-126 reporter represented a useful tool to gain insight into LSC function during induction chemotherapy in xenografts derived from diseases selected for varying initial LSC content ($n = 5$ diseases, 10-25 mice per patient sample; Fig. 3D). After confirmation of AML engraftment by BM aspirate, mice were assigned to a control or chemotherapy cohort (cytarabine 60 mg/kg on days 1−5; daunorubicin 1.5 mg/kg on days 1−3) matched for human CD45^+ engraftment levels within each patient-derived xenograft group (Supplementary Fig. 3E). We analyzed the mouse BM early after chemotherapy (day 8) and showed marked reduction of hCD45^+ blasts in the treated mice (Fig. 3E), with a relative increase of immunophenotypically primitive CD34^+38^- cells indicating that they exhibited relative chemotherapy-resilience (Fig. 3F). While the proportion of immunophenotypically defined LSCs increased, average miR-126 activity of the total blast population was unchanged or decreased (reflected by an increase in miR-126 reporter vector transgene ratio) after chemotherapy (Supplementary Fig. 3F).

However, when looking at the distribution of miR-126 activity across single cells, chemotherapy unmasked a small blast subpopulation with very high miR-126 activity in xenografts derived from 2 patients (PT01 and PT03), while the bulk of AML blasts downregulated miR-126 activity compared to the control cohort, most evident in xenografts from PT01, PT02 and PT15 (Fig. 3G and Supplementary Fig. 3G). Interestingly, the samples from PT01 and PT03 were enriched in the G0 cell cycle phase (Supplementary Fig. 3H) and manifested a higher LIC frequency in post-chemotherapy samples compared to controls, while LIC frequency was reduced in samples from PT15, where miR-126 activity decreased (Fig. 3H).

### miR-126^high subsets are characterized by lymphoid and stem cell gene expression signatures

To gain further insights into the functional differences between miR-126^low and miR-126^high cells in the presence or absence of chemotherapy, we performed gene expression profiling of human AML blasts from the BM of xenografted mice at steady-state and on day 8 after chemotherapy ($n = 4$ diseases, 5−12 mice per patient sample), sorting the highest and lowest 15−20% GFP-expressing populations, respectively (Supplementary Fig. 4A, Supplementary Data 3). After batch correction for patient variability, principal component (PC) analysis on the top 1000 most variable genes (Fig. 4A) showed a separation driven by miR-126 activity (PC1, explaining 24% of the dataset variance) and chemotherapy treatment (PC2, 14% variance). GSEA revealed strong enrichment for previously reported HSC and LSC signatures in miR-126^high blasts, whereas progenitor cell-, myeloid differentiation- and cell cycle signatures were enriched in miR-126^low blasts (Fig. 4B). Moreover, the prognostically relevant "LSC17" gene score[26], as well as the LSC-associated gene *ADGRG1*[27] were more highly expressed in the GFP^low/miR-126^high subpopulations (Supplementary Fig. 4B, C). We then investigated gene ontologies−biological processes (GO-BP) differentially enriched within miR-126^high and miR-126^low blasts (Fig. 4C). In addition to ontologies related to cell contacts and signaling axes with known relevance in AML (G-protein coupled receptors, phosphatidylinositol-mediated signaling, JAK-STAT), GO-BPs characterizing miR-126^high fractions prominently contained lymphoid categories, and we confirmed strong and consistent upregulation of bona fide lymphoid genes, including *RAG1*, *RAG2*, *CD79A*, *CD79B*, *ZAP70*, *IGH* genes and *CD7* in the individual miR-126^high samples across all 4 patients, irrespective of chemotherapy (Supplementary Fig. 4D). On the other hand, miR-126^low blasts were enriched for features associated with differentiated myeloid effector cells and cell cycle (Fig. 4C).

We next interrogated the transcriptomic data for chemotherapy-related effects on AML blasts. Upon therapy, we found, in both miR-126^high LSCs and miR-126^low blasts, an enrichment of hallmarks related to inflammatory signaling, apoptosis, angiogenesis, KRAS signaling, epithelial-mesenchymal transition and heme metabolism (Fig. 4D, Supplementary Fig. 4E). Recent work has investigated the induction of cellular senescence in AML cells by chemotherapy, associated with cellular reprogramming through activation of the Notch/Wnt

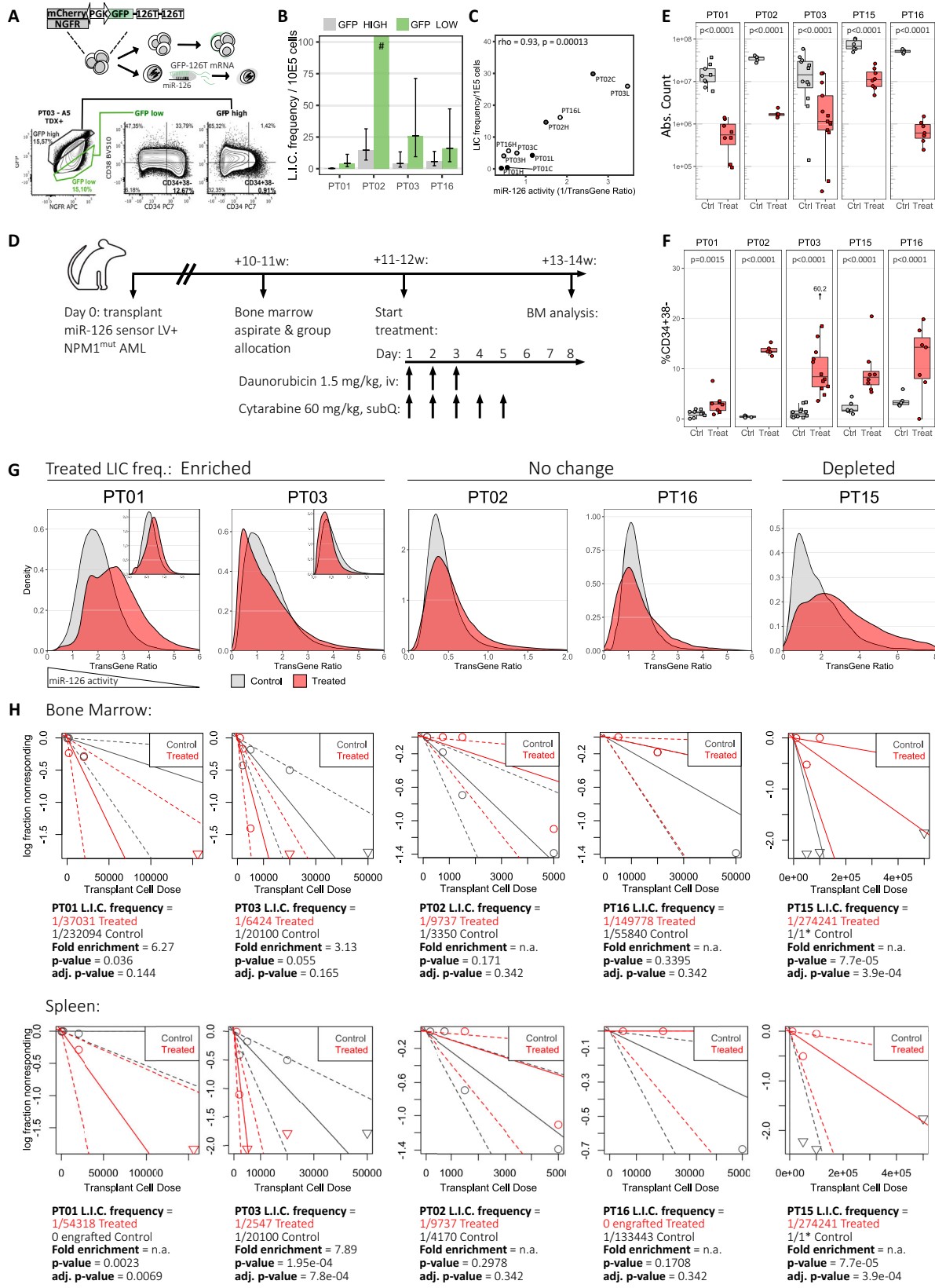

pathways[15]. Our data support a coherent induction of senescence-associated gene signatures in AML blasts exposed to chemotherapy in the xenograft model (Fig. 4E), which did not, however, result in a general increase in LSC marker expression (Supplementary Fig. 4B, C).

Overall, miR-126[high] cells in *NPM1*[mut] AML were characterized by a gene signature enriching both for previously reported HSC/LSC and lymphoid genes, at baseline and after chemotherapy. Chemotherapy imposed an inflammatory and senescence-like response on AML blasts.

**Fig. 3 | Lentiviral miRNA-126 reporter allows prospective enrichment of LSCs in patient-derived AML xenografts and reveals AML subpopulations with diverging miR-126 activity upon chemotherapy. A** miR-126 reporter vector schematic: GFP and mCherry/NGFR are coordinately expressed by a promoter with bidirectional activity. The GFP 3′UTR is tagged with 2 miR-126 target sites resulting in miR-126high cells downregulating GFP compared to the untagged control (mCherry/NGFR) and being identifiable by flow cytometry within the GFPlow gate. **B** Limiting-dilution transplantation estimate of leukemia-initiating cell (L.I.C.) frequency for sorted GFPhigh/miR-126low and GFPlow/miR-126high populations. Barplot: mean, error bars: ±95% CI. # denotes failure to reach a limiting dose for PT02's GFPlow subpopulation. **C** Correlation of L.I.C. frequency and miR-126 activity (1/TransGene Ratio) in xenotransplanted AML (*n* = 10). Spearman rho correlation coefficient and *p*-value are reported. **D** Mice engrafted with miR-126 reporter transduced *NPM1*mut AML were allocated into treated or control groups matching for AML burden (Supplementary Fig. 3E). Treated mice received daunorubicin/cytarabine chemotherapy. Control mice received equivalent saline doses. Mice were euthanized on day 8, when persisting AML was analyzed. **E** Day 8 absolute number of AML blasts in the BM of control (Ctrl) or treated (Treat) mice. *n* = 78 PDX from 5 patients over 7 independent experiments. Comparisons were performed by linear-mixed effects (LME) models accounting for mice injected with the same donor and experimental replicates. Adjusted *p*-values with Holm's correction are reported. Data are presented as standard Tukey boxplots: center line: median; box: interquartile range; whiskers: IQR*1.5. **F** Percentage of transduced hCD34+38- immature blasts in the BM of Treat or Ctrl PDX. *n*, statistical testing and boxplot definition as per panel E. **G** miR-126 sensor vector TransGene Ratio (1/miR-126 activity) of day 8 residual BM AML from Treat (red) or Ctrl (grey) mice. Experimental replicates for PT01 and PT03 are displayed in the plot inlets. **H** Limiting-dilution transplant of day 8 BM AML cells recovered from Treat or Ctrl PDX. Proportion of non-engrafted mice (log) as a function of transplanted cell doses for the Treat (red line) or Ctrl group (grey line), with respective CI (dotted lines). Down-pointing triangles indicate doses with 100% engraftment. L.I.C. frequency estimation and p-values are reported below each plot. Engraftment was evaluated in the BM (top) or spleen (bottom) of recipient mice. Source data are provided as a Source Data file.

## Single-cell RNA sequencing of xenografts reveals persistence of miR-126high LSCs with enhanced stemness features upon in vivo chemotherapy

To gain further insights into the chemotherapy response of *NPM1*mut AML subpopulations represented in the xenograft model, we performed scRNAseq on BM AML blasts from control or treated mice on day 8 from start of treatment (*n* = 4 AML, 3–7 mice per condition, *n* = 96,931 cells) (Fig. 5A, Supplementary Fig. 5A, Supplementary Data 4). After batch correction and unsupervised clustering, marker gene analysis coupled with reference-based annotations[28,29] allowed us to discriminate between clusters enriched in HSC-like blasts (cl. 1: *FAM30A, EGFL7, CD34, HOPX*), progenitor blasts (cl. 0: *IGFBP2, MYC, MCM4,5,6,7, NASP, CDCA7*; cl. 8: *MYC, CENPF, PTTG1, HMGB1, CALR*), cycling progenitor blasts (cl. 2, 9), cycling GMP-like blast (cl. 4), quiescent GMP-like blasts (cl. 7), erythroid-like blasts (cl. 5, 6), myeloid-like blasts (cl. 3, 10) and plasmacytoid DC-like blasts (pDC, cl. 11) (Fig. 5A, Supplementary Fig. 5A–C, Supplementary Data 5).

To investigate population-specific dynamics upon chemotherapy, we compared the proportion of each cluster to control xenografts. While steady-state control AMLs displayed enrichment for progenitor and cycling populations, chemotherapy radically reshaped AML, with an expansion of the more differentiated myeloid and, especially, erythroid-like blasts (Fig. 5B, E). On the other hand, the proportion of HSC-like blasts remained unchanged. We next mapped miR-126high blasts onto our scRNAseq landscape, using 2 complementary approaches. First, miR-126 sensor-transduced, sorted GFPlow/miR-126high blasts localized most strongly on the HSC-like cluster 1, with some signal present also in the activated/cycling progenitor clusters, the pDC-like and the erythroid clusters (Fig. 5C). The latter two cell types have been previously associated to miR-126 activity in healthy hematopoiesis, even though to a lower degree than HSC[22,30]. Secondly, we mapped the previously identified transcriptional signature of miR-126high cells (126High, Supplementary Data 3), based on 989 upregulated genes (FDR < 0.05) with respect to miR-126low blasts, on our scRNAseq data reaching similar conclusions (Fig. 5D). Generally, chemotherapy-exposed AML displayed a bimodal distribution of 126High expression, reminiscent of our previously described miR-126 activity reporter (Fig. 3G), with decreased average expression (right shift) from most blasts while preserving a demarcated subpopulation of blasts with high 126High module score (Fig. 5F).

To explore cluster-specific transcriptional responses to chemotherapy, we calculated enrichment scores for GSEA hallmarks for each single cluster, comparing, within a given cluster, treatment-exposed to treatment-naïve blasts. In line with our bulk RNA sequencing results, chemotherapy-induced inflammatory and senescence-associated responses mainly driven by TNF/NFKB and interferon signaling, along with p53, UV and DNA repair pathway activation across most AML clusters (Fig. 5G, H). Moreover, we observed divergent chemotherapy responses in the progenitor/immature compartments, reminiscent of the day 14 samples from *NPM1*mut patients (see Fig. 2C). The progenitor clusters 0 and 8 showed enrichment for energy-related pathways and cell cycle genes. EdU (5-ethynyl-2′-deoxyuridine) incorporation confirmed early proliferation of AML blasts after chemotherapy also within the miR-126high LSC-enriched subpopulation (Fig. 5I), which nonetheless maintained a constantly high proportion of cells in the G0 cell cycle phase (Fig. 5J).

scRNAseq may provide a higher resolution to further dissect the miR-126high compartment. We hypothesized that the 126High+ HSC-like cluster 1, which differed from most other clusters by downregulation of OxPhos and MYC proliferative responses upon chemotherapy exposure, may contain dormant LSCs with increased resistance to stress-induced activation (Fig. 5G). We therefore searched, within this cluster, for genes specifically enriched in chemotherapy-exposed samples as compared to controls. Several LSC-associated genes, including *CD34, CD36, CD99, GUCY1A1, TNFRSF4*, as well as *DNTT*, were significantly more expressed after chemotherapy (Fig. 5K, Supplementary Data 5). Next, we increased the clustering resolution thus subdividing cluster 1 into 3 subclusters, denominated 5′, 6′ and 19′ (Supplementary Fig. 5D). Subcluster 5′ was depleted, while subclusters 6′ and 19′ were enriched by chemotherapy (Fig. 5L). Again, we found multiple LSC-associated genes among the top 10 marker genes characterizing cluster 6′ (Fig. 5M, Supplementary Data 5). Moreover, the HSC latency-associated gene *INKA1*[31] was highly expressed in clusters 6′ and 19′, while the cell cycle kinase *CDK6*[32], a robust marker for active HSC[32–34] was mostly detected in cluster 5′, suggesting that the former two clusters represented more dormant LSC states.

We treated miR-126 reporter+ AML cells from PT16 in vitro with IACS-010759 (IACS)[35], a selective mitochondrial electron transport chain complex I-inhibitor, alone or in combination with AraC chemotherapy. IACS-treated cells had higher miR-126 activity (Supplementary Fig. 5E–G), consistent with preferential depletion of OxPhoshigh miR-126low progenitors. IACS-treated cells showed strongly enhanced repopulation activity when transplanted into NSG mice as compared to the mock-treated group, underlining the functional relevance of these miR-126high OxPhoslow LSCs, at least in *NPM1*mut AML (Fig. 5N).

## miR-126high LSCs at diagnosis correlate with AML outcome

Next, we mapped the miR-126high gene signature (126High) (Supplementary Data 1) from the AML xenograft study onto the patients' single-cell landscapes described in Fig. 1. Cells positive for 126High were predominantly found in the early immature progenitor clusters in both *NPM1*mut and del(7) AML (Fig. 6A). This may be expected from the presence of gene sets enriched in HSC/LSC signatures. However, also

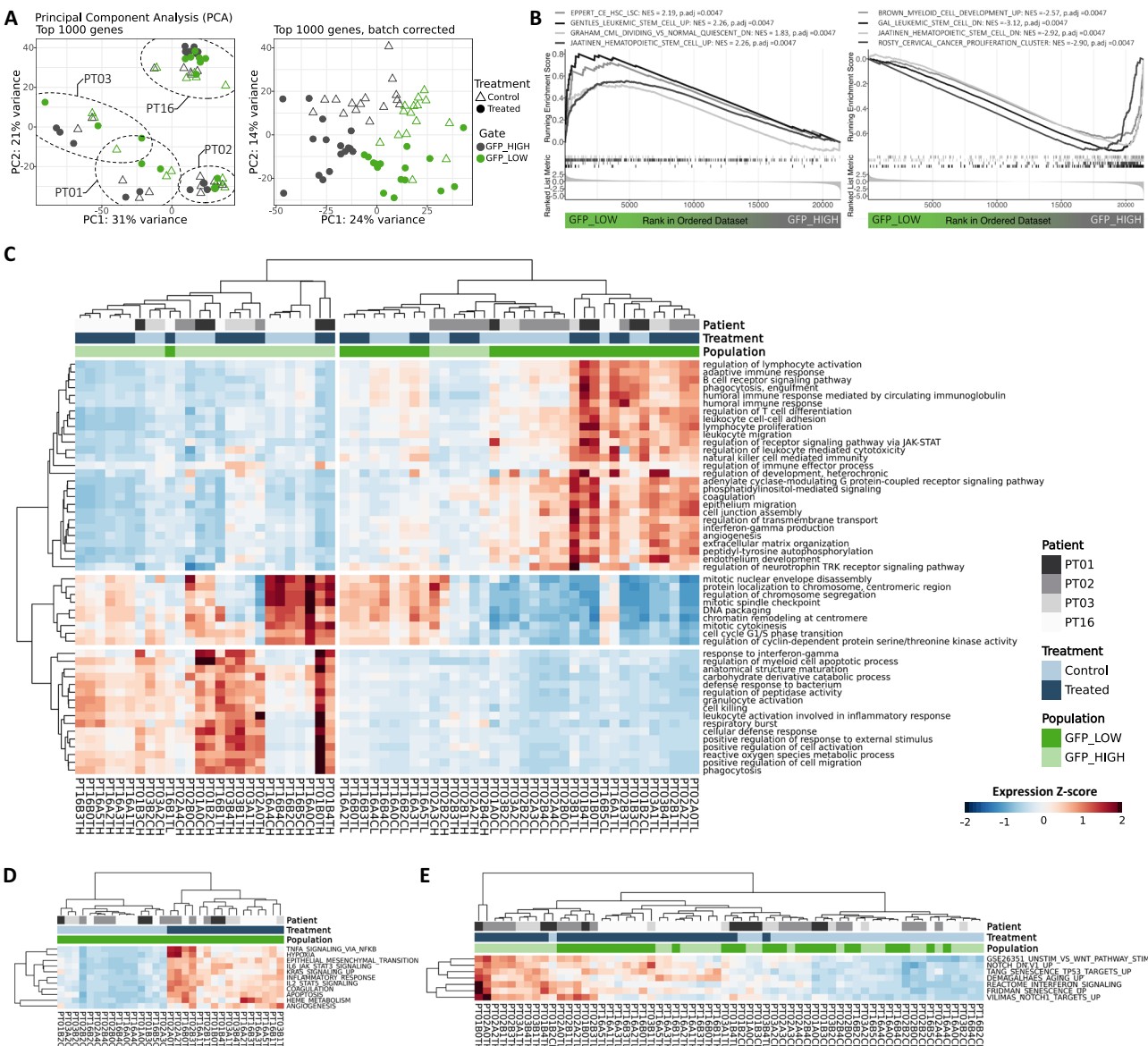

**Fig. 4 | miR-126^high LSCs co-express stem cell and lymphoid transcriptional profiles and up-regulate inflammatory signaling upon chemotherapy.**
**A** Principal component analysis (PCA) on the top 1000 most variable genes prior to (left) or after batch correction for patient-derived variability (right). Samples are coded for sorted GFP subpopulations (green: GFP^low, grey: GFP^high) and treatment groups (circles: control, triangles: chemotherapy-treated mice). n = 31 PDX from 4 patients. **B** Enrichment plot of Gene Set Enrichment Analysis (GSEA) of indicated gene lists within DEGs between GFP^low/miR-126^high and GFP^high/miR-126^low blasts. Normalized Enrichment Score (NES) and q-value are reported for each signature. **C** Heatmap of agglomerated z-scores for selected Gene Ontology−Biological Processes (GO-BP) enriched terms (rows), grouped by semantic similarity, from the comparison of GFP^low/miR-126^high vs. GFP^high/miR-126^low populations in over-

representation analysis (ORA). Each column represents a sorted sample. Treatment group, patient ID and GFP population are annotated on top. n = 62 samples from 31 PDX from 4 patients. **D** Heatmap of agglomerated z-scores for enriched Hallmark gene sets (MSigDB H group) (rows) from the chemotherapy-treated vs. control comparison in GFP^low/miR-126^high blasts only. Each column represents a sorted sample. Treatment group, patient ID and GFP population (GFP^low) are annotated on top. n = 31 samples from 31 PDX from 4 patients. **E** Heatmap of agglomerated z scores for enriched senescence-associated gene signatures (rows) from the chemotherapy-treated vs. control blast comparison. Each column represents a sorted sample. Treatment group, patient ID and GFP population (GFP^low or GFP^high) are annotated on top. n = as per panel **A**. Source data are provided as a Source Data file.

the gene sets related to lymphocyte function co-localized with the LSC profile (Supplementary Fig. 6A). On the contrary, the signature containing the genes depleted in miR-126^high cells (126Low) overlapped with the region containing monocyte-like CD14^+/64^+ blasts, supporting a differentiation gradient from miR-126^high LSCs to miR-126^low blasts (Supplementary Fig. 6B). While the LSC17 gene signature also mapped to the immature-like area, it provided a less clear demarcation of intra-tumor heterogeneity compared to our miR-126 derived gene lists (Supplementary Fig. 6C). Notably, in pre-treatment diagnosis samples, our 126High gene set was expressed both in a higher proportion of

AML cells and to higher levels in patients with refractory disease or early relapse compared to patients, who obtained long-term remission (Fig. 6B).

To further confirm that the 126High gene signature measured at diagnosis was associated with patient outcome, we analyzed publicly available datasets encompassing multiple AML subtypes with respect to overall survival (OS). We first reduced our signature to 24 informative genes by selecting those with a significant correlation (Spearman rho >0.3, Bonferroni *p* < 0.05) between their expression and that of the overall gene signature in our *NPM1*^mut scRNAseq dataset

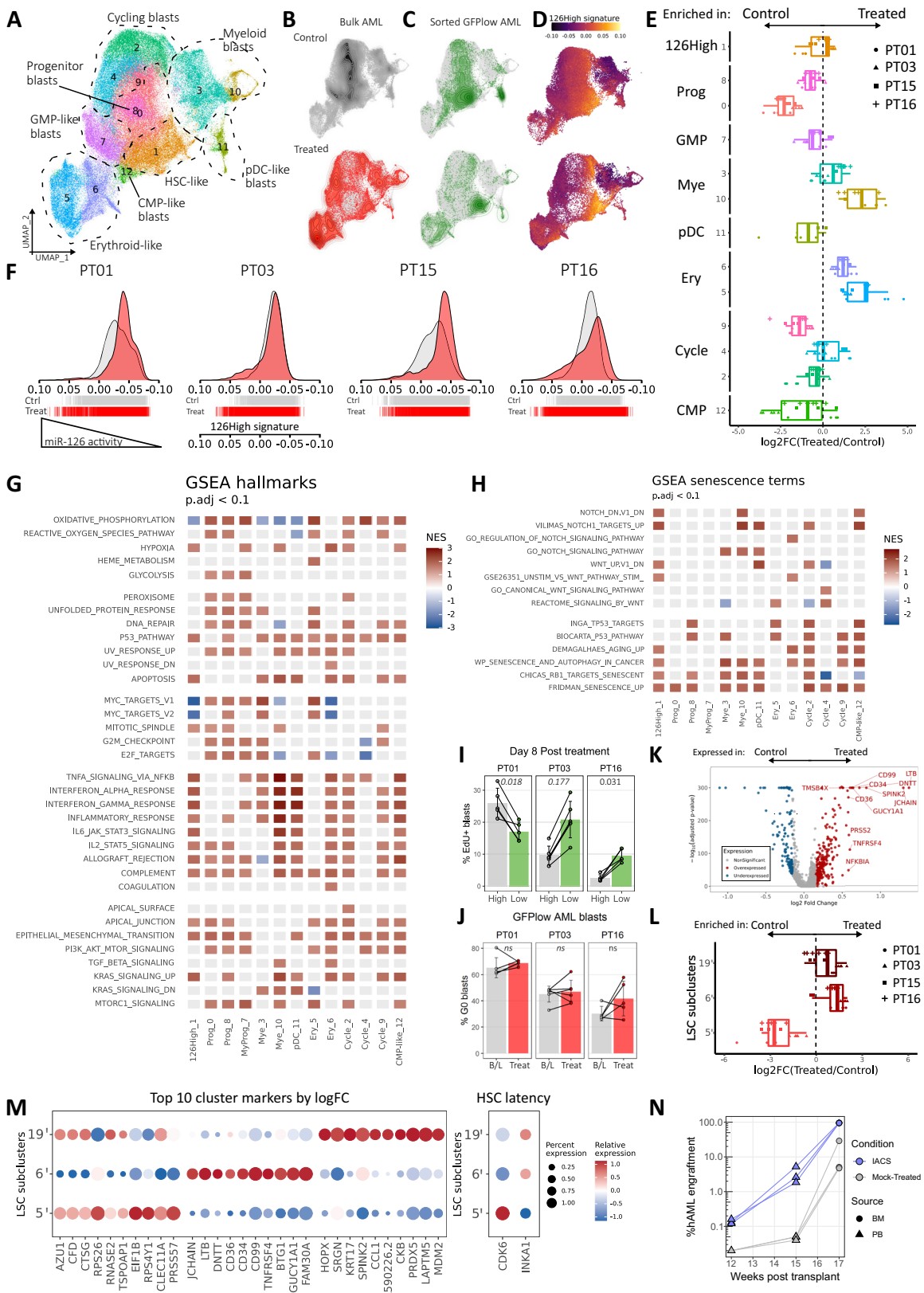

(Supplementary Data 3). Next, we estimated Cox's proportional hazard model with lasso penalty on the standardized intensity of the transcripts on 3 independent AML datasets separately, encompassing $n = 767$, $n = 337$ and $n = 161$ non-M3 AML patients, respectively. In all three datasets, the two corresponding risk groups (obtained with the median split of the linear combination of the genes of the model)

showed clearly separated survival curves (log-rank $p < 0.0001$) (Supplementary Fig. 6D, Supplementary Data 6). To identify and validate our risk groups, we used $n = 767$ patients from the microarray study as a training set, and $n = 337$ patients from the OSHU RNAseq dataset as the testing set, where 20 of the 24 signature genes could be interrogated, as compared to 15 in the TCGA dataset. To compare gene

**Fig. 5 | Single-cell RNA sequencing of xenografts reveals persistence of miR-126high LSCs with enhanced transcriptional features of stemness upon in vivo chemotherapy. A** UMAP plot of day 8 BM AML blasts from treated or control PDX, colored by unsupervised clustering at resolution = 0.6. Dashed lines show manual annotation of clusters. n = 96,931 cells from 35 PDX from 4 patients. **B** UMAP density plots of AML blasts from control (top, black) or treated (bottom, red) xenografts. **C** UMAP density plots of sorted GFPlow/miR-126high blasts from control (top) or treated (bottom) xenografts. **D** 126High signature expression in scRNAseq AML blasts from control (top) or treated (bottom) xenografts. **E** Enrichment (right) or depletion (left) of scRNAseq cluster abundance in treated AML compared to the average abundance of each cluster in control xenografts from the same donor. n = 35 PDX from 4 patients over 4 independent experiments. Data are presented as standard Tukey boxplots (center line: median, box: interquartile range, whiskers: IQR*1.5). **F** Ridge plot of 126High signature expression across cells in scRNAseq from treated (red) or control (grey) xenografts. Single-cell events are displayed in the underlying rug plots. **G** Tile plot of normalized enrichment scores (NES) from GSEA of hallmark terms on DEGs between blasts from treated vs. control xenografts within the indicated clusters (see Fig. 5A for unsupervised clustering). Hallmarks (rows) are grouped by semantic similarity. Columns are ordered by decreasing 126High expression. Non-significant enrichment results are plotted in light grey (Benjamini–Hochberg adjusted p-value > 0.1). **H** Same as **G** but for senescence-associated gene signatures. **I** Percent of day 8 EdU-incorporating BM blasts within sorted GFPhigh or GFPlow AML cells from treated PDX. n = 16 PDX from 3 patients over 3 independent experiments. Comparisons between paired groups were performed with the two-sided paired Wilcoxon test. For PT01 and PT03 (n ≤ 5 with a minimum two-sided p-value=0.0625), a bootstrap sampling-based nonparametric test was employed and p-values are marked in italics. n = 16 PDX from 3 patients over 3 independent experiments. Data are presented as mean ± SD. **J** Percent of G0 GFPlow/miR-126high blasts at baseline (B/L) or after in vivo chemotherapy (Treat). n and statistical comparisons and data presentation as per panel **I**. **K** Volcano plot of DEGs between treated vs. control in cluster 1 AML blasts. Top 12 genes by log2FC with increased expression in treated blasts are labelled (red). **L** Enrichment (right) or depletion (left) of LSC subclusters in treated AML compared to the average abundance of each cluster in control xenografts from the same donor. n = 35 PDX from 4 patients over 4 independent experiments. Standard Tukey boxplots: center line: median, box: interquartile range, whiskers: IQR*1.5. **M** Expression of top 10 marker genes for LSC subclusters 5′, 6′, 19′ and of HSC-latency-associated genes in LSC subclusters. Dot size reflects percentage of cells expressing the queried gene (columns), color scale represents relative gene expression within the subclusters (rows). **N** Longitudinal evaluation of AML engraftment within mice transplanted with PT16 AML blasts treated ex-vivo with an OxPhos inhibitor (IACS-010759) or solvent (Mock-treated). n = 6 PDX from 1 patient over 1 experiment. Source data are provided as a Source Data file.

expression data obtained by different platforms, we performed a stringent quartile-based categorization of the expression data of each patient into −1 (downregulated, below the 1st quartile), 0 (normal, between the 1st and 3rd quartile) and 1 (upregulated, above the 3rd quartile). By applying the Cox's proportional hazards model with lasso penalty on the categorized expression data of the training set, we identified two risk groups with clearly demarcated OS (log-rank p < 0.0001) and confirmed a significant association with OS (log-rank p = 0.0343), also when applied to the test cohort (Fig. 6C).

## AML cells mapping to miR-126high LSC clusters enrich during chemotherapy and at relapse

Next, we sought to more precisely localize miR-126high LSCs within the early immature progenitor compartment in longitudinal patient samples throughout induction chemotherapy. To increase the resolution within the early immature compartment of NPM1mut AML, we performed subclustering of the early progenitor clusters at diagnosis (cl. 0, 11, 14) (Fig. 6D). Expression of the top 10 marker genes from the xenograft LSC subcluster 6′, which was enriched after chemotherapy (see Fig. 5L), indicated subclusters 0″ and 1″ as the respective counterparts in patients (Fig. 6E). Indeed, these 2 subclusters showed strongest expression of the 126High signature (Fig. 6E), were INKA1+CDK6low/neg and expressed genes implicated in WNT-MYC self-renewal programs (TCF4)[36,37], inflammatory and anti-apoptotic responses (TNFRSF4)[38], epigenetic/post-transcriptional gene regulation (NEAT1)[39–41] and niche interactions (CD36)[42] (Supplementary Fig. 6E, Supplementary Data 2). Notably, early progenitor cells in diagnosis samples from patients obtaining a persistent CR were depleted of cells belonging to subclusters 0″ and 1″, in stark contrast to diagnosis samples from relapse/refractory patients (Fig. 6D). Chemotherapy radically reshaped the LSC compartment, with a progressive increase of 126High signature positive cells and almost exclusive detection of cells within the miR-126high LSC cluster 1″ in day 14 and day 30 samples (Fig. 6D). Similarly, within the immature-like clusters (cl. 10, 1, and 5) enriched after chemotherapy in the del(7) cohort (see Fig. 2E), cluster 10 had the highest expression of the 126High signature and the PDX LSC markers from subcluster 6′ (Fig. 6F).

To further corroborate the relevance of these 126High LSCs, we performed scRNAseq on relapse samples from 4 patients with NPM1mut AML. Distribution of 126High signature expression was shifted towards higher module scores in the relapses, including 2 cases where paired diagnosis-relapse samples were available (Fig. 6G). Notably, while 126High LSC were mostly predicted to be non-cycling in the diagnosis samples of NPM1mut AML, 3 out of 4 relapse samples showed significant cell cycle activity also in the 126High compartment (Fig. 6H). These data suggest a co-existence of stemness and proliferation programs in relapsed disease, as opposed to diagnosis, as also reflected in the GSEA hallmarks showing enrichment for proliferative, oxidative phosphorylation, DNA repair pathways, and depletion of inflammatory, apoptotic and p53 responses throughout all clusters (Fig. 6I Relapse vs Diagnosis).

PT08 underwent reinduction chemotherapy (FLAG-IDA) and showed disease persistence, with the emergence of a complex karyotype. We performed scRNAseq on NPM1mut blasts collected on day 30 of reinduction chemotherapy. Persistent disease was further enriched for 126High+ cells (Fig. 6G, J), which now constituted the majority of leukemic cells and were predicted to be mostly in a quiescent state (Fig. 6H) with enrichment for hypoxia, stress responses, inflammatory and KRAS signaling, and depletion of proliferative pathways and oxidative phosphorylation (Fig. 6I, Rel NR vs Rel), reproducing the behavior observed in NPM1mut persistent HSC-like blasts at day 14 (see Fig. 2C) and of del(7) progenitor blasts, which reacquired quiescence on day 30 post-induction chemotherapy (see Fig. 2F).

Overall, our data support the notion that LSC states persist throughout early chemotherapy in patients, may be exploited as markers for risk prediction and are enriched at relapse.

## Discussion

We here performed extensive single-cell analysis of NPM1mut and del(7) AML populations from 15 representative patients at multiple timepoints, uncovering early patterns of response within the blast and LSC compartments following induction chemotherapy. We show that LSCs persist after induction chemotherapy in patients with treatment failure, exhibiting inflammatory gene signatures and consolidating a miR-126high state. The 2 molecular AML subtypes we studied differed in their hierarchical structure, with del(7) AML showing a more shallow, stem cell dominant hierarchy and NPM1mut AML maintaining a myeloid differentiation trajectory under homeostatic conditions, the latter more closely mimicking non-malignant hematopoiesis. Indeed, evidence for a persisting, OxPhoslow quiescent LSC state after chemotherapy was mostly evident within NPM1mut samples. Very recently, a quiescent LSPC cellular state has been described in scRNAseq of AML patient samples[43]. Deconvolution analysis of bulk RNA sequencing from functionally characterized AML samples has established a correlation

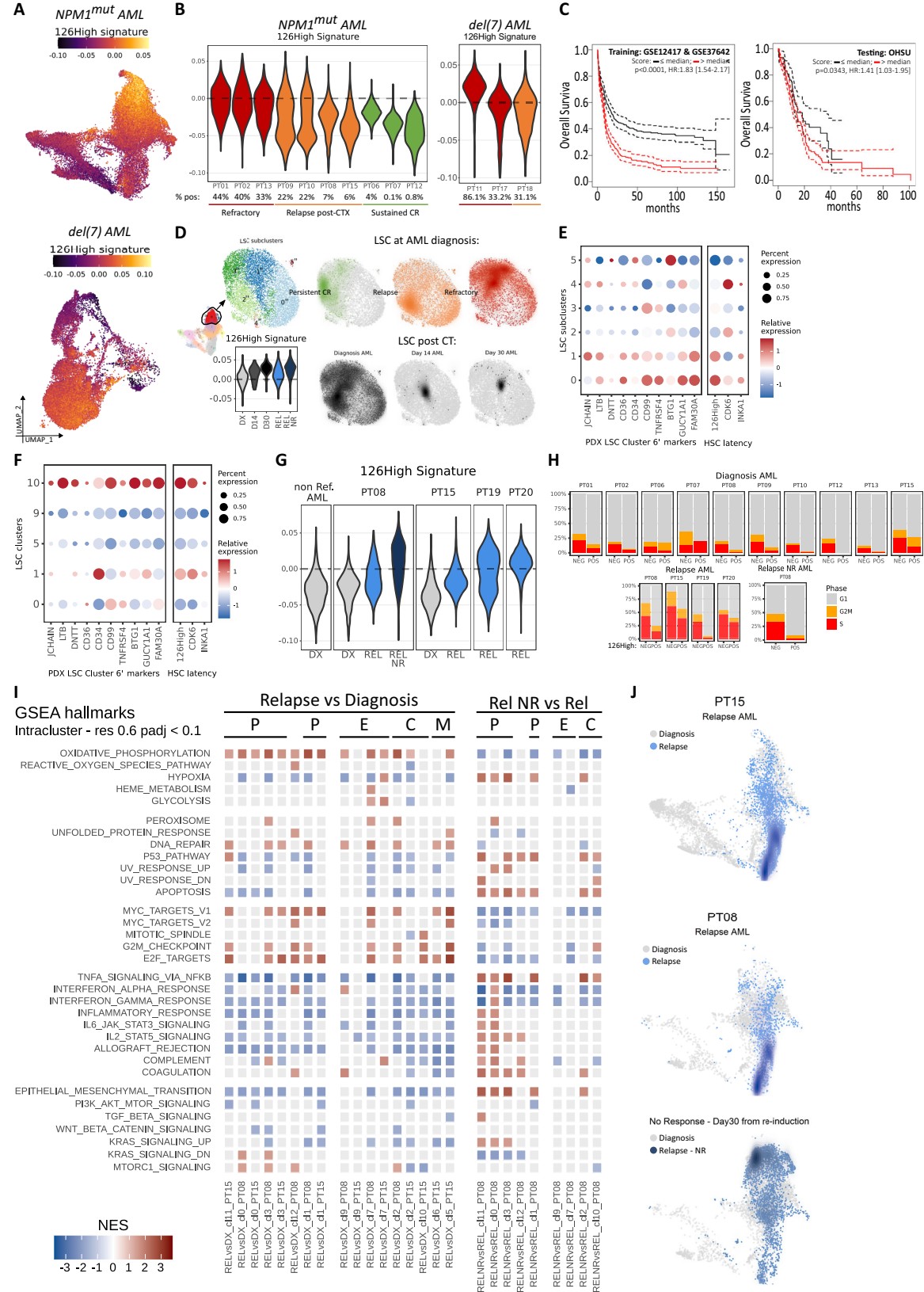

**Nature Communications** | (2023)14:1285

between this quiescent state and LIC activity across a broad range of AML subtypes[43], suggesting a more general relevance of our findings.

By leveraging on miR-126 as a key LSC master regulator[17], we demonstrate prospective LSC enrichment based on miR-126 activity from AML xenografts, with a tight association between miR-126 activity and leukemia-initiating capacity. Subpopulation-specific

transcriptional profiling of miR-126[high] LSCs uncovered, along with canonical stem cell programs, a previously underappreciated bias towards lymphoid transcriptional programs and expression of genes involved in the molecular machinery for V(D)J rearrangement (i.e. *DNTT*, *RAG2*). Recently, a specific role for terminal deoxynucleotidyl transferase (*DNTT*) in the acquisition of *NPM1* mutations and *FLT3*

**Fig. 6 | miR-126[high] gene signature identifies a subset of chemotherapy-resilient LSCs enriched in refractory and relapsed AML. A** Mapping of 126High signature on UMAP plots of *NPM1*[mut] (top, *n* = 32,194 cells) or del(7) (bottom, *n* = 21,650 cells) AML cells at diagnosis. **B** Violin plots of 126High signature expression at diagnosis for each patient, colored by treatment response (dark red: primary refractory, orange: early relapse, green: persistent CR). Percent of cells with 126High > 0 are reported below each violin plot. **C** Kaplan–Meier plot for overall survival of AML patients within the training (GSE12417 *n* = 240 and GSE37642 *n* = 527 newly diagnosed AML patients) and testing (OSHU) gene expression datasets. Log rank test *p*-value and Cox's proportional hazards model with lasso penalty hazard ratios (HR) with 95% confidence interval are reported. Red line: 126High signature > median; black line: 126High signature ≤ median. **D** Top-left: UMAP plot of LSC subclusters colored by unsupervised clustering at resolution = 0.6. Bottom-left: Violin plot of 126High module scores within LSC subclusters. Right: Density UMAP plot of LSCs at diagnosis grouped by outcome categories (green: persistent CR, orange: early relapse, dark red: primary refractory) or at sampled timepoints (diagnosis, Day 14 and Day 30). **E** Expression of the top 10 marker genes for LSC subcluster 6′ from the PDX scRNAseq dataset, HSC latency genes and 126High signature across *NPM1*[mut] AML LSC subclusters (0″, 1″, 2″ 3″, 4″ and 5″). Dot size represents the percentage of cells expressing the queried gene (columns), color scale represents relative gene

expression within the subclusters (rows). **F** Same as **E** for del(7) AML LSC clusters (0, 1, 5, 9 and 10). **G** 126High signature module score distribution within a balanced pool of AML blasts at diagnosis from all non- refractory patients (non Ref. AML – DX), in paired DX-relapse (REL) -post-reinduction (REL NR) samples from PT08, in paired DX-REL samples from PT15 and in REL samples from PT19 and PT20. **H** Predicted cell cycle phase distribution within sequenced AML blasts grouped by 126High signature expression (126High > 0 POS, 126High ≤ 0 NEG) in DX, REL or REL NR samples from the indicated patients. **I** GSEA on DEG between REL and DX (left) or REL NR and REL (right) in *NPM1*[mut] AML blasts assigned to the indicated clusters (cl., see for unsupervised clustering). The tile plot shows the normalized enrichment scores (NES), where rows are hallmark terms and columns represent single-cluster, single-patient comparisons between the specified timepoints. Hallmarks (rows) are grouped by semantic similarity. Columns are grouped by cluster phenotype (P, progenitor blasts; E, erythroid-like; C, cycling; M, myelo/monoblasts) and timepoint. Non-significant enrichment results are plotted in light grey (adjusted *p* > 0.1). **J** Top: UMAP density projection of blasts at relapse (blue) or diagnosis (grey) on the *NPM1*[mut] AML landscape for PT15. Bottom: UMAP density projection of blasts at relapse (blue), relapse refractory to reinduction (dark blue), or diagnosis (grey) on the *NPM1*[mut] AML landscape for PT08.

internal tandem duplications has been hypothesized in AML[44,45], and our data linking this gene specifically to the LSC compartment may further support a role of LSCs for the development and progression of AML. Our findings also extend the rationale for investigating lymphoid marker-directed immunotherapies, e.g. against CD7[46] and TNFRSF4 (OX-40), as their expression within the LSC compartment may be more pervasive than currently thought.

While some recent data have supported a link between LSCs and relapse[7], few studies have addressed the molecular mechanisms and dynamics of early therapy resistance[13–15,47]. Administration of cytarabine chemotherapy to mice xenografted with AML has highlighted the emergence of a leukemia regenerative response characterized by oxidative phosphorylation-dependent, proliferating blasts devoid of stem cell signatures, termed leukemia-regenerating cells (LRC)[13,14]. In our hands, we confirmed the presence of a proliferating, OxPhos[high] AML population, both in xenografts and in patients following induction chemotherapy. Single-cell analysis, both by the miR-126 reporter and unbiased scRNAseq in xenografts and patients, provided us with a high cellular resolution. In *NPM1*[mut] AML, we were able to discriminate a regenerative response within the more committed blast compartment from a rare subpopulation of LSCs that expressed master regulators of a WNT-MYC developmental program[37], showed an inflammatory response, persisted in a quiescent state upon chemotherapy and correlated with increased LIC frequency in serial xenografts, which would have been missed if residual AML were analyzed in bulk. This divergent response between committed progenitors and LSC is reminiscent of recently published work in solid cancer, where two distinct types of persister cells were described to emerge from distinct lineages: non-proliferating cells, which expressed IFNα & NOTCH signatures and relied on cholesterol metabolism, and proliferating cells dependent on fatty acid oxidation[48].

The combined induction of inflammation and cell cycle arrest in response to stress have been portrayed as cardinal features of premature senescence[49]. Recently, cellular senescence has been described to induce reprogramming towards cancer stemness through the WNT/ β-catenin axis in several mouse models[16], and senescence-like resilient cells have been linked with AML recurrence[15]. Our xenograft data suggest that senescence-like responses occur in residual AML, but in already established, persisting LSCs, offering alternative explanations to progenitor reprogramming for AML recurrence. Future work will address to what degree the inflammatory response induced by chemotherapy is causally linked to enhanced stemness features, either through cell autonomous or paracrine crosstalk between miR-126[low] differentiated blasts and miR-126[high] LSCs, or whether it represents a

by-stander effect. Indeed, when comparing transcripts differentially expressed between chemotherapy- and un-treated cells specifically within the highly refined LSC cluster 6′ (Supplementary Data 2), we noted, among other stemness genes, induction of *MLLT3*, a recently described HSC self-renewal regulator[50]. Thus, chemotherapy may further potentiate stemness features in LSCs. Interestingly, *MLLT3* also acts as a master regulator of human erythroid and megakaryocyte fate decision acting through *GATA1* regulation[51], possibly explaining the accumulation of erythroid-like AML clusters upon chemotherapy. Longitudinal fate tracing studies will be required to formally prove the hierarchical relationships and plasticity between LSCs, early and late progenitors and LRC states, and their relative contribution to relapse.

The main scope of our study was to investigate the early effects of chemotherapy on AML populations. Nevertheless, we extended our scRNAseq analysis to a limited number of relapse samples. In line with published studies[7,52], relapse samples were characterized by a more shallow hierarchy and an enrichment in the proportion of LSCs. In contrast to the diagnosis samples, we found that relapsed *NPM1*[mut] AML and del(7) AML showed more active proliferation and metabolic rewiring, as evidenced by a uniform upregulation of oxidative-phosphorylation hallmarks among all disease clusters. This is consistent with a recent report that inhibition of the electron transport chain with mubritinib was effective in poor-outcome AML, but not chemotherapy-sensitive disease[53], highlighting a relapse- and, possibly, AML subtype-specific LSC vulnerability. Our data suggest that therapeutic targeting of oxidative phosphorylation, at least at the level of ETC complex I inhibition, may not be effective in *NPM1*[mut] AML patients with induction failure, where Ox-Phos hallmarks were downregulated in persisting LSCs.

The persistence of dormant LSC upon chemotherapy, as suggested by our study, implies that cytotoxic chemotherapy alone, which preferentially targets proliferating progenitors, is insufficient in curing these patients, also when sequential cycles of high-dose chemotherapy are administered to progressively recruit LSCs into cycle, rendering them potentially susceptible to chemotherapy. Indeed, a recent randomized controlled trial has shown feasibility and rapid achievement of CR in the sequential high-dose chemotherapy arm, with improvements in outcomes in AML subgroups for young patients with favorable risk[54]. Although, the benefit was least evident in intermediate-risk patients, particularly in patients with NPM1 mutations, where we hypothesize that the persistence of quiescent miR-126[high] LSCs, resilient to proliferative exhaustion even under the regenerative pressure imposed by chemotherapy or, even sustained by chemotherapy-induced senescence-like transcriptional states, may explain the lack of

benefit of intensified chemotherapy regimens. Combining targeted agents with activity in dormant LSC with chemotherapy directed against the proliferating leukemia bulk may improve response rates. Proof-of-concept therapeutic inhibition of miR-126 has recently been shown in PDX models of core-binding factor AML[55]. Moreover, the addition of the BCL2 inhibitor Venetoclax, thought to be active in LSCs, to high-dose chemotherapy has shown promising clinical results[56,57]. As such combinations often come with increased toxicity, a personalized approach with careful patient selection becomes instrumental. Drug sensitivity testing in relation to the cellular AML hierarchy composition has provided an initial framework for personalized selection of targeted agents[43].

We here provide further evidence of the complexity and importance of non-genetic drivers to the development of therapy resistance and proliferation in cancer. While the classical LSC model has been extensively characterized in steady-state and relapse AML, the role and persistence of LSC early after induction chemotherapy has been recently challenged[13–15]. Our work exploits high-resolution single-cell analysis to provide evidence for a "classical" LSC model in *NPM1*[mut] AML, where non-response/relapse was strongly correlated to a high proportion of quiescent miR-126[high] LSC already present at diagnosis. A weakness of our study is the relatively limited number of patients and the lack of longitudinal monitoring of single LSCs, as a direct demonstration that relapse unequivocally derives from a quiescent LSC persisting through induction chemotherapy. Nevertheless, our work provides a framework to stratify patients based on the presence of a miR-126[high] LSC component readily detectable in scRNAseq data and forms a starting point to identify therapeutic targets for their eradication. Prospective clinical implementation of targeted single-cell approaches to quantify miR-126[high] LSCs at diagnosis represents a promising next step to improve stratification of patients with intermediate-risk AML.

## Methods

Research presented within this study has been approved by an institutional review board (Comitato Etico Ospedale San Raffaele di Milano, protocol N°RPC, v2 22/01/2019). All animal procedures were performed according to protocols approved by the Animal Care and Use Committee of the San Raffaele Hospital (IACUC #807, #923, #1102) and communicated to the Italian Ministry of Health and local authorities according to the Italian law.

### miRNA reporter vectors
miR-126 sensor lentiviral vector constructs were designed and produced as previously described[22,58]. Briefly, miRNA target sequences for miR-126 were cloned into the 3' UTR of eGFP, downstream of a hPGK promoter in a bidirectional lentiviral vector expressing mCherry or dNGFR. miRNA activity can be assessed by flow cytometry analysis in live cells as the relative downregulation of GFP fluorescence compared to mCherry or dNGFR fluorescence. Third-generation LVs were produced by transient four-plasmid cotransfection into HEK-293T cells and concentrated by ultracentrifugation, as previously described[59]. LV titers, determined on HEK-293T cells by limiting dilution, ranged from $10^9$ to $10^{10}$ TU/ml with infectivity from $10^4$ to $10^5$ TU/ng of p24.

### Cell lines
HEK-293T cells were cultured in Iscove's modified Dulbecco's medium (Corning) supplemented with 10% heat-inactivated fetal bovine serum (FBS) (Euroclone), 100 IU/ml penicillin/streptomycin and 2% glutamine.

### Primary human AML samples
Primary AML patient samples were collected between 2005 and 2021 at the Ospedale San Raffaele Biobank (Milan, Italy). Informed consent to biobanking of peripheral blood and bone marrow cells had been

previously signed by all patients. The research proposal has been reviewed and approved by an institutional review board (Comitato Etico Ospedale San Raffaele di Milano, protocol N°RPC, v2 22/01/2019). Samples obtained for routine diagnostic or monitoring purposes were processed by density gradient centrifugation and mononuclear cells were frozen in liquid nitrogen until use.

### Cell culture and transduction
Bone marrow or peripheral blood mononuclear cells were thawed and cultured at a concentration of $5 \times 10^6$ cells/ml in X-VIVO 15 (Lonza BE04-418), 20% BIT9500 (StemCell Technologies #09500), SCF (Peprotech 100 ng/ml), FLT3L (Peprotech 100 ng/ml), TPO (Peprotech 50 ng/ml), IL-3 (Peprotech 10 ng/ml), IL-6 (Peprotech 10 ng/ml) and G-CSF (Peprotech 10 ng/ml), L-glutamine (Lonza 4 mM) and penicillin/streptomycin (100 UI/ml). Cells were plated at concentrations of $5 \times 10^6$ cells/ml for lentiviral transduction at a multiplicity of infection of 20. Cells were washed and transplanted by tail vein injection into NSG or NSGW41 mice within 24–36 h from thawing.

### Ex-vivo treatment of primary human AML blasts
Thawed primary human AML blasts were cultured on irradiated stromal feeder cell layers with cytokine- and SR1- (1uM) supplemented media. Live blast purification was achieved by magnetic bead-based dead cell removal prior to culture (Miltenyi # 130-090-101). Fresh media was added every 2–3 days and cells were re-plated prior to reaching 70–80% confluence. After 12 h of preconditioning, IACS-010759, diluted in DMSO, was added to a final concentration of 30 nM. Similarly, Cytarabine, diluted in PBS, was added to a final concentration of 100 ng/ml. Equivalent volumes of PBS and DMSO were added to the mock-treated conditions. Cells were harvested at Day 5 post-culture and analyzed by flow cytometry and transplanted into recipient-irradiated female NSG mice.

### RNA extraction and miRNA ddPCR
Total RNA was extracted with miRNeasy Micro Kit (Qiagen #217084) following manufacturer instructions. cDNA synthesis was performed using miRCURY LNA miRNA RT Kit (Qiagen #339340) following the company's guidelines for miRNA profiling. ddPCR was performed using EvaGreen supermix (Bio-Rad #1864034) and one of the following miRCURY LNA PCR primer sets (Qiagen): hsa-miR-126-3p (ID 204227), hsa- let-7a-5p (ID 205727), hsa-miR-16-5p (ID 205702), SNORD24 (ID 206999), SNORD48 (ID 203903), UniSP6 (ID 203956). cDNA input for miRNA ddPCR quantification ranged from 50 to 80 pg/well. Artificial RNA spike-in (UniSP6) was added to cDNA synthesis and assessed to monitor RT reaction efficacy. After droplet generation with Biorad Automated DG (#1864101), PCR was performed immediately with the following protocol: 95 °C for 5 min (ramp 2 °C/s), 40x cycles of 95 °C for 30 s and 56 °C for 1 min, three final steps at 4 °C for 5 min, 90 °C for 5 min, and a 4 °C indefinite hold. Data was acquired with Biorad QX200 Droplet Digital PCR System and analyzed with Biorad QuantaSoft Analysis Pro Software. miRNA expression was normalized by dividing the absolute concentration of miRNA by the geometric mean of four normalizers (miR-16, SNORD24, SNORD48, Let-7a). Each value is derived from the average of two replicate wells. The normalized value was multiplied by reaction volume (22 μl) and divided by the cDNA input to obtain a normalized concentration of miRNA (NV/ng cDNA).

### In vivo xenotransplantation experiments
NSG (NOD.Cg-*Prkdc*[scid] *Il2rg*[tm1Wjl]/SzJ) and NSGW41 (NOD.Cg-*Prkdc*[scid] *Il2rg*[tm1Wjl]/SzJ Kit[W41/W41]) female mice were purchased from Charles River Laboratories (Calco, IT) and held in specific pathogen-free conditions with a 12 hour dark-light cycle, standardized temperature (22 ± 2 °C) and humidity (55 ± 5%). All animal procedures were performed according to protocols approved by the Animal Care and Use

Committee of the San Raffaele Hospital (IACUC #807, #923, #1102) and communicated to the Italian Ministry of Health and local authorities according to the Italian law. Primary leukemic blasts were suspended in 200 μl PBS and transplanted by tail vein injection into 6-10 week old NSG or NSGW41 recipients. NSG mice were sub-lethally irradiated (150-200 rad) 12-24 hours prior to transplantation. Disease engraftment was monitored by periodic tail-vein or retro-orbital blood sampling. At sacrifice, 16 ± 2 weeks after transplantation if not specified otherwise, bone marrow cells were harvested by crushing femurs and tibiae under sterile conditions, filtered through a 40-μm cell strainer and resuspended in ice cold PBS + 2% FBS. Splenocytes were recovered by smashing whole spleens through a 40-μm cell strainer and resuspended in ice cold PBS + 2% FBS.

### Integration site analysis

Genomic DNA (gDNA) was extracted using the QIAamp DNA Micro Kit from Qiagen (ID: 56304), according to manufacturer's instructions and quantified with Qubit dsDNA HS Assay (Cat: Q32851) with Qubit Fluorometer (ThermoFisher Scientific). To retrieve lentiviral vector integration sites we adopted a sonication-based linker-mediated (LM) PCR method described previously[60,61]. Briefly, for each sample 20–900 ng of gDNA was fragmented by ultrasonication with Covaris E220 (Covaris Inc., Woburn MA) to obtain fragments with an average size of 1000 bp. The fragmented DNA was split into three technical replicates and subjected to end repair and A-tailing using the NEBNext Ultra Library Prep Kit for Illumina (New England Bio-Labs, Ipswich, MA), followed by ligation of a custom-made linker cassette using the NEBNext Ultra DNA Library Prep Kit for Illumina (New England BioLabs Ipswich, MA), as by the manufacturer's instructions. The custom-made linker cassettes harbor an 8 nucleotide-long barcode used for sample identification, a 12 random nucleotide sequence used for quantification purposes and sequences required for Read 2 Illumina paired-end sequencing. Ligated fragments were subjected to exponential amplification: The first PCR was performed using primers specific for the lentiviral vector LTR and the linker cassette with the following conditions: initial denaturation at 95 °C for 5′, denaturation 95 °C for 1′, annealing 60 °C for 45″, elongation 72 °C for 1′30″, for 35 cycles and final elongation for 10′ at 72 °C. The amplification products were then re-amplified with additional 10 PCR cycles using a primer specific for the linker cassette and primers complementary to the LTR containing an 8 nucleotide-long barcode, 12 random nucleotides and sequences compatible for Read 1 Illumina sequencing following the same PCR conditions as previously described. The resulting libraries were quantified by qRT-PCR using the KAPA Library Quantification Kit Illumina platforms (Kapa biosystems), according to the manufacturer's instructions. Libraries were sequenced using the Illumina HiSeq platform (Illumina, San Diego, CA). Processing of sequencing data, sequence quality analysis and IS Identification were performed by y-TRIS, a graph-based genome-free alignment tool based on the generation of consensus sequences, as described[62]. Datasets were pruned from potential contaminations and false positives between each primary mouse and from IS deriving from unrelated secondary mice. Each IS was used as marker to track an individual clone.

### In vivo chemotherapy experiments

For chemotherapy experiments, mice (NSGW41) were treated with 1.5 mg/kg Daunorubicin hydrochloride (Pfizer) by tail vein injection on days 1, 2, and 3. Cytarabine (Hospira) was administered at a dose of 60 mg/kg by subcutaneous injection on days 1 to 5 of the treatment regimen. Both drugs were prepared in saline and mice weight was checked daily for proper dosing. Control animals received equal dose of saline solution. Mice were allocated to treatment or control groups with equal AML burden based on a pre-treatment bone marrow aspirate evaluation of human AML chimerism. Litter mates were divided into both groups. For EdU (5-ethynyl-2′-deoxyuridine) incorporation assays, mice received daily intraperitoneal injections of EdU (ThermoFisher Scientific # E10187, 50 mg/kg body weight) from day 4 to day 7 of the in vivo treatment cycle.

### Limiting-dilution assays

Limiting-dilution assays (LDA) were performed as previously described[63]. Briefly, primary human AML cell were thawed or recovered from donor xenograft BM and FACS sorted for GFP[high] vs GFP[low] LDA experiments. Sorting cell yield was counted, washed twice in PBS, serially diluted and aliquoted in order to obtain the defined transplant cell dose per group for tail-vein injection in NSGW41 or sub-lethally irradiated NSG mice. Human engraftment was monitored with periodic blood sampling. Mice were euthanized at 16 ± 2 weeks post-transplant (primary endpoint) or in case of human engraftment in PB > 80%. Cells harvested from bone marrow or spleen at euthanasia were stained for hCD45, mCD45, hCD34, hCD38 and NGFR (for experiments with dNGFR-expressing miRNA reporters). Hierarchical gating was performed as follows: Singlets (FSC-H vs FSC-A), Physical parameters (FSC-A vs SSC-A), human cells (hCD45 vs mCD45), transduced cells (GFP vs mCherry or NGFR). Mice with human cell engraftment higher than 0.1% by flow cytometry analysis on total bone marrow or spleen cellularity (Physical gate) were scored as engrafters. LIC frequency estimation was calculated with the statmod R package (v 1.4.34) using the limdil function with a confidence of 0.95.

### Flow cytometry

Immunophenotypic analysis was performed using one of the following instruments: BD FACSCanto II, BD LSRFortessa II, BD FACSymphony A5, Beckman Coulter Cytoflex S or Beckman Coulter Cytoflex LX. Antibodies used for immunophenotypic analysis are listed in Supplementary Data 7. Cells were incubated with a human (Miltenyi Biotec, #130-059-901, dilution 2:100) and mouse FcR blocking reagent (BD #553141, dilution 1:100) prior to staining with antibody cocktail. All steps were performed at 4 °C in PBS 2% FBS. Data were analyzed with FCS Express 6 and 7 (DeNovo Software). Staining for cell cycle analysis was performed by fixing and permeabilizing cells with the eBioscience™ Foxp3 / Transcription Factor Staining Buffer Set as per manufacturer's indications, followed by 30 min incubation at +4ºC with human FcR blocking reagent (Miltenyi Biotec, #130-059-901, dilution 2:100) and overnight staining at +4ºC with Ki67-AF647 (BD #558615, dilution 2:100). Samples were acquired on the FACSymphony A5 with low flow rates after addition of Hoechst 33342. Staining for EdU incorporation was performed with the Click-iT™ Plus EdU Pacific Blue™ Flow Cytometry Assay Kit (TermoFisher Scientific #C10636) following the manufacturer's indications.

### Fluorescence-activated cell sorting

For GFP population sorting, freshly recovered or vitally frozen xenograft-derived bone marrow cells were thawed in RPMI 20% FBS, washed, pelleted and resuspended in MACS buffer (Miltenyi Biotec, 130-091-221) for cell count by trypan blue exclusion and staining. Dead cells were excluded with Annexin V staining. Sorting gates were defined based on the top and bottom 10–15% GFP/NGFR or GFP/mCherry fluorescence ratio on the human AML miR-126 sensor-transduced population (hCD45+/NGFR or mCherry+). Sorted cells were counted and diluted for transplantation. For single-cell RNA sequencing, leukemia was enriched by a patient-tailored combination of markers capturing as broadly as possible the entire leukemic population capturing either progenitors (CD34+ and/or CD117+) or myeloid cells (negatively selected as CD34-CD117-CD3-CD19-CD235a-) (Supplementary Fig. 1A). AML patient bone marrow mononuclear cells were thawed and processed as described above. After incubation with FcR blocking reagent, cells were stained either with sorting strategy 1 (CD45 APC-eFluor780, CD34 PE, CD38 PerCP-Cy5.5),

2 (CD45 PerCP-Cy5.5, CD34 APC, Annexin-V Pacific Blue, CD117 PE-Cy7, CD3 FITC, CD235a FITC, CD19 FITC) or 3 (CD45 PerCP-Cy5.5, CD34 APC, CD33 VioBlue, CD117 PE-Cy7, CD3 FITC, CD235a FITC, CD19 FITC, Annexin-V FITC) for 30 minutes on ice (Supplementary Fig. 1A). PT01 and PT06, timepoints DX and D14, were sorted with strategy 1, as the totality of leukemia cells expressed either CD34 or CD117. PT01 (D30), PT02 (DX, D30), PT07 (DX, D14), PT08 (DX, D14, D30, REL, REL_NR), PT09 (DX, D14), PT10 (DX, D14) were sorted with strategy 2, in order to exclude non-myeloid cells which made up a relevant proportion of the sample especially at post-therapy timepoints (B, T and GlyA⁺ erythroid cells). PT12 (DX, D14), PT13 (DX, D14, D30), PT15 (DX, REL), PT19 (REL), PT20 (REL), PT11 (DX, D30), PT17 (DX, D14, D30) and PT 18 (DX, D14, D30) were sorted with strategy 3. For strategies 2 and 3, the residual lineage negative CD34-CD117- fraction was included in scRNAseq analysis when the blast cluster did not fully express CD34 or CD117 in order to capture a more comprehensive picture of AML subpopulations. Cells were sorted with a BD FACS Aria Fusion SORP under sterile conditions at 4 °C in 1.5 ml DNA LoBind Eppendorf tubes or 5 ml polypropylene round-bottom tubes. Recovered cells were resuspended in known volume and counted with trypan blue for vitality when FACS cell recovery was >10,000 cells.

## RNA sequencing

Sequencing libraries were produced with SMART-Seq v4 Ultra Low Input RNA Kit (Takara Bio USA, Mountain View, CA, USA), as per the manufacturer's instructions. Briefly, 1 ng of input RNA was reverse transcribed with switch oligos for full-length cDNA conversion and amplification by PCR (8 cycles). Next-generation sequencing libraries were prepared using Nextera library prep kit (Illumina). Libraries were sequenced on Illumina platform with single-read SBS technology. Approximately 30 million 75-nucleotide-long reads per sample were generated. After quality check with fastQC, a preprocessing step including trimming and adapter removal using TrimGalore (v0.5.0) was performed. Trimmed reads were mapped to the GRCh38 reference genome assembly provided by the 10X reference data repository (refdata-cellranger-GRCh38-3.0.0) using STAR (v2.7.0d) setting the parameter outFilterMultimapNmax to the value of 1 in order to consider only uniquely mapped reads. The gene transfer file with exon and transcripts intervals provided by the 10X (refdata-cellranger-GRCh38-3.0.0.gtf) was used as the reference gene annotation file in order to ensure the best cross-compatibility with scRNAseq. Post-alignment metrics, including coverage distribution across gene length and percentage of reads mapping to exons were collected by using Qorts (v1.3.6). Reads were assigned to genes by using feature counts (v1.6.3), with the parameter minOverlap set to 10 and discarding chimeric reads.

## Differential gene expression on RNA sequencing data

Data preprocessing, exploration, and differential gene expression (DGE) analyses were performed with DESeq2 (v1.26.0) R packages. Explorative data analysis was carried out by Principal Component Analysis (PCA) on regularized log-transformed data. Two samples (PT01, mouse A3 GFP^low and mouse A1 GFP^low), with abnormal post-alignment metrics and behaving as outliers in PCA plots and TGR distribution, were discarded along with their paired GFP^high samples (PT01, mouse A1 and A3 GFP^high). DGE analysis was performed with the Wald test (p.adj Benjamini–Hochberg correction <0.05) by correcting for patient variability. We assessed the overall transcriptomic differences between GFP^low and GFP^high samples & the transcriptomic differences between treated and control samples. The comparison between GFP^low vs GFP^high samples produced 1957 DEGs; of these, 989 had log2FC > 0 (overexpressed in the GFP^low samples) and were included in the 126High signature; the remainder 968 genes with log2FC < 0 were included in the 126Low signature (Supplementary Data 3). DGE between treated and control conditions produced 3559

DEGs; 1768 upregulated and 1791 downregulated (Supplementary Data 3). DGE results from subset-specific comparison are reported in Supplementary Data 3. Genes with a corrected p-value (Benjamini–Hochberg correction) <0.05 were considered differentially expressed.

DGE results were inspected by performing Over Representation Analysis (ORA) and Gene Set Enrichment Analysis (GSEA). We queried several databases including Gene Ontology, KEGG, Reactome and Molecular Signatures Database (MSigDB). Both analyses were performed using clusterProfiler (v3.8.1) package. We implemented a data analysis workflow that allows to remove terms redundancy (applicable to GO terms) by pruning redundant terms according to semantic similarity with the simplify function from the GOSemSim package (v2.10.0). The algorithm iteratively decreases the cut-off of the simplify function until it reaches a number of terms less than 70 (for GO sets starting with less than 70 terms this step is omitted). Reduced GO terms are clustered based on the calculation of a kappa score between each pair of terms. For each pair an incidence matrix reporting the number of DEGs is constructed and a kappa score is calculated. The final output is a matrix of kappa scores reflecting the degree of similarity weighted for the different length of compared DEG vectors. A heatmap with hierarchical clustering on redundancy-reduced terms is then produced by using the pheatmap R package (Supplementary Fig. 8A, B).

## Single-cell RNA sequencing of longitudinal *NPM1*^mut AML patient samples

Sorted BM (PB for PT06 at diagnosis) mononuclear cell subsets were counted and pooled with conservation of their relative frequency, while loading at least 500 cells for the less abundant subpopulations. Cells were then resuspended at the appropriate concentration for loading into the Chromium 10X single-cell 3′ Gene Expression v2 or v3 chip according to the manufacturer's protocol. 3′ gene expression libraries construction and sequencing on Illumina platforms NextSeq or NovaSeq S1 were performed following the manufacturer's indications. Sequenced libraries were demultiplexed and processed by Cell Ranger Single-Cell Software Suite (version 3.1.0, 10X Genomics) using GRCh38 reference genome and gene annotations (v3.0.0) provided by the manufacturer. From primary patient samples, we retrieved 98,443 cells over 38 samples with a median of 54,856 mean reads per cell and 4129 Unique Molecular Identifier (UMI) median counts per cell. Of these, 42,398 cells were classified as blasts by our NPM1-MF algorithm.

Feature-barcodes filtered matrices from Cell Ranger were used as input for Seurat R package[64,65] (version 3.2.3). Seurat objects were merged in a single full dataset. Cells with a mitochondrial count ratio higher than 0.2 and <100 or >7000 expressed genes were removed from the dataset. UMI counts were log normalized and scaled for a factor of 10,000. The top 20% most variable genes were selected for downstream analysis. Cell cycle scores were assigned with the CellCycleScoring function using a reference gene lists[66]. We scaled data and regressed out unwanted variability by passing UMI count, percent of mitochondrial genes and cell cycle difference defined variables to the vars.to.regress argument. Cell cycle difference was defined as the difference between S phase and G/2 M phase module scores. Downstream analysis was performed on the top 100 principal components (PCA). In order to reduce patient-related and 10x chemistry version (v2 vs v3) bias, we performed data integration using the Harmony package (v1.0)[24]. Further analyses were performed on a subset of the full dataset, based on single-cell *NPM1* mutation detection (see below), retaining all cells labeled as AML. Briefly, UMAP dimensionality reduction[25] and clustering were performed with default parameters using the first 80 Harmony components. Modularity optimization was performed using classic implementation of the Louvain algorithm. Clustering resolution of 0.6 was used for downstream analysis. Markers identification was performed with FindAllMarkers function setting

a logFC threshold of 0.25 and a minimum percentage of cells expressing the gene greater than 25% in at least one of the compared clusters. Genes were considered markers of a specific cluster if their p-values returned by Wilcoxon Rank Sum test were $<1e10^{-5}$. Bulk RNA seq-derived gene signatures (126High and 126Low) were mapped on scRNAseq data by calculating the average expression level of each program at single-cell level, subtracted by the aggregated expression of a control feature set with the AddModuleScore function.

## Single-cell RNA sequencing of longitudinal del(7) AML patient samples

AML blasts were counted and resuspended at the appropriate concentration for loading into the Chromium 10X single-cell 3′ Gene Expression v3 chip according to the manufacturer's protocol. 3′ gene expression libraries construction and sequencing on Illumina NovaSeq S2 were performed following the manufacturer's indications. Sequenced libraries were demultiplexed and processed by Cell Ranger Single-Cell Software Suite (version 6.1.1, 10X Genomics) using GRCh38 reference genome and gene annotations (v3.0.0) provided by the manufacturer. We retrieved 44,170 cells over 8 samples with a median of 51,705 mean reads per cell and 9976 Unique Molecular Identifier (UMI) median counts per cell. We first ran a preliminary analysis using Seurat R package (v 3.2.3) on each single-patient dataset independently to discriminate AML cells (characterized by monosomy of chromosome 7 (Chr 7)) from their normal hematopoietic counterpart (non-AML). We leveraged the AddModuleScore function for evaluating the expression level of a Chr 7 signature by using as input gene list all genes located on it. The observed distribution of Chr 7 module scores in the datasets followed a bimodal distribution allowing us to classify cells as AML or non-AML by running a k-means clustering (n = 2) on the vector of Chr 7 signature scores and labelling cells in the high score group as non-AML and those in the low score group as AML (Supplementary Fig. 7D). We then subset the full dataset with AML-only cells. Single-cell data analysis was performed using the same workflow described for *NPM1*<sup>mut</sup> AMLs, except for harmony batch removal which accounted for both patient- and timepoint-dependent batch effects (orig.ident variable) and for dimensionality reduction computed on the top 30 Harmony corrected principal components.

## Single-cell RNA sequencing of treated and control AML xenografts

Freshly isolated BM mononuclear cells from control or chemotherapy-treated xenografts at day 8 from the start of treatment were sorted for isolation of bulk human AML cells or sorted miR-126 reporter GFP<sup>low</sup> blasts (Supplementary Fig. 4A). Cells were sorted with a BD FACS Aria Fusion SORP under sterile conditions at 4 °C in 1.5 ml DNA LoBind Eppendorf tubes or 5 ml polypropylene round-bottom tubes. Recovered cells were resuspended in known volume and counted with trypan blue for vitality when FACS cell recovery was >10,000 cells. Cells from sorted xenografts were then stained with TotalSeq-B Hashtag reagents (BioLegend) following the manufacturer's indications for pooling of different xenografts and sorted populations (Bulk or GFP<sup>low</sup> blasts) from the same AML patient prior to single-cell RNA sequencing chip loading. Pooled samples were then resuspended at the appropriate concentration for loading into the Chromium 10X single-cell 3′ Gene Expression and Feature Barcoding v3 chip according to the manufacturer's protocol. 3′ gene expression (GEX) and feature barcode (FBC) libraries were sequenced on Illumina platforms NovaSeq S1 or S2 following the manufacturer's indications. GEX and FBC libraries were demultiplexed and processed by Cell Ranger Single-Cell Software Suite (version 5.1.0, 10x Genomics) using GRCh38 reference genome and gene annotations (v3.0.0) provided by the manufacturer. We obtained 130,288 cells over 16 samples from 4 different donors with an average of 8143 cells for sample, 42,353 mean reads per cell, 2956 median genes per cell, 21,555 total expressed genes and 10,330 mean

UMI per cell (as detailed in Supplementary Data 4). scRNAseq gene expression data analysis was characterized by the following steps: preprocessing of single libraries, HTO demultiplexing to recover cell IDs belonging to single xenografts and sorted subpopulations, and analysis of the full dataset. Feature-barcodes filtered matrices from Cell Ranger (GEX and FBC) were used as input for Seurat. Each sample was pre-processed and cells with mitochondrial count ratios higher than 0.15 or a number of features greater than 7000 were removed. Next, for each sample we performed HTO demultiplexing in order to remove doublets and classify each cell with its hash.id for proper labeling of xenograft and population of origin. We leveraged the HTO demux pipeline as detailed in the Seurat vignette. We applied a CLR normalization using margin 2 for all samples except for three for which we applied margin 1. HTODemux function was run with positive.-quantile = 0.99 parameter. To overcome the presence of background noise and false positives, we setup a custom algorithm that allows to refine the results provided by HTODemux function. For each expressed HTO, it selects cells classified with a certain hash.ID, calculates a medoid from tSNE coordinates of such population and retrieves the range-limit coordinates (x and y) from the medoid according to 0.98 quantile of median Cartesian distances of cells to medoid. Doublets and singlets that do not fall within these boundaries are discarded, whereas singlets and or hashtag negative cells that lay within these boundaries are classified with the currently investigated hashtag ID (hash.ID). A tSNE plot with a box drawn with selected cells is produced as visual assessment of classification. We iteratively apply this procedure for each sample and hashtag. Metadata related to refined hash.ID for each sequenced cell was then added to the GEX Seurat object of all samples merged together. We then processed and analyzed the full merged GEX dataset in similar fashion to that described for the patient scRNAseq dataset. Briefly, UMI counts were log normalized and scaled for a factor of 10,000. Top 20% most variable genes were selected for downstream analysis and data was scaled and unwanted variability was regressed out by passing UMI count, percent of mitochondrial genes and cell cycle difference within the vars.to.regress argument. Top 70 PC from PCA analysis were used for dimensionality reduction and patient-derived batch effect was removed by running harmony integration using DonorID as batch variable. Clustering resolution of 0.6 was used for downstream analysis and resolution 1.2 for investigation of LSC subclusters. Markers identification was performed with FindAllMarkers function setting a logFC threshold of 0.25 and a minimum percentage of cells expressing the gene greater than 25% in at least one of the compared clusters.

## Cell type annotation

Cell type annotation was inferred by using the SingleR package[29] (v1.1.11), with several independent reference annotation datasets, including the Human Primary Cell Atlas[67], Blueprint[68], Encode[69] and published literature[28]. We performed a double annotation using for each dataset both main labels and refined labels. Cell labels were then imported in the Seurat object.

## *NPM1* mutation detection

Acquisition of the *NPM1* mutation is considered a transforming event converting pre-leukemia to an LSC[70], and this "trunk mutation" is maintained during clonal evolution making it a suitable MRD marker[71]. Due to its genomic location near the 3′ end of the gene, the *NPM1* mutation hotspot may be directly captured and sequenced in single-cell 3′ gene expression libraries[72]. We employed a purpose-developed strategy (*NPM1* Mutation Finder, NPM1-MF) to discriminate *NPM1* mutation status at the single-cell level in 3′ scRNAseq datasets on the assumption that the presence of a single mutant transcript is enough to define a cell as leukemic, while the opposite is not true for the classification of non-AML cells. The *NPM1* mutation A is defined as a TCTG tetranucleotide TCTG duplication at locus

chr5:171410540-171410543 (NM_002520.6(NPM1): c.860_863dup (p.Trp288fs)). In order to detect partial mutational patterns that occur at read boundaries, we searched for the following patterns: TCTCTGTCTGGC, GATCTCTGTCTGG$, GATCTCTGTCTG$, GATCTC TGTCT$, GATCTCTGTC$, ^CTCTGTCTGGCAG, ^TCTGTCTGGCAG, ^CTGTCTGGCAG, ^TGTCTGGCAG, ^GTCTGGCAG, where ^ denotes that the read begins with the next character, and $ that the read ends with the previous character. NPM1-MF considers the UMI counts supporting either *NPM1* mutant or WT allele to classify cells as MUT (≥1 UMI for the mutant allele); WT (>5 WT transcripts, no mutant transcripts), ND – not detected (cells with ≤ 5 WT transcripts and no mutant transcripts) or NoCall (cells without coverage over the NPM1 mutation region). Each patient was first analyzed separately. As we observed that NPM1 mutant cells did not co-cluster with wild type cells, we calculated for each cluster the relative abundance of NPM1-MF defined categories (MUT, WT, ND, NoCall) (Supplementary Fig. 7A–C, Supplementary Data 1). We extended the definition of AML to all cells belonging to clusters with the following sequential criteria: (1) MUT ≥ 40%, extend definition to whole cluster; (2) 20% ≤ MUT < 40% and WT < 5%, extend definition to whole cluster; (3) 10% ≤ MUT < 20% and WT < 5%, extend to whole cluster if cells exhibit myeloid lineage markers and the majority are NoCall; (4) 5% ≤ MUT < 10% and WT = 0%, extend to whole cluster if cells exhibit myeloid lineage markers and the majority are NoCall; (5) MUT < 5%, consider only MUT cells as AML. We then excluded all WT classified cells from further analysis. To complement this classification for clusters with high ND/NoCall cells, we investigated cell type composition by SingleR BluePrint + Encode correlations and confirmed myeloid/monocytic lineage for AML clusters and non-myeloid lineages for normal hematopoietic clusters (Supplementary Fig. 7A–C). By cluster-based assignment rules, the majority of cells could be unambiguously classified. Within a given cluster, cells in which mutant *NPM1* transcripts were detected were transcriptionally very similar to cells, in which insufficient read counts were obtained due to known limitations of the scRNAseq technology (Supplementary Fig. 1C), confirming accurate leukemia cell classification and reliable cluster-based AML annotation. A total of 42,398 cells from 10 patients were classified as AML.

### Pseudotime analysis

Pseudotime trajectory inference was performed with Monocle package[73] (version 3) as described in the official vignette. To maintain the previously calculated UMAP embedding from the Seurat analysis, we imported a cell dataset (CDS) containing PC and UMAP embedding annotations from the AML rds. Partitioning and clustering performed by monocle were skipped and associated data slots were replaced by setting the partition variable to 1 and cluster variable with the vector of clusters obtained from analysis in Seurat. After learning the graph by disabling partitions, we ordered the cells by leveraging the get_earliest_principal_node helper function to identify root start amongst *CD34*+ progenitor blasts (defined as *CD34* UMI ≥ 1).

### Over-representation analysis (ORA)

Cluster marker genes were investigated by ORA with the ClusterProfiler R package[74] on GO categories including biological process (BP), cellular components (CC) and molecular functions (MF). Briefly, we used the ego function, setting as query genes the cluster's markers and as gene universe the genes expressed in the dataset under analysis. GO terms with an adjusted p-value (Benjamini−Hochberg correction) <0.05 were considered significantly enriched. Query and universe genes sets were converted from gene symbols to ENTREZID with the bitr function in DOSE package (v3.10.2).

### Gene Set Enrichment Analysis (GSEA)

As post-chemotherapy timepoints exhibited high cell number variability between different patients and heterogeneous cell phenotype distribution, we decided to perform intra-cluster comparisons between timepoints for each eligible patient using the FindMarkers function, setting test.use = wilcox, a logFC threshold of 0 and return.thresh parameter equal to 1 in order to output all genes for which the test was performed. Only intra-cluster comparisons between groups with at least 5 cells were considered for differential gene expression analysis. Downstream analyses were carried out on the full output marker gene list, ranked by decreasing logFC with positive values indicating higher expression in post-chemotherapy timepoints. We performed Gene Set Enrichment Analysis using the GSEA function of ClusterProfiler R package[74] (v3.8.1) focusing only on the hallmarks gene set v7.0 (from MutSigDB) and GO genesets. GSEA results from all comparisons were combined into a single matrix of Normalized Enriched Scores (NES). A custom heatmap was produced by using ggplot2 R package (v3.2.1) ordering the terms (rows) by biological function similarity and comparisons (columns) by patient outcome category and median 126High signature expression value for that cluster. For the xenograft dataset, we performed GSEA analysis on clusters obtained at resolution 0.6. GSEA test was performed for each set of markers obtained from findallmarkers setup accordingly to retrieve all genes for which the test was performed (as stated above). Custom tile plots were produced from the GSEA results.

### Gene-signature correlations

To restrict the 126High signature to a lower number of informative genes, we identified which gene(s) provided a higher contribution to the overall signature module score value in scRNAseq data. As commonly done in reliability analysis, we performed an item-test correlation analysis, by computing the Spearman's rho correlation coefficient between each gene (scale data value) included in the 126High signature and the module score of the 126High signature across all *NPM1*^mut AML cells analyzed. A high correlation supports that the gene is a good "contributor" of the module score of the entire signature. The informative genes in the signature were defined as those with a positive non-negligible (rho > 0.3) significant (Bonferroni-corrected $p$ value < 0.05) Spearman correlation. Spearman's correlation coefficient was computed by using the corr.test function of the psych package (v 2.1.6) and Bonferroni correction for adjusting for multiple testing was applied. This analysis produced a reduced 24-gene signature (Supplementary Data 3), which was applied for survival analysis on published microarray and RNA sequencing data.

### Data preparation and survival analysis on published datasets

To test if the identified 24-gene signature could stratify patients with respect to overall survival, we used three public datasets including two AML microarray gene expression datasets (GSE12417, and GSE37642) and two AML RNA sequencing datasets TCGA[75] and OSHU cohorts[76]. ReadAffy from affy R package (v1.66.0) was used to create probe expression matrices from raw microarray data. Distinct matrices were produced accordingly to the microarray platform used. In particular, for both datasets, GPL570 and GPL96 were used. For each matrix, a standard preprocessing workflow including background correction and normalization was performed with rma function included in the affy package. We combined all matrices together using the probes common to both platforms. Full expression matrix was then batch-corrected (variables: combination of dataset and array) using the Combat function in the sva R package (v3.36.0). We then produced a gene expression matrix by calculating the median expression value of the probes associated with the same entrezID. In order to retrieve the highest number of valid genes names from the gene sets of interest provided as symbols, we converted them into entrezID using the convertSymbol function from the aliases2entrez R package (v0.1.1). RNAseq expression matrices from TCGA PanCan (RSEM batch normalized) and OSHU (Counts per million−CPM) datasets were used separately due to their different measurement units. The presence of

alternative genes aliases was manually curated. The evaluation of the initial 24-gene list was performed in two ways: first, we assessed whether the gene list was associated with overall survival (OS) within the three datasets independently (Supplementary Fig. 6D) and, second, we exploited the gene list to first derive and then validate two risk groups for OS (Fig. 6C). To assess the association of the gene list with OS, we estimated a Cox's proportional hazards model with lasso penalty (implemented in the glmnet R package) on the standardized intensity of the available genes, separately for each of the three datasets, which expressed a different number of genes from our signature: (i) 20 genes in the microarray dataset, (ii) 24 in the OSHU dataset, (iii) 19 in the TCGA. The lasso method (among the penalized regression methods) is widely used to overcome the multicollinearity issue caused by the usual correlations among genes belonging to the same signature, providing a further variable reduction. The penalty parameter lambda of the model was defined as the one minimizing the error obtained with a 10-fold cross-validation. The linear combination of the genes estimated from the model was then categorized with the median value obtained in the data defining two risk groups, with the purpose of showing the association of the linear combination with OS. The association of the two groups with OS was then assessed by estimating the corresponding Kaplan–Meier curves and testing their difference with the log-rank test and by estimating the hazard ratio with Cox's proportional hazard regression. For the definition and validation of risk groups defined from the initial 24-gene list, only genes present in both the training and testing datasets could be used. Therefore, for this assessment, we considered only the microarray and the OSHU datasets, which have 20 genes of the list in common (the genes common to all three datasets were only 15, which would have led to a higher loss of information). The microarray dataset was used as training set, due to the higher sample size. Due to the high heterogeneity of data among platforms and measurements, we used a training-testing procedure based on categorized expression data. In detail, for each patient, the expression data were categorized, by using their own quartiles: –1 (downregulated) if lower than the 1st quartile, 1 (upregulated) if greater than the 3rd quartile, 0 (normal expression) if between the 1st and 3rd quartile. In the training phase, the Cox proportional hazards regression with lasso penalty was applied on the categorized microarray data to derive a linear combination of genes, which was then categorized with the median for defining two risk groups. By using the same linear combination of the genes and cut-off, the two risk groups were then defined in the testing set. They were compared with respect to OS with the log-rank test and their hazard ratio was estimated by Cox's proportional hazard regression.

## Other statistical methods

Data were summarized as median ±interquartile (IQR) range, mean ± s.e.m., or mean ± SD depending on data distribution. Inferential techniques were applied in the presence of adequate sample sizes ($n \geq 5$), otherwise only descriptive statistics are reported. Two-sided tests were performed. For comparing numerical variables between paired samples, the Wilcoxon test for paired samples was applied and, in the case of multiple testing $p$-values were adjusted with Holm's correction. Since for $n = 5$, the minimum achievable two-sided $p$-value of the Wilcoxon test for paired samples is 0.0625, another nonparametric test based on bootstrap sampling was employed for conditions with such sample size (function boot.t.test in the MKinfer R package). For data in Fig. 3E, F and Supplementary Fig. 3E, F, comparisons between groups were performed by using linear mixed effects (LME) models to account for mice with the same donor and for replicate experiments. In each analysis, in order to meet the assumption of normality of the residuals of the model, when necessary, an appropriate transformation was applied to the response variable and, eventually, a few outliers were not included in the analysis. For testing differences between groups for each patient or testing overall differences between groups, a post-hoc analysis was performed by using the R package phia and, in the former case, by also applying Holm's correction in order to account for multiple testing. Figure 3E: The response variable y (absolute BM AML cells) was used with the square root transformation—one outlier was removed from the analysis: Patient PT03-Replicate B-Mouse B3- group Control. Figure 3F: The response variable y (percent $CD34^{+}38^{-}$ AML blasts) was used with the cube root transformation. Two outliers were removed from the analysis: Patient PT03-Replicate A-Mouse A3-group Treated and Patient PT16-Replicate A-Mouse B1-group Treated. Supplementary Fig. 3F: The response variable y (AML TGR) was used with the ordered quantile normalization transformation. One outlier was removed from the analysis: Patient PT15-Replicate A-Mouse C1-group Control.

## Reporting summary

Further information on research design is available in the Nature Portfolio Reporting Summary linked to this article.

## Data availability

RNA sequencing data generated in this study have been deposited in the GEO database under accession code GSE185993 under public access. In addition, all processed single-cell RNA sequencing data can be accessed and queried through our online, interactive user interface [http://www.bioinfotiget.it/mnaldini_natcomm2023/]. All data accessed from external sources and prior publications have been referenced in the text, GSE12417, GSE37642. Source data are provided with this paper.

## Code availability

Code used for the generation of results reported in this manuscript is available at the following GitLab repository [http://www.bioinfotiget.it/gitlab/custom/mnaldini_natcomm2023].

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

## Acknowledgements

The authors thank members from the Gentner lab for help with experiments, discussion and insight; Fractal facility personnel for cell sorting; Dejan Lazarevic and the center for Omics sciences (COSR) for advice and assistance with genomic sequencing; Cristina Tresoldi and the OSR biobank for sample collection and storage; Raffaella Milani and the Hematology department for fruitful discussions and Luigi Naldini for critically reviewing the manuscript. R.D. M. is a New York Stem Cell Foundation—Robertson Investigator. This research was supported by grants to B.G. from the Italian Association for Cancer Research (AIRC-IG 2018 Id.22143), a John Goldman Clinical Research Fellowship from the European Hematology Association (EHA 2014) and from the Telethon Foundation (TIGET 2016 core grant no. C1).

## Author contributions

M.M.N. and G.C. planned and performed experiments, analyzed and interpreted the data, performed sequencing data analysis; M.Ba. developed and performed the bioinformatics analysis; P.M.V.R. supervised statistical analysis and performed LME model and survival analysis; E.Z., G.D., C.C., and D.G. helped with experiments; F.P. and L.V. helped with patient sample selection and analysis; A.C. and M.V. performed and analyzed integration site analysis; I.M, E.M., R.D.M., and C.D.S. provided intellectual input and supervision; L.V., M.G.C., M.Be., and F.C. supervised patient selection and analysis and provided clinical and intellectual input; B.G. designed and coordinated the research, interpreted the data, supervised research; M.M.N and B.G. wrote the manuscript; M.M.N., G.C., M.Ba., P.M.V.R., M.G.C., L.V., and B.G. revised the manuscript.

## Competing interests

The authors declare no competing interests.
