## [Peer Review File · Nature Communications]

Longitudinal single-cell profiling of chemotherapy response in acute myeloid leukemiaREVIEWER COMMENTS

Reviewer #1 (Remarks to the Author):

In Naldini et al., the authors use single-cell RNA-sequencing to study the role of leukemia stem cells in NPM1-mutated AML patients and xenografts. They were able to apply a new method, termed NMF, to identify blasts in the single-cell samples, which enabled the investigation of response patterns in blasts and leukemia stem cells following chemotherapy. Overall, this paper is clear in its description of experimental details and results, well designed, and sufficiently statistically supported. Addressing the minor comments on the bioinformatics below would strengthen the manuscript and make it a strong candidate for publication.

1. The pipeline for detecting and counting mutant NPM1 transcripts should not be called NMF, as non-negative matrix factorization is a widely adopted method for cluster definition in single-cell genomics and is abbreviated as NMF. Changing this will likely minimize confusion for the readers. Further, this pipeline should work for all genes where the mutation hotspot is near the 3' end of the gene. Generalizing the method (and its name) could be helpful for others in the community.
2. For scRNAseq of the AML xenografts it is explained in detail in the methods section how HTO dextramultiplexing was used to remove doublets from the data. Since this same technology wasn't applied to the longitudinal AML patient samples, how did you ensure that doublets weren't skewing the data in those evaluations? Similarly, for the longitudinal AML patient samples the effects of cell cycle were regressed out of the data, but it doesn't look like this was also applied to the xenografts. My larger concern is that either 1) different single-cell pipelines and parameters were used for different portions of the paper or that 2) some details are missing from the methods (assuming the same pipeline and parameters were used for both experiments). Could the authors please comment on this discrepancy in bioinformatics application.
3. The paper would benefit from highlighting if and how the use of peripheral blood for PT06 affects the results compared to other bone marrow samples.

Reviewer #2 (Remarks to the Author):

Naldini et al. have performed extensive scRNAseq analysis of AML patients and xenografts before and after chemotherapy treatment. They describe enrichment of inflammatory signaling pathways and senescence pathways following chemotherapy treatment. They additionally use a reporter system which allows enrichment for functional leukemia initiating cells, in order to develop a gene expression signature that has prognostic value in larger patient datasets. Overall, the authors conclude that their study demonstrates the persistence of dormant LSCs as the mechanism of therapy failure in AML. However, no mechanistic experiments were performed, nor were traditional cell cycle assays. Functional transplantation assays were performed, but do not necessarily support the authors' conclusions.

Main concerns:

1. LSC dormancy: Throughout the manuscript (including the title), there is liberal use of conclusive language to describe LSC quiescence, dormancy, and hibernation; however, no formal cell cycle assays have been performed at any point.
 - Several times, the authors use scRNAseq expression of just a pair of genes (CDK6 and INKA1) to label a cluster of cells as dormant (Supplementary Figure 6C) or quiescent (Figure 6D).
 - In other areas, GSEA is used as evidence of quiescence (Figures 2, 5, and 6), however the authors are not consistent with their interpretation of GSEA results. In Figure 5G, cell cycle gene lists are equally downregulated in Cluster 4, which the authors have designated the "cycling" population. Does this mean the cycling population becomes quiescent after chemotherapy?
 - The authors also apply Seurat's CellCycleScoring function (Figures 6D and E), but this only infers G1, G2M, or S phases, but not G0 which truly represents a quiescent state.
 - While both of these latter methods can be used to claim differences in overall cell cycle status between two clusters, neither is sufficient to conclude quiescence or stability of a quiescent state. Flow cytometry analysis is necessary to discriminate G1 from G0.

- Finally, loss of quiescence following chemotherapy has been shown to be extremely brief in AML cells, and likely would not be detected 3 days post-treatment when the authors have performed their analysis.

2. Stability of LSC populations before vs. after chemotherapy: In Figure 3H, the authors show limiting dilution transplantation of human AML cells after high dose chemotherapy treatment vs. non-treated controls. One patient sample has a significant depletion of functional leukemia initiating cells (LICs), whereas the remaining 4 patient samples have similar LIC frequencies in treated vs. control conditions (based on the adjusted P values shown). The authors actually make claims that are not supported by the statistics they show. 2 of the patients have slightly higher LIC frequencies but don't reach statistical significance. This raises concerns of a desire to draw a specific interpretation vs. objective assessment of the data.

This functional result is consistent with their miR-126 reporter, which suggests a reduction of prospective LSC frequencies in 2/5 of the patients, whereas prospective LSC frequencies are unchanged for the remaining 3 patients (Supplementary Figure 3F).

- These results are nearly identical to the findings of Farge et al. (2017) where 3/4 patient samples showed no change in LIC frequencies after chemotherapy treatment, and the fourth patient sample showed a significant reduction in LICs. However, the authors have interpreted their findings completely differently from Farge et al. The authors need to explain what additional evidence leads them to this contradictory interpretation of essentially the same data.

- Based on their arguments, the authors appear to fundamentally misunderstand the controversy around the persistence of LICs/LSCs. Of course, if any functional LSCs can be detected following chemotherapy treatment, it can be argued that LSCs have persisted. However, if LSCs are protected from chemotherapy due to their quiescent features, they should be preferentially spared following chemotherapy treatment (i.e., found in higher proportions than pre-treatment). This is not the case in the current manuscript, nor in recent published reports from multiple groups. This suggests that LSCs are just as sensitive to chemotherapy as any other leukemic cell and therefore, the more relevant therapeutic limitation is the inability to escalate chemotherapy dosage without killing healthy cells. In contrast, it would not be productive to focus on chemo-sensitization of LSCs by targeting their quiescent properties. This is the essence of the insights raised by these recent studies, and the authors' data reinforces these same conclusions.

3. 126High signature: The survival analysis shown in Figure 6C is an impressive application of the 126High signature, however it is not clear what practical value this signature adds over the existing LSC-17 signature that has been previously described. The expression patterns of the 126High signature (Figure 6A) look nearly identical to that of the LSC17 signature (Supplementary Figure 6B). Furthermore, one of the same microarray datasets was analyzed in Ng. et al., 2016 (GSE37642) and already showed that patient survival can be predicted by primitive LSC signatures at diagnosis. Therefore, it is not surprising that a similar gene set would also predict patient survival.

4. Survival analysis: It is unclear why different gene lists were applied in Violin plots in Figure 6B (989 genes?) vs. survival analysis in Figure 6C (24 genes) vs. survival analyses in Supplementary Figure 6C (10 genes vs. 12 genes, where only 6 genes are shared).

- First, the rationale to reduce gene list size to 24 genes by correlation with a module score derived from the same gene list is unconventional and somewhat circular in logic.

- Second, the application of Lasso penalty in subsequent survival analyses is inappropriate. This is a method for optimal model selection in a training dataset, which should then be validated in a testing set (as was done in Ng. et al., Nature, 2016). Here, the authors have independently applied Lasso penalty across 3 different data sets, which effectively selects the subset of genes for each dataset that are most biased towards a significant result.

- If the 126High signature were to be applied prospectively to new patients, which is the robust gene list that should be used? 24 genes? 10? 12? In the prospective setting, it is not possible to refine a gene list based on the patient outcome, as the outcome remains unknown. If a gene list cannot be applied prospectively, it has little practical value.

5. Human patient scRNA seq analysis: The "patient-tailored" FACS processing of human AML patient cells prior to scRNAseq analysis is concerning. If the authors desired to remove non-

leukemic lymphoid and erythroid cells, they should simply stain for lymphoid and erythroid markers and apply an exclusion gate similarly across all patient samples, in order to preserve the composition of the myeloid populations.

- It is unclear why samples were instead fractionated into multiple different subpopulations and then re-combined. Even if an effort was made to preserve their original proportions, this would be difficult to accomplish accurately without introducing bias.
- Furthermore, different sorting strategies were used for different patients, and even samples from the same patient were subjected to different sorting gate strategies at separate time points (PT01). Therefore, it is unclear how samples can be directly compared to each other; e.g., it is inappropriate to describe a near extinction of myelo-/monoblastic cells after chemotherapy treatment if it is possible that the proportions of these cells could have been inadvertently biased by sorting gate or cell pooling steps.
- This is further complicated by the cluster-based assignment of NPM1 mutational status, where only less than half of cells classified as "AML" at diagnosis actually have direct evidence of NPM1 mutant transcripts.

6. Controversy in the field: In the Discussion section, the authors claim to provide a missing link that reconciles classical LSC models of therapy resistance vs. more recent challenges to the concept of preferential persistence of LSCs. It is not clear how any of their data re-contextualizes these recent studies that were based on functional analysis of AML, particularly given that the authors' study concentrates on molecular descriptions rather than functional biology. The only functional biology that they do show (Figure 3G) demonstrates that functional leukemia-initiating cells either significantly decrease or are unchanged following high dose chemotherapy treatment. This is inconsistent with classical notions that LSCs would be preferentially spared by chemotherapy treatment. However, the authors emphasize classical ideas of LSC persistence and dormancy throughout the written text.

Rather than resolving biological processes, the strong conclusions in the absence of clear data is adding to confusion.

Additional points:

1. In Main Figure 4B, the same gene list is shown as both an upregulated and downregulated gene set when comparing GFP low vs GFP high populations (JAATINEN_HEMATOPOIETIC_STEM_CELL_UP).

2. In Figure 6G-I and Supplementary Figure 6D-E, the LSC sub-clustering is not convincing. It is evident from Supplementary Figure 6E that there is little distinction between the subclusters, despite that this heatmap is showing the top 10 marker genes for each subcluster. It is not clear how these panels add to the story.

3. The terminology related to the reporter system is very difficult to follow because it is inconsistent throughout the figures, including the direction of the axes and how this relates to miR-126 activity. In Figure 3C the x axis is labeled "1/TransGene Ratio" then the axis is reversed in Figure 3G ("TransGene Ratio") and Figure S3 ("TGR"), which then reverts back in Figure 5F ("126High signature"). This is counter-intuitive and would be easier to keep the axes consistent, such that higher values always indicate higher miR-126 activity.

4. In Figures 3B and 3C, the authors have excluded PT15 from their analysis. Why? Based on Figure 3H, this patient sample appears to have the highest baseline LIC activity, suggesting it could have presented a greater challenge to define clear LIC- vs LIC+ populations. This patient is also completely omitted from molecular analyses in Figure 4, which is unfortunate given the unique functional biology associated with this patient in Figure 3. It is important to capture as much diversity as possible in molecular studies of AML.

5. A comparison of Figure 4D vs. Figure S4E suggests that miR-126high vs miR-126low AML subsets have nearly identical gene expression responses to chemotherapy treatment. Furthermore, in Supplementary Figures 4C and 4D, the authors show no change in LSC17 score

before vs. after chemotherapy in either purified population. This is inconsistent with claims of enhanced features of stemness among miR-126high populations in Figure 5.

RESPONSE TO REVIEWERS' COMMENTS

Reviewer #1 (Remarks to the Author):

In Naldini et al., the authors use single-cell RNA-sequencing to study the role of leukemia stem cells in NPM1-mutated AML patients and xenografts. They were able to apply a new method, termed NMF, to identify blasts in the single-cell samples, which enabled the investigation of response patterns in blasts and leukemia stem cells following chemotherapy. Overall, this paper is clear in its description of experimental details and results, well designed, and sufficiently statistically supported. Addressing the minor comments on the bioinformatics below would strengthen the manuscript and make it a strong candidate for publication.

We thank this reviewer for his/her positive assessment of our work and for providing constructive input to improve the manuscript.

1. The pipeline for detecting and counting mutant NPM1 transcripts should not be called NMF, as non-negative matrix factorization is a widely adopted method for cluster definition in single-cell genomics and is abbreviated as NMF. Changing this will likely minimize confusion for the readers. Further, this pipeline should work for all genes where the mutation hotspot is near the 3' end of the gene. Generalizing the method (and its name) could be helpful for others in the community.

We thank the reviewer for this observation and have changed the acronym accordingly to NPM1-MF. The generalization of such a procedure is theoretically possible and could be hypothesized for genes with mutation hotspots near the 3' end of the gene such as DNMT3A or GATA2 in the setting of AML. In our case, such mutations are either not present within the selected patient cohort (GATA2), or do not represent a leukemia-initiating event (DNMT3A), and as such would not aid in the discrimination between AML and normal hematopoietic progenitors, the latter potentially bearing pre-leukemic DNMT3A mutations.

We have now added longitudinal scRNAseq data from 3 patients with a different molecular AML subgroup, namely chromosome 7 deletion. In this case, we computed the expression of the module score of genes located on chromosome 7 in single cells to distinguish leukemia from normal hematopoietic cells, thus extending the scope of our work. This second approach is generally applicable to AML with chromosomal gains or losses, which constitute, in addition to NPM1 mutations, an important and clinically relevant patient subgroup.

Three additional del(7) AML patients, who were refractory to induction chemotherapy (PT11, PT17) or experienced early relapse (PT18), respectively, were included (Fig. 1A). To unambiguously distinguish malignant from non-leukemic cells, a particularly critical aspect in post-chemotherapy samples where LSC may hide within regenerating hematopoiesis by sharing surface markers with non-malignant HSC, we determined the NPM1 mutational status or expression of the module score of genes located on chromosome 7 in single cells, respectively (see Methods). Both alterations unambiguously mark leukemia cells and gave us confidence in excluding bias from HSC contamination. By cluster-based assignment rules, the majority of cells could be unambiguously classified. Within a given cluster, cells in which mutant NPM1 transcripts were detected were transcriptionally very similar to cells, in which insufficient read counts were obtained due to known limitations of the scRNAseq technology

(Supplementary Fig. 1C), confirming accurate leukemia cell classification and reliable cluster-based AML annotation.Likewise, 53,513 del(7) AML cells were projected on a 2D UMAP landscape, distinguishing a prominent early progenitor cell compartment (cl. 0, 1, 5, 9), lymphoid-like clusters (cl. 2, 7), erythroid-like clusters (cl. 3, 8), a cycling compartment (cl. 6), differentiated myeloid-like cells (cl. 4) and a microcluster of NK-like cells (cl. 11) (Fig. 1E, Supplementary Fig. 1G and Supplementary Table 2). In the del(7) cohort, we obtained 8,970 cells at day 14 and 3,930 cells at day 30, from 2 patients each. Differently to the NPM1^{mut} cases, del(7) AML accumulated high numbers of differentiated myeloid cells on day 14, which disappeared on day 30 (Fig. 2D,E). This may be compatible with therapy-induced differentiation along a primitive to mature axis and exhaustion of mature blasts. At the same time, progenitor cells persisted, with accumulation of clusters 10, 1 and 5 by day 30. GSEA highlighted a prominent proliferative response on day 14 with upregulation of MYC, E2F targets, cell cycle checkpoints and OxPhos across all progenitor clusters, followed by a depletion of these signatures on day 30 (Fig. 2F). Notably, blasts from del(7) patients showed a persistently elevated inflammatory expression profile (Fig. 2F).

Taken together, longitudinal post-chemotherapy assessment highlighted divergent (for NPM1^{mut} AML) or time-dependent (for del(7) AML) transcriptional responses in AML cell subpopulations characterized by (1) early proliferation within a regenerating, Ox-Phos^{high} cell compartment and (2) cell cycle quiescence coupled to inflammatory hallmarks within the early progenitor cell compartment. Detection of an Ox-Phos^{low} early progenitor cell subpopulation on day 14 in the NPM1^{mut} cases may argue for a pre-existing, discrete subpopulation of LSC persisting through chemotherapy in this AML subgroup, while the time-dependent changes observed in del(7) AML may hint to broader plasticity within the progenitor cell compartment.

2. For scRNAseq of the AML xenografts it is explained in detail in the methods section how HTO dextramultiplexing was used to remove doublets from the data. Since this same technology wasn't applied to the longitudinal AML patient samples, how did you ensure that doublets weren't skewing the data in those evaluations? Similarly, for the longitudinal AML patient samples the effects of cell cycle were regressed out of the data, but it doesn't look like this was also applied to the xenografts. My larger concern is that either 1) different single-cell pipelines and parameters were used for different portions of the paper or that 2) some details are missing from the methods (assuming the same pipeline and parameters were used for both experiments). Could the authors please comment on this discrepancy in bioinformatics application.

We have now better clarified the bioinformatics analysis pipeline that was employed in the different data sets in the revised version of our paper.

To generate the PDX-derived single cell dataset, we combined AML blasts recovered from different mice of the same experimental group within a single well of a chromium 10x chip. Indeed, emergence of the CiteSeq approach with Cell Hashing (Stoeckius et al, Nat Methods 2017 [PMID:28759029] and Genome Biol 2018 [PMID: 30567574]) allowed "super-loading" of the well and computational removal of doublets by filtering out cells containing multiple Hash-tags. This approach allowed us to increase the total number of sequenced cells, while maintaining the cost of consumables in an acceptable range. Furthermore, by tagging cells derived from each PDX with a different hashtag, we were also able to maintain such distinction during the analysis, allowing us to confirm the consistency of results across replicates.

Differently to the PDX model, we performed conservative loading (with a predicted doublet rate generally < 5%) of the AML patient derived cells, keeping each single timepoint in a separate 10x well, also due to logistical considerations. Indeed, the number of available cells, especially at timepoints obtained after chemotherapy, was well below the recommended loading limits. Moreover, QC filtering (mitochondrial count ratio higher than 0.2 and/or cells with <100 or >7,000 expressed genes) allowed removal/depletion of non-viable cells and doublets.

Of note, no direct comparison between the PDX and patient sample scRNAseq datasets were performed, but their independent analysis came to similar conclusion. This per se is a strong indication for the validity of our approach. To maintain a consistent data analysis approach across the manuscript, the same bioinformatic pipeline was applied to all single cell RNA sequencing datasets. Depending on number of recovered cells and expressed genes, the number of calculated principal components (PC) used for dimensionality reduction was adapted. Both scRNAseq datasets were partially corrected for the effect of cell cycle by accounting for differences between G1 and G2M/S. We have better specified this point in the appropriate methods of the revised manuscript. The text within the methods section concerning the processing of the PDX single cell dataset now briefly reports all the same steps that were included in the analysis applied to the NPM1^{mut} and del(7) AML patients dataset. The text now reads:

We then processed and analyzed the full merged GEX dataset in similar fashion to that described for the patient scRNAseq dataset. Briefly, UMI counts were log normalized and scaled for a factor of 10,000. Top 20% most variable genes were selected for downstream analysis and data was scaled and unwanted variability was regressed out by passing UMI count, percent of mitochondrial genes and cell cycle difference within the vars.to.regress argument. Top 70 PC from PCA analysis were used for dimensionality reduction and patient-derived batch effect...

3. The paper would benefit from highlighting if and how the use of peripheral blood for PT06 affects the results compared to other bone marrow samples.

We agree with the reviewer that the use of the same cell source throughout the manuscript would have been preferable. Unfortunately, no BM samples were available for the analysis of PT06 at diagnosis. We feel, though, that this “outlier” has no impact on the interpretation of our data, for the following reasons: Both blood and bone marrow blasts can be routinely used for the diagnosis and subtyping of AML, as studies assessing morphologic features, cytochemistry, cytogenetics or immunophenotype of AML blasts, have found no differences between patient-matched BM or PB blasts, with exception of their abundance, which may be higher in the BM (PMID: 10361507, 12942032). Furthermore, both sources have been employed in preclinical settings with successful identification of leukemia initiating cells and engraftment of PDX models (PMID: 7509044). Thus, when addressing the heterogeneity of AML blasts at diagnosis, peripheral blood may be a valid and easily accessible source of blasts faithfully recapitulating marrow leukemia.

That said, there may be some differences, especially when assessing chemotherapy response. The bone marrow niche has been extensively studied in relation to chemoresistance where it plays a pivotal role in supporting and inducing escape mechanisms to therapy in leukemic

blasts. Missing pro-survival signals from the niche in blood-derived AML cells may alter the recovery of AML subpopulations. This was, however, not relevant in our study, as BM was the only used source for post-treatment timepoints. Furthermore, PT06 did not anyway significantly contribute to the post-treatment AML dataset, as PT06 belonged to the patients that went into sustained complete remission.

We have now clarified in the text that only the diagnosis sample of PT06 was taken from peripheral blood, due to sample availability:

For PT06, peripheral blood instead of BM was available at diagnosis.

Reviewer #2 (Remarks to the Author):

Naldini et al. have performed extensive scRNAseq analysis of AML patients and xenografts before and after chemotherapy treatment. They describe enrichment of inflammatory signaling pathways and senescence pathways following chemotherapy treatment. They additionally use a reporter system which allows enrichment for functional leukemia initiating cells, in order to develop a gene expression signature that has prognostic value in larger patient datasets. Overall, the authors conclude that their study demonstrates the persistence of dormant LSCs as the mechanism of therapy failure in AML. However, no mechanistic experiments were performed, nor were traditional cell cycle assays. Functional transplantation assays were performed, but do not necessarily support the authors' conclusions.

We thank the reviewer for taking his/her time to critically examine our manuscript. We have now performed additional experiments and added new data supporting our findings that the persistence of quiescent LSCs represents a mechanism of treatment failure in a relevant subgroup of AML patients, namely in NPM1-mutated AML. Showing parallel findings in xenograft models (with functional LSC validation by gold-standard assays) and patients using single cell readouts represents a unique strength of our manuscript, which we hope this Reviewer will acknowledge. Being a heterogeneous disease, it is entirely possible that additional mechanisms, e.g. metabolic adaptation, may contribute to AML therapy failure, which is not necessarily in conflict with LSC-based concepts. We now include mechanistic experiments (electron transport chain complex inhibition) and cell cycle analysis (as detailed below) to consolidate our claim that a quiescent, OxPhos^{low} LSC subset drives AML persistence in NPM1^{mut} AML. The manuscript has been revised as follows:

If these OxPhos^{low} LSC were at the origin of relapse, we hypothesized that our diseases would be resistant to pharmacological OxPhos inhibition. Hence, we treated miR-126 reporter⁺ AML cells from PT16 in vitro with IACS-010759 (IACS), a selective mitochondrial electron transport chain complex I-inhibitor, alone or in combination with AraC chemotherapy. IACS-treated cells had higher miR-126 activity (Supplementary Fig5E), consistent with selection of OxPhos^{low} LSCs. Strikingly, as compared to the mock-treated group, IACS-treated cells showed strongly enhanced repopulation activity when transplanted into NSG mice, underlining the functional relevance of these miR-126^{high} OxPhos^{low} LSCs, at least in NPM1^{mut} AML (Fig.5N).

Moreover, we performed longitudinal scRNAseq on an additional AML subtype, namely AML with del(7), sampled directly from patients before and after chemotherapy. These del(7) diseases showed a prominent proliferative response with induction of OxPhos pathway genes, and no clear evidence for a quiescent LSC subset persisting through induction chemotherapy. Still, both NPM1^{mut} and del(7) AML were characterized by miR-126^{high} cells expressing LSC signatures, which increased during chemotherapy. That may indicate that cell cycle control is differently wired in LSC from different AML subgroups, an intriguing hypothesis that merits further investigation in follow up studies, but is beyond the scope of this paper. The manuscript has been revised as follows:

Three additional del(7) AML patients, who were refractory to induction chemotherapy (PT11, PT17) or experienced early relapse (PT18), respectively, were included (Fig.1A)..... Likewise, 53,513 del(7) AML cells were projected on a 2D UMAP landscape, distinguishing a prominent immature progenitor cell compartment (cl. 0, 1, 5, 9), lymphoid-like clusters (cl. 2, 7), erythroid-like clusters (cl. 3, 8), a cycling compartment (cl. 6), differentiated myeloid-like cells (cl. 4) and a microcluster of NK-like cells (Fig. 1E and Supplementary Fig. 1G)..... In the del(7) cohort, we obtained 8,970 cells at day 14 and 3,930 cells at day 30, from 2 patients each. Differently to the NPM1^{mut} cases, del(7) AML accumulated high numbers of differentiated myeloid cells on day 14, which disappeared on day 30 (Fig. 2D,E). This may be compatible with therapy-induced differentiation along a primitive to mature axis. At the same time, immature-like cells persisted, with accumulation of clusters 1 and 5 by day 30. GSEA highlighted a copious proliferative response on day 14 with upregulation of MYC, E2F targets, cell cycle checkpoints and OxPhos across all immature clusters, followed by a depletion of these signatures on day 30 (Fig. 2F). Notably, blasts from del(7) patients showed a persistently elevated inflammatory expression profile (Fig. 2F)..... Similarly, within the immature-like clusters 1 and 5 enriched after chemotherapy in the del(7) cohort (see Fig.2E), cluster 1 had highest expression of the 126High signature and the PDX LSC markers from subcluster 6' (Fig. 6F). These data indicate that a 126High LSC subset is selected during induction chemotherapy..... Detection of an Ox-Phos^{low} immature cell subpopulation on day 14 in the NPM1^{mut} cases may argue for a pre-existing, discrete subpopulation of LSC in this AML subgroup, while the time-dependent changes observed in del(7) AML may hint to plasticity within the immature cell compartment.

Main concerns:

1. LSC dormancy: Throughout the manuscript (including the title), there is liberal use of conclusive language to describe LSC quiescence, dormancy, and hibernation; however, no formal cell cycle assays have been performed at any point.
 - Several times, the authors use scRNAseq expression of just a pair of genes (CDK6 and INKA1) to label a cluster of cells as dormant (Supplementary Figure 6C) or quiescent (Figure 6D).
 - In other areas, GSEA is used as evidence of quiescence (Figures 2, 5, and 6), however the authors are not consistent with their interpretation of GSEA results. In Figure 5G, cell cycle gene lists are equally downregulated in Cluster 4, which the authors have designated the “cycling” population. Does this mean the cycling population becomes quiescent after chemotherapy?
 - The authors also apply Seurat’s CellCycleScoring function (Figures 6D and E), but this only

infers G1, G2M, or S phases, but not G0 which truly represents a quiescent state.

- While both of these latter methods can be used to claim differences in overall cell cycle status between two clusters, neither is sufficient to conclude quiescence or stability of a quiescent state. Flow cytometry analysis is necessary to discriminate G1 from G0.
- Finally, loss of quiescence following chemotherapy has been shown to be extremely brief in AML cells, and likely would not be detected 3 days post-treatment when the authors have performed their analysis.

In previous work, we and others have conclusively demonstrated a causal link between microRNA-126 expression and quiescence in AML LSC, as well as in normal HSC (Lechman*, Gentner* et al, Cell Stem Cell 2012; Lechman*, Gentner* et al, Cancer Cell 2016). Mechanistically, this is mediated by downregulating multiple targets feeding into the Pi3K/AKT/MTOR signaling pathway (reducing S-G2-M transit), as well as direct targeting of the cell cycle regulator CDK3 by miR-126, inhibiting G0 exit (Lechman*, Gentner* et al, Cancer Cell 2016). We have now performed Ki67/Hoechst stainings of miR-126^{high} and miR-126^{low} populations from multiple PDX from 3 patients. These confirm that a lower proportion of miR-126^{high} blasts are in S/G2/M as compared to the matched miR-126^{low} blasts. Moreover, in 1 out of 3 diseases, the miR-126^{high} cells also had a higher G0 cell fraction. We conclude that, at steady state, and when the murine BM space is not saturated, miR-126 is a faithful marker for cells with slower cycling characteristics. The manuscript has been revised as follows:

Cell cycle analysis on 3 representative diseases confirmed depletion of miR-126^{high} cells in the S/G2/M phase and enrichment in G0 (PT01) or G1 (PT03, PT16) (Supplementary Fig. 3B).

To address how cell cycle distribution changes in miR-126^{high} and miR-126^{low} fractions upon chemotherapy, we performed EdU incorporation during chemotherapy in mice (n=2 diseases, 16 mice). Moreover, Ki67/Hoechst staining was performed on day 8 of chemotherapy from sorted miR-126^{high} subpopulations showing that between 40 and 65% of cells maintained a G0 state after chemotherapy (n=3 diseases, 16 mice). Please note that the number of AML cells, which can be recovered after chemotherapy is extremely low, making secondary transplant of sorted miR-126^{high} and miR-126^{low} fractions practically impossible. Likewise, a dynamic analysis of cell cycle state in marked single cells would be desirable, yet out of the scope of this manuscript. Nevertheless, our data are in line with an interpretation that the miR-126^{high} fraction combines the potential (1) to remain quiescent and (2) to enter cycle driving leukemia regeneration. Indeed, both capabilities are necessary in a dynamically-timed manner to drive treatment failure, as also clearly illustrated by the relapse sample of Patient 08. Our work highlights how LSC quiescence and regeneration capabilities are expressed differentially between different AML diseases. This message has been clarified in the discussion. The manuscript has been revised as follows:

Interestingly, the samples from PT01 and PT03, where a distinctive miR-126^{high} population emerged, **were enriched in the G0 cell cycle phase (Supplementary Fig. 3G)** and manifested a higher LIC frequency in post chemotherapy samples compared to controls, while LIC frequency was reduced in samples from PT15, where miR-126 activity decreased (Fig. 3H and Supplementary Fig. 3I). Taken together, chemotherapy resulted in the enhancement of a miR-126^{high} LSC population in 2 out of 5 AML samples, which correlated **with quiescence** and higher LIC frequency.

EdU (5-ethynyl-2'-deoxyuridine) incorporation confirmed early proliferation of AML blasts after chemotherapy (Fig. 5I). Interestingly, this regenerative response was more evident in the miR-126^{high} population in 2/3 diseases, indicating that LSCs may switch to an active, proliferative state, while still maintaining a constantly high proportion of cells in the G0 cell cycle phase (Fig. 5J). We hypothesized that the 126High⁺ HSC-like cluster 1, which differed from most other clusters by downregulation of OxPhos and MYC proliferative responses upon chemotherapy exposure, may contain dormant LSCs with increased resistance to stress-induced activation (Fig. 5G).

We have also adjusted the language about dormancy/quiescence throughout the manuscript and clarified the significance of INKA1/CDK6 expression, as suggested by this reviewer.

Moreover, the HSC latency-associated gene INKA1³¹ was highly expressed in clusters 6' and 19', while the cell cycle kinase CDK6³², a robust marker for active HSC³²⁻³⁴ was mostly detected in cluster 5', suggesting that the former two clusters represented more dormant LSC states.

2. Stability of LSC populations before vs. after chemotherapy: In Figure 3H, the authors show limiting dilution transplantation of human AML cells after high dose chemotherapy treatment vs. non-treated controls. One patient sample has a significant depletion of functional leukemia initiating cells (LICs), whereas the remaining 4 patient samples have similar LIC frequencies in treated vs. control conditions (based on the adjusted P values shown). The authors actually make claims that are not supported by the statistics they show. 2 of the patients have slightly higher LIC frequencies but don't reach statistical significance. This raises concerns of a desire to draw a specific interpretation vs. objective assessment of the data. This functional result is consistent with their miR-126 reporter, which suggests a reduction of prospective LSC frequencies in 2/5 of the patients, whereas prospective LSC frequencies are unchanged for the remaining 3 patients (Supplementary Figure 3F).

- These results are nearly identical to the findings of Farge et al. (2017) where 3/4 patient samples showed no change in LIC frequencies after chemotherapy treatment, and the fourth patient sample showed a significant reduction in LICs. However, the authors have interpreted their findings completely differently from Farge et al. The authors need to explain what additional evidence leads them to this contradictory interpretation of essentially the same data.

- Based on their arguments, the authors appear to fundamentally misunderstand the controversy around the persistence of LICs/LSCs. Of course, if any functional LSCs can be detected following chemotherapy treatment, it can be argued that LSCs have persisted. However, if LSCs are protected from chemotherapy due to their quiescent features, they should be preferentially spared following chemotherapy treatment (i.e., found in higher proportions than pre-treatment). This is not the case in the current manuscript, nor in recent published reports from multiple groups. This suggests that LSCs are just as sensitive to chemotherapy as any other leukemic cell and therefore, the more relevant therapeutic limitation is the inability to escalate chemotherapy dosage without killing healthy cells. In contrast, it would not be productive to focus on chemo-sensitization of LSCs by targeting their quiescent properties. This is the essence of the insights raised by these recent studies, and the authors' data reinforces these same conclusions.

We respectfully disagree with this reviewer's interpretation of our data. As we have stated in the original version of our manuscript, 5 out of 5 assayed NPM1^{mut} AML (100%) showed a statistically significant enrichment of CD34⁺CD38⁻ cells following chemotherapy:

“We analyzed the mouse BM early after chemotherapy (day 8) and showed marked reduction of hCD45⁺ blasts in the treated mice (Fig. 3E), with a relative increase of immunophenotypically primitive CD34⁺38⁻ cells indicating that they exhibited chemotherapy-resistance (Fig. 3F)”.

More importantly, limiting-dilution transplantation, the gold-standard assay to detect functional LSC, clearly suggested that LSCs were enriched following a 3+5 day course of Daunorubicin + Cytarabine chemotherapy, respectively, in 2 of the 5 studied patients, namely PT01 and PT03. As the reviewer may acknowledge, secondary transplants of primary human AML following *in vivo* chemotherapy (where the BM is basically empty) is technically challenging, necessarily restricting the number of mice that may be assayed. Under these circumstances, we considered the p values of 0.036 and 0.055 as an indication for an increase in LSC frequency following induction chemotherapy. The p-value is the probability of rejecting the null hypothesis, assuming it is true. In general, the absence of evidence cannot be interpreted as evidence of absence of a difference (Altman and Bland, BMJ, 1995). The value of the p-value may largely depend on the sample size and not only on the effect size. Thus, its interpretation with respect to the standard significance level 0.05 should account also for the limitations due to the small sample size (Amrhein et al, Nature, 2019) and may not be strict. In any case, to gain stronger evidence of increased LSC frequency following induction chemotherapy, we have now analyzed the spleens of these mice (Fig. 3H). PT01 and PT03 show a highly significant increase in LSC frequency after chemotherapy (p=0.0023 and p=0.000195, respectively), fully confirming the BM results presented in the original version of the manuscript. Furthermore, it should be noted that in this specific situation, p value adjustment is not requested, since the aim of the analysis is to show patient-specific changes and not the “discovery” of an overall effect.

Our data are fundamentally different from those published by Farge et al. (2017), starting from a higher proportion of samples where the %CD34⁺CD38⁻ cells was enriched after chemotherapy (5/5=100% in our data, vs. 3/22=13.6% in Farge et al, Fig. 2A), stability of absolute CD34⁺CD38⁻ cell numbers and an increase of cells in the G0 phase of the cell cycle (see response to point 1), indicating LSC resistance. Notably, using an AraC/anthracycline combination, we applied a stronger selective pressure resulting in a mean 26.4 (range: 4.2-69.1) fold AML depletion, as compared to the 4-46 fold reduction reported by Farge et al., likely amplifying a differential response between LSC and leukemia bulk. As AraC + anthracycline treatment has formed the backbone of induction protocols with curative intent for more than 3 decades and, being still the standard of care today, we sustain that our experimental setup is clinically relevant. Secondly, the samples analyzed differ in their molecular characteristics. In our study, *in vivo* chemotherapy response was comprehensively tested on AML with NPM1 mutations and an adverse clinical outcome, which may have enriched for cases, in which LSCs are resistant to chemotherapy. Thirdly, the fact that these chemotherapy-resistant LSCs were low in OxPhos during first line treatment, both in xenografts and in patients, rather than the OxPhos high population described in other manuscripts (Farge et al, 2017; Baccelli et al, Cancer Cell 2019) underlines that we are

capturing a different biology in our paper. This is now further confirmed by new data, where we treated NPM1^{mut} AML with an OxPhos inhibitor (IACS-010759), which enriched for miR-126^{high} cells and increased leukemia engraftment following xenotransplantation (see above), in support of an OxPhos^{low}, quiescent LSC as the cause of initial treatment failure (see also our cell cycle analysis performed on this disease and described above).

Indeed, the true point of our paper is the claim that chemotherapy does enrich for quiescent LSCs in a subset of patients, and that these are the patients, which show refractory disease or early relapse, and require treatments other than chemotherapy. Prospectively identifying such patients with high confidence and in a timely manner will have substantial clinical impact, avoiding morbidity and mortality from ineffective chemotherapy. Such patients may indeed benefit from alternative induction approaches, e.g. Venetoclax + hypomethylating agents or emerging targeted therapies. When saying that we “fundamentally misunderstand the controversy around the persistence of LICs/LSCs”, this reviewer seems to overlook part of the literature, which continues to provide timely evidence for the “classic” LSC model (Shlush et al, 2017; Zeng et al, Nat Med 2022).

Differently, in relapsed NPM1^{mut} AML or AML with del(7), we see OxPhos^{high} LSCs after chemotherapy. The discussion has been revised as follows:

The 2 molecular AML subtypes we studied differed in their hierarchical structure, with del(7) AML showing a more shallow, stem cell dominant hierarchy and NPM1^{mut} AML maintaining a myeloid differentiation trajectory under homeostatic conditions, the latter more closely mimicking hematopoiesis. Indeed, evidence for a persisting, OxPhos^{low} quiescent LSC state as the substrate of chemotherapy selection was mostly obtained on NPM1^{mut} diagnosis samples. Very recently, the J. Dick laboratory has identified a quiescent LSPC cellular state in scRNAseq data and performed deconvolution analysis of bulk RNA sequencing from functionally characterized samples establishing a correlation between this quiescent state and LIC activity across a broad range of AML subtypes, suggesting a more general relevance of our findings⁴³. At relapse, miR-126^{high} LSCs were also strongly enriched, but showed metabolic rewiring and active proliferation reminiscent of progenitors, which may rapidly convert back to quiescence upon re-induction chemotherapy.

3. 126High signature: The survival analysis shown in Figure 6C is an impressive application of the 126High signature, however it is not clear what practical value this signature adds over the existing LSC-17 signature that has been previously described. The expression patterns of the 126High signature (Figure 6A) look nearly identical to that of the LSC17 signature (Supplementary Figure 6B). Furthermore, one of the same microarray datasets was analyzed in Ng. et al., 2016 (GSE37642) and already showed that patient survival can be predicted by primitive LSC signatures at diagnosis. Therefore, it is not surprising that a similar gene set would also predict patient survival.

We thank the reviewer for acknowledging the predictive power of our miR-126 signature on survival. The goal of this analysis was to demonstrate that the principle of our miR-126-based LSC signature has more general validity also in AML groups other than the NPM1-mutated subsets, similarly to the LSC17 signature. Our miR-126 signature contains different gene sets and is more sharply demarcated and localized to the most primitive clusters with respect to

LSC17, both in NPM1mut and del(7) AML. We found it was therefore relevant to show the predictive power of our signature. In terms of clinical application, we foresee a targeted single cell analysis rather than bulk RNA sequencing. Implementing this targeted scRNAseq in prospective AML cohorts to predict chemotherapy response will be a follow up work of this manuscript.

We added the following phrase to the discussion:

Prospective clinical implementation of targeted single cell approaches to quantify miR-126^{high} LSCs at diagnosis represents a promising next step to improve stratification of patients with intermediate risk AML.

4. Survival analysis: It is unclear why different gene lists were applied in Violin plots in Figure 6B (989 genes?) vs. survival analysis in Figure 6C (24 genes) vs. survival analyses in Supplementary Figure 6C (10 genes vs. 12 genes, where only 6 genes are shared).

- First, the rationale to reduce gene list size to 24 genes by correlation with a module score derived from the same gene list is unconventional and somewhat circular in logic.

We thank the reviewer for the comment and, reading it, we felt the need of better clarifying the rationale and methodology of this survival analysis in the revised version of the paper. The main goal of gene-list reduction was to reduce the 126^{High} signature to a lower number of informative genes, which can then be used more easily in practice. For this purpose, we used an approach commonly used in questionnaire reliability analysis, aiming at evaluating the explicative “role” of a gene in the module score. The item-test correlations are computed as the correlation coefficients between each item score and the total test score. Due to the distribution of the data, the Spearman correlation coefficient was employed. The greater the value of the coefficient, the stronger is the correlation between the item and the total test score. Questionnaire developers strive to select items for a test that have a high correlation with the total score to ensure that the test is internally consistent. The item-test correlation is often used to support the contention that the item is a “good” contributor to what the test measures. In our analysis, we defined the genes as “good” contributors to the module score as the ones with a positive non-negligible significant Spearman correlation (i.e. $\rho > 0.3$ and Bonferroni’s adjusted p-value < 0.05).

- Second, the application of Lasso penalty in subsequent survival analyses is inappropriate. This is a method for optimal model selection in a training dataset, which should then be validated in a testing set (as was done in Ng. et al., Nature, 2016). Here, the authors have independently applied Lasso penalty across 3 different data sets, which effectively selects the subset of genes for each dataset that are most biased towards a significant result.

Although Cox’s proportional hazards regression with lasso penalty (Tibshirani, Stat Med, 1997) is commonly used for optimal model selection, it is also commonly applied on gene expression data to overcome issues in regression estimation. In general, penalized regressions are more suitable than the corresponding standard regressions (in this setting, the Cox’s proportional hazards regression) in the presence of collinearity (i.e. highly correlated covariates). Collinearity is expected, for example, when dealing with the expression of genes within a signature. Penalized regressions also avoid overfitting in the presence of several covariates.

Validation is not always requested, but its need depends on the analysis purpose. For instance, in the original version of the manuscript, the aim of the analysis was to show an association of the expression of the 24-gene list with overall survival, rather than proposing a tool that could be automatically implemented for clinical decision making based on the expression of a subset of genes. Thus, validation was not strictly requested.

In the manuscript, three different public datasets were employed for the evaluation of this association. The different datasets were used separately, since they contained different types of expression data (microarray, RSEM batch normalized RNAseq and a CPM RNAseq data), which are not formally comparable. Due to the different types of platforms used, a different subset of the 24-gene list was present in each of the datasets. The models obtained in the three datasets were different due to the different types of expression measurements and the gene sets used. The original analysis with the above-described purpose is now presented in the new Supplementary Fig. 6D and Supplementary Table 6. We have also revised the Supplementary Table 6 to show explicitly the subset of the 24-gene list that was present in each dataset.

- If the 126High signature were to be applied prospectively to new patients, which is the robust gene list that should be used? 24 genes? 10? 12? In the prospective setting, it is not possible to refine a gene list based on the patient outcome, as the outcome remains unknown. If a gene list cannot be applied prospectively, it has little practical value.

As stated above, in the original version of the manuscript, the aim of the analysis was to show an association between expression of the 24-gene list with overall survival. In order to also provide a tool that could be used in clinical practice, we added a new analysis in the revised version of the manuscript. Since this kind of analysis needs the use of a training set and a testing set including the same subset of genes, we employed in the analysis only the Microarray dataset (as training set) and the OSHU RNAseq dataset (as testing set), which had 20 genes of the list in common. Considering also the TCGA RNAseq dataset as testing set would have reduced the list of genes to 15 with a high loss of information. Due to the substantial heterogeneity of data among platforms and measurements, we used a training-testing procedure based on categorized expression data. Please note that this is a stringent approach, which may underestimate the predictive power of our signature. In detail, for each patient, the expression data was categorized into -1 (downregulated), 0 (normal) and 1 (upregulated), by using their first and third quartiles. This categorization is the only way to make data measured on two types of platforms comparable. The resulting model has the big advantage of generalizability, and it can be applied to expression data of any platform after preprocessing with the same type of categorization. We then applied a methodology similar to the one used in the association analysis with OS, for the definition of the risk groups in the training set (Cox's proportional hazards regression with lasso penalty on the categorized expression data and then categorizing the resulting linear combination of the genes with the median). By applying the same linear combination and cut-off to the categorized expression data of the testing set, we found that the two risk groups confirmed their association with OS (log-rank $p=0.0343$).

The revised manuscript has been modified as follows:

To further confirm that the 126High gene signature measured at diagnosis was associated with patient outcome, we analyzed publicly available datasets encompassing multiple AML subtypes with respect to overall survival (OS). We first reduced our signature to 24 informative genes by selecting those with a significant correlation (Spearman rho >0.3, Bonferroni p <0.05) between their expression and that of the overall gene signature in our NPM1^{mut} scRNAseq dataset (Supplementary Table 3). Next, we estimated a Cox's proportional hazard model with lasso penalty on the standardized intensity of the transcripts on 3 independent AML datasets separately, encompassing n=767, n=337 and n=161 non-M3 AML patients, respectively. In all three datasets, the two corresponding risk groups (obtained with the median split of the linear combination of the genes of the model) showed clearly separated survival curves (log-rank p<0.0001) (Supplementary Fig. 6D, Supplementary Table 6). To identify and validate our risk groups, we used n=767 patients from the microarray study as a training set, and n=337 patients from the OSHU RNAseq dataset as the testing set. Twenty of the 24 signature genes were expressed in both data sets, a greater overlap than in the TCGA data set, where only 15 signature genes were in common. To compare gene expression data obtained by different platforms, we performed a stringent quartile-based categorization of the expression data of each patient into -1 (downregulated, below the 1st quartile), 0 (normal, between the 1st and 3rd quartile) and 1 (upregulated, above the 3rd quartile). By applying the Cox's proportional hazards model with lasso penalty on the categorized expression data of the training set, we identified two risk groups with clearly demarcated OS (log-rank p<0.0001) and confirmed a significant association with OS (log-rank p=0.0343), also when applied to the test cohort (Fig. 6C).

Please note that the OSHU cohort showed poorer outcome than the TCGA or the microarray studies, making a distinction between different subgroups less evident. Under these very stringent conditions, the fact that our model resulted in a statistically significant separation of the OSHU cohort strongly argues for the robustness of our score.

5. Human patient scRNA seq analysis: The “patient-tailored” FACS processing of human AML patient cells prior to scRNAseq analysis is concerning. If the authors desired to remove non-leukemic lymphoid and erythroid cells, they should simply stain for lymphoid and erythroid markers and apply an exclusion gate similarly across all patient samples, in order to preserve the composition of the myeloid populations.

- It is unclear why samples were instead fractionated into multiple different subpopulations and then re-combined. Even if an effort was made to preserve their original proportions, this would be difficult to accomplish accurately without introducing bias.
- Furthermore, different sorting strategies were used for different patients, and even samples from the same patient were subjected to different sorting gate strategies at separate time points (PT01). Therefore, it is unclear how samples can be directly compared to each other; e.g., it is inappropriate to describe a near extinction of myelo-/monoblastic cells after chemotherapy treatment if it is possible that the proportions of these cells could have been inadvertently biased by sorting gate or cell pooling steps.
- This is further complicated by the cluster-based assignment of NPM1 mutational status, where only less than half of cells classified as “AML” at diagnosis actually have direct evidence of NPM1 mutant transcripts.

All sortings have been prospectively planned and performed following careful study of the clinical flow cytometry results of each patient. When progenitor markers were applied, the diseases did not have a relevant mature population to be included. Retrospective assessment of the sorting plots confirmed that we captured the majority of blast cell populations, at least qualitatively, for all diseases and timepoints. We acknowledge that a common sorting strategy would have been more elegant. However, the different sample characteristics and the extremely low amounts of cells obtainable post chemotherapy mandated for a tailored approach, maximizing recovery of AML cells. We would argue that precise proportions are already intrinsically biased by the freezing step, resulting in different recovery of subpopulations (mature blasts tend to have lower recovery than progenitors). Cryopreservation is, however, the only practical way to routinely perform costly single cell sequencing analyses on patient material. We never made quantitative claims on specific clusters, but rather described the qualitative differences between AML cellular states. Indeed, the statement that myeloid differentiation states disappeared was based on the overall combined dataset, not on a single patient basis. In any case, we have moderated this claim.

Regarding the cluster-based assignment of NPM1-mutational status: due to “drop-outs” inherent to this scRNAseq technology, it is expected that mutational calling results in false negatives. Importantly, when comparing within the same cluster the cells where the NPM1 mutation is detected (MUT) versus the remaining cells which have been included in the analysis by the NPM1-MF pipeline and decision algorithm, we found very few differentially expressed genes, excluding NPM1 or ribosomal protein genes, except for Cluster 7, as reported below. We acknowledge that Cluster 7, which we defined as “erythroid-like”, may thus contain some non-malignant erythroid precursors, although AML cells harboring the NPM1 MUT transcript showed higher expression of erythroid lineage transcripts as detailed in Supplementary Table 2. However, this cluster is not central to any of our claims in the manuscript.

The text has been modified as follows:

Post-chemotherapy leukemic cells prominently mapped to the progenitor clusters, **with underrepresentation of differentiated myelo-/monoblasts**, but enrichment in the erythroid-like clusters and accumulation of cells bridging the early and myeloid immature-like progenitor area on day 30 (Fig. 2A,B).

Within a given cluster, cells in which mutant NPM1 transcripts were detected were transcriptionally very similar to cells, in which insufficient read counts were obtained due to known limitations of the scRNAseq technology (Supplementary Fig. 1C), confirming accurate leukemia cell classification and reliable cluster-based AML annotation.

INTRA-Cluster comparison of AML cells harboring the NPM1mut transcript (MUT cells) vs other cells classified as AML by NPM1-MF		
Cluster	# DEGs logFC >0 & p.adj <0.05	DEGs:
0	2	NPM1, RPL31
1	2	NPM1, RPS4Y1
2	4	RPS4Y1, ELANE, SPINK2, RPS10
3	2	NPM1, RPS4Y1

4	8	NPM1, RPL10, RPS3, RPS18, RPS5, RPL10A, RPS10, RPS26
5	1	NPM1
6	1	NPM1
7	224	See Supplementary Table 2, "Intra-cluster 7 MUT vs Rest"
8	1	NPM1
9	2	NPM1, REXO2
10	0	
11	2	NPM1, RPL13A
12	0	
DEGs were computed with a logistic regression framework in Seurat to determine differentially expressed genes while controlling for PatientID and Timepoint latent variables.		

Moreover, differently to what we classified as AML, the cells classified as “non-leukemic” by our cluster-based assignment precisely overlap with healthy donor CD34+ cells on an extensive scRNAseq/CiteSeq dataset we have generated. These data will be subject to a follow up publication on the healthy progenitor cell populations from AML patients.

6. Controversy in the field: In the Discussion section, the authors claim to provide a missing link that reconciles classical LSC models of therapy resistance vs. more recent challenges to the concept of preferential persistence of LSCs. It is not clear how any of their data re-contextualizes these recent studies that were based on functional analysis of AML, particularly given that the authors’ study concentrates on molecular descriptions rather than functional biology. The only functional biology that they do show (Figure 3G) demonstrates that functional leukemia-initiating cells either significantly decrease or are unchanged following high dose chemotherapy treatment. This is inconsistent with classical notions that LSCs would be preferentially spared by chemotherapy treatment. However, the authors emphasize classical ideas of LSC persistence and dormancy throughout the written text.

Rather than resolving biological processes, the strong conclusions in the absence of clear data is adding to confusion.

Please see our response to point 2 on our interpretation of the data. Our conclusions are based on a combination of functional studies and cutting-edge molecular characterization done on primary AML cells at steady-state and after chemotherapy, in xenograft mouse models and directly in patients. To our knowledge, no other work has described a similarly deep single cell characterization of AML cells after chemotherapy. The discussion has been adapted to better highlight the controversies in the field, rather than trying to reconcile evidently different views, as proposed in the original version of the text.

Additional points:

1. In Main Figure 4B, the same gene list is shown as both an upregulated and downregulated

gene set when comparing GFP low vs GFP high populations (JAATINEN_HEMATOPOIETIC_STEM_CELL_UP).

We thank the reviewer for pointing out this oversight. We have corrected Figure 4B, right panel, with the appropriate label reading JAATINEN_HEMATOPOIETIC_STEM_CELL_DOWN instead of JAATINEN_HEMATOPOIETIC_STEM_CELL_UP, as also reported in the figure's native figure legend.

2. In Figure 6G-I and Supplementary Figure 6D-E, the LSC sub-clustering is not convincing. It is evident from Supplementary Figure 6E that there is little distinction between the subclusters, despite that this heatmap is showing the top 10 marker genes for each subcluster. It is not clear how these panels add to the story.

While we agree with the reviewer that LSC subclustering is performed within a group of highly-similar progenitor-like AML cells, which may blur distinction between clusters, we nonetheless retain that further separation of a chemo-refractory quiescent LSC subset (cluster 1, New Fig. 6D,E) from the rest of the diagnosis LSCs, represents an interesting and novel finding which highlights the potential of single cell resolution approaches and conveys important clinical and biological implications. Identification and quantification of highly-refractory LSC subset within diagnosis or remission samples could open new possibilities for early and prospective disease stratification as well as promote the development of novel therapeutic approaches for disease subtypes with very poor prognosis such as chemorefractory AML.

3. The terminology related to the reporter system is very difficult to follow because it is inconsistent throughout the figures, including the direction of the axes and how this relates to miR-126 activity. In Figure 3C the x axis is labeled "1/TransGene Ratio" then the axis is reversed in Figure 3G ("TransGene Ratio") and Figure S3 ("TGR"), which then reverts back in Figure 5F ("126High signature"). This is counter-intuitive and would be easier to keep the axes consistent, such that higher values always indicate higher miR-126 activity.

We thank the reviewer for highlighting this source of confusion and giving us the opportunity to better represent our data. In the revised version of the manuscript, we have now inverted the axis of Fig. 5F so that it is consistent with Fig. 3G and Supplementary Fig. 3G, allowing for easier comparison of miR-126 activity plots from both flow cytometry and single cell expression data. We, respectfully however, have not changed panel 3C as, in our opinion, it graphically better conveys the striking positive correlation between miR-126 activity (measured by the miR-126 sensor vector) and LIC frequency.

4. In Figures 3B and 3C, the authors have excluded PT15 from their analysis. Why? Based on Figure 3H, this patient sample appears to have the highest baseline LIC activity, suggesting it could have presented a greater challenge to define clear LIC- vs LIC+ populations. This patient is also completely omitted from molecular analyses in Figure 4, which is unfortunate given the unique functional biology associated with this patient in Figure 3. It is important to capture as much diversity as possible in molecular studies of AML.

While we surely agree with this reviewer on the relevance of capturing as much diversity as

possible during scientific investigations, we hope that the reviewer may appreciate that the choice of performing all experimental investigations with primary human samples, while adding great technical and sample-availability challenges, represents an added value to our work. Unfortunately, PT15 AML cells had a very low yield of miR-126 sensor vector-transduced blasts which impaired the execution of downstream transduction-dependent experiments such as the GFP-population based LDA and bulk RNA sequencing. We did however characterize PT15 AML in single cell RNA sequencing from control and treated xenografts, as well as from ex vivo patient samples with compatible findings as from other patients within our study cohort (i.e., Figure 1C, 2E, 2F, 5E, 6I).

5. A comparison of Figure 4D vs. Figure S4E suggests that miR-126^{high} vs miR-126^{low} AML subsets have nearly identical gene expression responses to chemotherapy treatment. Furthermore, in Supplementary Figures 4C and 4D, the authors show no change in LSC17 score before vs. after chemotherapy in either purified population. This is inconsistent with claims of enhanced features of stemness among miR-126^{high} populations in Figure 5.

As stated within the original manuscript, we do observe a conserved transcriptional response to chemotherapy across both miR-126^{high} and miR-126^{low} blast subpopulations, especially when investigated at the bulk resolution:

“We next interrogated the transcriptomic data for chemotherapy-related effects on AML blasts. Upon therapy, we found, in **both miR-126^{high} LSCs and miR-126^{low} blasts**, an enrichment of hallmarks related to inflammatory signaling, apoptosis, angiogenesis, KRAS signaling, epithelial-mesenchymal transition and heme metabolism (Fig. 4D, Supplementary Fig. 4E).

In Supplementary Figures 4B and C (previously Supplementary Fig. 4C and D), although limited by low number of samples, we do observe higher LSC17 scores for miR-126^{high}/GFP^{low} samples after chemotherapy in 3 out of 4 patients along with also higher expression of the ADGRG1 gene (GPR56) which has been reported as a robust LSC marker both when measured at the protein and transcript level (Pabst et al, Blood 2016 [PMID: 26834243]).

Recently a senescence mediated reprogramming of more differentiated blasts towards stem-like profiles has been reported in AML after chemotherapy (Duy, Cancer Discovery 2021 [PMID: 33500244]). While we also observe induction of senescence transcriptional features within both miR-126^{high} and miR-126^{low} compartments, we do not observe similar induction of stemness features within the miR-126^{low} compartments.

As such, these findings nicely reconcile with the enhanced stemness features reported in the single cell RNA sequencing data presented in Fig.5, with a reinforced stemness state being observable only within the miR-126^{high} LSCs, and further support, at least in our model of NPM1^{mut} AML, the hypothesis of a predetermined cell state with intrinsic chemotherapy resistance.

REVIEWERS' COMMENTS:

Reviewer #1 (Remarks to the Author):

The authors have thoroughly responded to my comments and made appropriate modifications to the manuscript to address my concerns.

Reviewer #2 (Remarks to the Author):

The authors have performed additional experiments and made new or modified claims to the study. Eg. "chemotherapy does enrich for quiescent LSCs in a subset of patients...' these patients show refractory disease or early relapse, and require treatments other than chemotherapy "

This is a very different conclusion and claim than the previous version, despite the wording changes by the authors. This would indeed have value, but the distinction between quiescent before vs. after or during chemotherapy is still not addressed, nor is the basis of this patient stratification and could be stochastic in readout and not biological stable, and new treatments (suggestion or target) is not provided.

This work has improved, and would be interesting for a focussed and specific group of clinician-scientists and worthy of publication. The advance, given the gaps in our current understanding of AML disease is very unclear. For example, the authors indicate the relationship of miR126 has already been made, and published, and the reuse of this reporter and recreation of this established biology is circular. Even the quiescent nature of some cells has been suggested by Dick's group recently in a published study, further questions the advance and relevance of this study.

RESPONSE TO THE REVIEWERS' COMMENTS

Reviewer #1 (Remarks to the Author):

The authors have thoroughly responded to my comments and made appropriate modifications to the manuscript to address my concerns.

We sincerely thank the Reviewer for his/her time and commitment to the revision of our manuscript.

Reviewer #2 (Remarks to the Author):

The authors have performed additional experiments and made new or modified claims to the study. Eg. "chemotherapy does enrich for quiescent LSCs in a subset of patients..." these patients show refractory disease or early relapse, and require treatments other than chemotherapy ". This is a very different conclusion and claim than the previous version, despite the wording changes by the authors. This would indeed have value, but the distinction between quiescent before vs. after or during chemotherapy is still not addressed, nor is the basis of this patient stratification and could be stochastic in readout and not biological stable, and new treatments (suggestion or target) is not provided.

We thank the Reviewer for acknowledging the value of our studies and the effort that has been put into the revised version of our manuscript. We would like to point out that the interpretation of our data has remained constant throughout the revision process and that, in both the original version and the revised text, our claims remain consistent. In the original manuscript, we claimed that "Dormant leukemia stem cells reinforced by chemotherapy drive early treatment failure in acute myeloid leukemia patients", which has been toned down to "Quiescent leukemia stem cells reinforced by chemotherapy drive early treatment failure in acute myeloid leukemia patients" in the revised version. Such claims have been thoroughly discussed throughout the original version of the manuscript and were summarized in the letter of response to the Reviewer's comments, as quoted by Reviewer 2 above.

To the best of our knowledge, there are only a handful of studies addressing the response of AML blasts to the early, yet crucial, phases of treatment. Such studies were conducted by analyzing all persisting blasts in "bulk" fashion and thus lack the depth of characterization presented in our manuscript, where we employed state-of-the-art single cell readouts. Although scRNAseq allows the profiling of thousands of cells from a single timepoint, this technology is per se not able to map the fate of single cells longitudinally, but relies on inferring population dynamics by analyzing serial samples. To fate-map single LSCs in vivo, and thus to discriminate between an "a priori" quiescent LSC from an LSC which enters quiescence upon exposure to therapy, is, in our opinion, out of the scope of this manuscript. Such distinction would entail sorting live LSCs according to cell cycle status, labelling of sorted LSC fractions (e.g., by genetic marking with a barcode library), exposing these sorted and uniquely-labeled LSC to chemotherapy, possibly in vivo, and sampling of surviving LSCs for re-evaluation of cell cycle status and back-tracking to prior cell cycle state. Each and every single one of these steps is technically very challenging or unfeasible when applied to clinically-relevant patient-derived samples or xenografts, the latter representing the gold standard for studying LSCs.

Nevertheless, our work allows us to hypothesize different scenarios for the appearance of quiescent miR-126^{high} LSC after therapy. One possibility is the persistence of a quiescent baseline LSC population throughout chemotherapy. Such hypothesis aligns with our observations that a prominent miR-126^{high} LSC content prior to therapy associates with chemotherapy-refractory AML, and with the unchanged proportion of G0 LSC that we observe at baseline and after therapy. We cannot exclude the recruitment of activated/cycling LSC to quiescence upon chemotherapy. Yet, such a hypothesis diverges from the observed patient risk prediction derived from miR-126^{high} LSC content at diagnosis, making this second possibility far less likely. Indeed, we showed, in both the original version and in the revised manuscript, enrichment of quiescent LSCs at diagnosis in patients refractory to conventional chemotherapy or with early disease relapse.

Although open questions remain regarding the fate of single LSCs, our detailed work providing multiple longitudinal snapshots on the LSC population in patients and in xenograft models goes beyond currently available knowledge and is thus very important, as we could learn valuable biological insights applicable to clinical translation. For instance, our data shed new light on the use of OxPhos inhibitors, a strategy which has been advocated in the recent AML literature. Our scRNAseq data predicted that this therapeutic strategy may not be effective in newly diagnosed NPM1-mutated AML where miR-126^{high} OxPhos-low LSCs persist. Indeed, we presented experimental evidence in the revised version of the manuscript that OxPhos inhibitor treatment selects for LSCs, which rapidly lead to leukemia regeneration. We also describe the expression of classical lymphoid markers in miR-126^{high} LSC, which could promote the rationale for investigating lymphoid marker-directed immunotherapies, e.g. against CD7 and TNFRSF4 (OX-40), as their expression within the LSC compartment may be more pervasive than currently thought.

Importantly, the stratification of AML patients we propose is clearly not stochastic. Patients who exhibit miR-126^{high} LSC-like blasts at diagnosis show refractory or early relapsing disease, both when analyzed at single cell resolution and when translated to risk prediction in large bulk RNA sequencing AML cohorts, where sample sizes allow to exclude stochasticity.

This work has improved, and would be interesting for a focussed and specific group of clinician-scientists and worthy of publication.

We respectfully disagree with the Reviewer and feel our work may be of interest to a broader audience of both clinicians and scientists working in the field of hematological and solid cancers, from translational to more basic research. First, we have applied state-of-the-art technologies to longitudinal clinical samples directly from patients, which are both difficult to retrieve and challenging to process, generating, to our knowledge, the largest scRNAseq dataset of early post chemotherapy/MRD AML blasts. This alone highlights the relevance of our data as a resource for the scientific community, especially, but not only, for investigations in AML and LSC biology. Furthermore, our analysis pipeline has been thoroughly optimized and may be easily translated to the analysis of datasets from different disease entities. Any group performing AML scRNAseq analysis or LSC characterization would be highly interested in seeing these data published.

The advance, given the gaps in our current understanding of AML disease is very unclear. For example, the authors indicate the relationship of miR126 has already been made, and published, and the reuse of this reporter and recreation of this established biology is circular. Even the quiescent nature of some cells has been suggested by Dick's group recently in a published study, further questions the advance and relevance of this study.

As the reviewer correctly points out, there still are many gaps in our knowledge of AML biology, LSC dynamics and chemotherapy responses. Literature on such hot topics has often been contradictory. For instance, Farge et al. and Boyd et al. have challenged the LSC concept by describing a population of leukemia regenerating cells with OxPhos-high status, challenging the long-advocated dogma that LSCs are intrinsically resistant to chemotherapy. Shlush et al., instead, have very elegantly demonstrated that AML relapse originates in a significant number of cases from rare LSC-like cells already present at diagnosis. By studying chemotherapy-response in patient samples with single cell resolution, we find evidence for both instances: In NPM-mutated AML, we detect LRC-like populations within more committed OxPhos-high progenitors undergoing active proliferation after therapy, which however coexist with persisting miR-126^{high} quiescent LSCs. Such fine observations may be captured only at single-cell resolution, which none of the above studies had employed. The reconciliation of discrepancies, particularly if directly within primary patient samples, should be, in our opinion, valued as relevant advances for the field of AML and LSCs.

Furthermore, as mentioned above, we have generated a well characterized single cell RNA sequencing dataset on rarely available clinical samples, which may serve as a valuable resource for the community.

Regarding the prior art on miR-126, it was our intention to exploit the miR-126 reporter technology only as a tool for robust validation of findings derived from new technologies such as single cell RNA sequencing applied to longitudinal clinical samples. Indeed, we clearly cite in the original text all relevant literature regarding miR-126 in AML without making novelty claims on the biology of miR-126 in AML, but rather on the transcriptional characterization of miR-126 defined blast subpopulations directly within clinical samples, their response to chemotherapy and their relevance for patient prognostication.

During the evaluation and revision of our manuscript, the work of John Dick emerged, which through different and complementary approaches highlights, as we also do, the relevance of LSC populations for the prognostication and prediction of treatment response in AML patients. Our paper dwells deeper in experimental modeling and characterization of transcriptional responses of AML and AML LSCs to chemotherapy, rather than focusing on prognostication alone. The emergence of independent manuscripts describing complementary and non-redundant findings, should be a token of reproducibility in science and an added value to our work. To this purpose, we understood that Nature Communication's policy, as made explicit within editorial correspondence, would not use a newly published paper during the review process as an argument for rejection.

REVIEWER COMMENTS

Reviewer #2 (Remarks to the Author):

General over/mis-interpretation and lack of data to support claims continues to inflict this study. Furthermore, the authors now have added data on a new genetic subgroup of patients (chromosome 7 deletion... it is unclear why as this was not raised or requested). However, this new data conflicts with the authors original narrative, and instead suggests that for these patients the primitive cells get recruited into cycle transiently similar to Farge, Boyd etc,. From a conceptual point of view, the variable observations seen across patients makes it difficult for the authors to clearly resolve the controversies in the field that they set out to address. Although investigated with state-of-the-art technology, the study does not present a new understanding of the underlying biology, and the observations are difficult to clearly interpret, in part due to reliance on molecular descriptions. The authors' main defence of their claims rests heavily on the observations that miR-126high LSC content at diagnosis predicts therapy outcomes in patients, however the observation that LSC signatures predict survival is not surprising or novel.

I have provided very detailed inconsistencies, and concrete comments below. I hope this is helpful to the editors and authors alike.

Inconsistencies:

- The authors now introduce a second molecular subtype of AML (del7), with a different pattern of chemotherapy response than their initially characterized NPM1+ AML patients. Specifically, they describe a transient proliferative response across all primitive subsets at Day 14 post induction chemotherapy. This challenges the overarching conclusion that cell cycle dormancy represents the central mechanism of therapy failure in AML. It instead suggests that the original conclusions of the study are restricted to NPM1+ subsets of AML, where results were even inconsistent across different patients examined (e.g., Fig 3H, 5F, 5I).
- The authors also provide new evidence that at relapse, miR126High subsets consistently showed significant cell cycle activity, unlike miR126High cells at diagnosis (Fig 6H). Relapsed AML is notoriously more chemoresistant than initially diagnosed AML (e.g., as seen for Patients PT08, PT10, PT15, and PT19 in Figure 1A), yet Fig 6H suggests that relapsed LSCs are less quiescent than at diagnosis. This observation opposes the central conclusion that LSC quiescence is the main determinant of chemoresistance, but is consistent with the more recent published literature. For some reason, the authors refuse to acknowledge this fact.
- The authors have also incorporated new data showing that miR126High cells can show increased EdU incorporation following chemotherapy treatment (Fig 5I), yet they do not acknowledge this as evidence that LSCs may be recruited into cell cycle by chemotherapy. Why? Early in the manuscript, they firmly assert that miR126 faithfully identifies functional LSCs. However, they avoid using the term LSCs in their interpretation of this cell cycle result, and instead simply address this observation as early proliferation of miR126High "progenitors". ??? There is no functional or molecular data to support this distinction of LSCs vs transient leukemic progenitors within the cycling miR126High populations, other than the authors' personal views.
- There is a general disconnect between molecular descriptions and functional assessments. Some of the strongest claims are derived from scRNAseq data, such as the persistence of LSCs with "enhanced stemness features upon in-vivo chemotherapy". In contrast, functional assessment of LSCs does not suggest evidence of enhanced stemness, and in fact sometimes suggests reduced stemness (Fig 3G). This is a problem, since functional assays represent the gold standard for studying LSCs (as acknowledged by the authors in their response).
- Overall, the Title, Abstract, and Discussion oversimplify the complexity of the biology and fail to address the fact that the authors have exclusively examined NPM1+ AML in most of their figures. In the absence of functional intervention experiments in vivo or other sophisticated cell tracking experiments as the authors described, it is not possible to infer what cells are responsible for driving therapy failure.

Specific Scientific Comments:

Comment 1:

Lines 62-64: Knowledge on the cell biology and population dynamics of relapse-promoting cells during chemotherapy is scarce, with the general assumption that their proportion tends to increase with successive cycles of chemotherapy, to which LSCs are resistant.

No citation is provided. As the authors state earlier this is a controversial claim (lines 28-29) and having this statement without a source is setting up a "begging the question" fallacy as it orients their final conclusion. A study in which LSCs have been shown to be consistently preferentially resistant to chemotherapy functionally is required, aside for the dogma and belief. This does not mean to provide a paper that shows they are clinically predictive (Ng 2016) or sometimes retrospectively shown origin of relapse (Shlush 2017) as these are different statements entirely. There are a handful of papers from independent groups that show that LSCs are functionally and phenotypically depleted (Farge, Duy, Boyd) and it is unacceptable to make this statement and speak about the bias for generation of conclusions.

Comment 2:

Lines 80-82: As part of this score, miR-126 was tightly coupled with LSC and HSC function preserving their quiescent state and promoting chemotherapy resistance by orchestrating a bona fide stem cell program (Lechman 2016).

I once AGAIN suggest the authors critically analyze the work that provides the foundation of their analysis. In Lechman et al 2016, it is apparent that although a conclusion of the paper, there is no data conclusively supporting that miR-126 induces LSC specific chemotherapy resistance in primary AML. The paper demonstrates that miR-126 cells are enriched post chemotherapy in vitro, but miR-126 is part of general cellular senescence marker biology as the authors state (lines 188-190), and many cells within the hematopoietic/leukemic systems are non-cycling (dc/monocytes/macrophages) and express miR-126, as the authors acknowledge (lines 272-273). With evidence suggesting that monocytic leukemia's demonstrate more chemotherapy resistance (Pei 2020, Canaani 2017), I respectfully urge the authors to consider that enrichment of miR-126 is selecting for non-cycling cells, and not LSCs specifically.

I acknowledge that the miR-126 population is enriched for LSC functional activity, but a maximum 20fold enrichment is not enough to rule out the enrichment of other cell cycle dormant cells within the miR-126 high population, and what the Lechman 2016 Figure 7 conclusion should be is that quiescent cells are resistant to chemotherapy. This is not a novel finding, as we understand the mechanisms of action of cytarabine affects cycling cells. This unfounded statement combined with not citing more conclusive evidence providing the opposite idea, where LSCs are recruited into the cell cycle following a chemotherapy exposure (Boyd et al 2018) in more biologically relevant models (in vivo and from patients), the authors once again attempt to adhere to an unproven ideas, which they hold and use for interpretation, while developing their unsupported claims.

Comment 3:

Lines 144-147 Post-chemotherapy leukemic cells prominently mapped to the progenitor clusters, with underrepresentation of differentiated myelo-/monoblasts, but enrichment in the erythroid-like clusters and accumulation of cells bridging the early and myeloid immature-like progenitor area on day 30 (Fig. 2A,B).

One of the final conclusions of the paper is that LSCs are enriched post chemo, which is contrary to this finding and not acknowledged by the authors.

Comment 4: 176-178: Due to the limited understanding of the response of LSCs to induction chemotherapy, and how this contributes to early treatment failure.

Once again, there is no evidence besides theoretical that LSCs contribute to early treatment failure

Comment 5: 190-192: Cell cycle analysis on 3 representative diseases confirmed depletion of miR-126high cells in the S/G2/M phase and enrichment in G0 (PT01) or G1 (PT03, PT16) (Supplementary Fig. 3B).

This data supports that quiescent cells are preferentially resistant to cytarabine, which is not a novel finding.

Comment 6: 198 – 201: We analyzed the mouse BM early after chemotherapy (day 8) and showed

marked reduction of hCD45+ blasts in the treated mice (Fig. 3E), with a relative increase of immunophenotypically primitive CD34+38- cells indicating that they exhibited chemotherapy resistance (Fig. 3F).

The authors must report the total number immunophenotypically primitive AML cells, or alter the statement to say partial chemotherapy resilience. With what appears to be a ~10-100 fold decrease of hCD45+ cells in the grafts, the total number of immunophenotypically primitive cells post likely decreases following treatment. The statements of conclusion do not coincide with the data being reported.

Comment 7: Line 201: "While immunophenotypically-defined LSCs increased"

This statement is incorrect. A statistical increase in cell proportion but on current presented data there is no evidence of an increase in immunophenotypically defined LSCs. See above statement. The embellishment of this data to support their hypothesis is unacceptable.

Comment 8: Line 201-202: Overall miR-126 activity remained stable or decreased after chemotherapy.

This statement is in opposition to the conclusion of this paper, and that fact is not acknowledged by the authors. The terminology "remains stable" is misleading, as this actually means the cell population decreased in the same proportion as the rest of the bulk leukemia (~10-100x). Another attempt to support the authors narrative.

Comment 9: Figure 5H:

The inconsistent nature of LSC response to chemotherapy demonstrated here (1 responds, 2 don't change, 2 are enriched [one of which has $p = 0.055$ which is not significant]) speaks to the inconsequential nature of LSC persistence following chemotherapy treatment as it relates to relapse. There is not enough evidence here to conclude that LSCs preferentially survive treatment, working against the hypothesis of the paper, and is not acknowledged or considered by the authors while developing the conclusions. When analyzing in an unbiased way, the most reasonable explanation of this data is that functional LSCs are not preferentially spared. This highlights a troubling pattern of selecting pieces of data that supports their ideas and claims vs looking at the data generated in its collective form.

Comment 10: 265-268: While steady-state control AMLs displayed enrichment for progenitor and cycling populations, chemotherapy radically reshaped AML, with expansion of more differentiated myeloid and, especially, erythroid-like blasts (Fig. 5B,E). On the other hand, the proportion of HSC-like blasts remained stable.

This result is in contrast to the authors hypothesis and claims, and is not considered in the conclusions. HSC like blasts are most enriched for miR-126 and are not enriched following treatment and depleted to a similar level as the bulk disease, indicating chemo sensitivity. The term "remains stable" used by the authors needs to be adjusted as it implies an absence of chemotherapeutic effect. The proportion of cells is similar, which means it reduced just as much as the bulk leukemia (~10-100 fold reduction).

Comment 11: 305-312: If these OxPhoslow LSC were at the origin of relapse, we hypothesized that these diseases would be resistant to pharmacological OxPhos inhibition. Hence, we treated miR-126 reporter+ AML cells from PT16 in vitro with IACS-010759 (IACS)32, a selective mitochondrial electron transport chain complex I-inhibitor, alone or in combination with AraC chemotherapy. IACS-treated cells had higher miR-126 activity (Supplementary Fig. 5E-G), consistent with selection of OxPhoslow LSCs. Strikingly, as compared to the mock-treated group, IACS-treated cells showed strongly enhanced repopulation activity when transplanted into NSG mice, underlining the functional relevance of these miR-126high OxPhoslow LSCs, at least in NPM1mut AML (Fig.5N).

It is unclear how this experiment supports OxPhos low LSCs are the origin of relapse as stated by the authors ?

This experimental design shows that functional LSCs are selected for by treating with IACS. It speaks nothing to the origin of relapse. Relapse is a clinical state of regenerated disease and not biologically represented or modelled in this experiment. Leukemia initiating activity as represented by LSC engraftment in immune deficient mice is a different biological process than chemotherapy induced disease regeneration resulting in clinical relapse and there is no scientific rationale to

compare the two. Drawing this conclusion while the data supports no such claim is misrepresentation of the results.

Comment 12: 314: miR-126high LSCs at diagnosis correlate with AML outcome
LSC markers being predictive of AML outcome is not a new.

Comment 13: 315-316: Our xenograft data suggest that a reinforced LSC state, identifiable by high miR-126 activity and persisting through induction chemotherapy, may be responsible for treatment failure

miR-126 activity has not been demonstrated to preferentially persist through chemo treatment in the xenografts. Other cell types appear to exhibit more preferential chemotherapy resistance. The same statement about being "responsible for treatment failure" can be made about any other cell type that persists through chemotherapy, and making this statement exclusive to LSCs is misrepresentative

Comment 14: 370-371: These data indicate that a 126High LSC subset is selected during induction chemotherapy.

Once again, this isn't a novel concept. More quiescent cells will preferentially evade chemotherapy. This does not mean they drive treatment failure. A productive of unclear and dogmatic interpretation that should be avoided by the field and not supported.

Comment 15: 351- 395: Chemotherapy-refractory miR-126high LSCs are the likely cause of treatment failure in AML patients

This entire section of the results makes leaps in the logic which their data does support. Calling the miR-126 high LSCs refractory is not supported by their data.

First, refractory is a clinical state applied to patients, and cannot appropriately be used to describe a cell population. From this data type, the appropriate descriptor should instead be resistant/resilient. Using a clinically determined term without supporting data is non-scientific. From their presented graph (Figure 5L), it appears as though the Cluster 6' has a $\sim 1.4 \log_2$ FC (~ 2.64) in Treated vs. Control. Considering there is 26.4 median fold decrease in bulk leukemia in the xenografts (provided by the authors), for every 10 miR-126 high LSCs pre chemotherapy, there will be 1 miR-126 high LSC remaining after treatment. This is not chemotherapy resistance. If the authors wish to include this in their findings, they need to use the term "preferentially resistant" (also should mention it is not the most preferentially resistant population, Figure 5E)). The disease burden dynamics are not provided for the patient Day 0, Day 14 and Day 30 scRNA data and nor are the proportions of the primitive cell subclusters and therefore the term resistance/resilience/refractory cannot be used to describe the 0'' and 1'' miR-126 high LSCs here without reporting that data. The use of the clinical term "refractory" without any scientific reasoning is troubling and speaks to a pattern of embellishment of the data and its impact/importance.

It is not a novel finding that quiescent cells are more resilient to chemotherapy exposure, so it makes sense that miR-126 high LSCs survive chemotherapy better than miR-126 low LSCs. This, however, implies nothing about treatment failure, or the clinical states of relapse or refractory. Stretching the partial preferential resilience to chemotherapy treatment of subclusters 0'' and 1'' to treatment failure is flawed, especially when it is not the most preferentially resistant cell type in their analysis.

Comment 16: 383-385: PT08 underwent re-induction chemotherapy (FLAG-IDA) and showed disease persistence, with emergence of a complex karyotype. We performed scRNAseq on NPM1mut blasts collected on day 30 of re-induction chemotherapy.

This patient was the sample that was depleted of LSCs in xenografts. If LSCs drive therapy failure, how do the authors explain this patient having treatment failure while their LSCs are depleted by chemotherapy treatment in a PDX model ? This however is consistent with more recent published works by 3 independent groups, but is ignored by the authors. Another example of how the authors are preferentially select data to align with their claims.

Comment 17: 423: While the link between LSCs and relapse has been well established

There is debate within the field on this and shouldn't be stated as fact. The authors cite one study which have used genetic profiling to identify the source of relapse is an LSC in $\frac{1}{2}$ patients. This is

not a "well established link".

Comment 18: Our work exploits high resolution single cell analysis to provide evidence for a "classical" LSC model in NPM1mut AML, where non-response/relapse is driven by persisting, quiescent LSC already present at diagnosis.

No evidence was provided in which LSCs drive chemotherapy induced leukemic regeneration. In fact, this process was never modelled. The term "driven" is inappropriate to use here, as the only thing that was demonstrated is presence of these cells following treatment, with a cytoreduction comparable to the bulk of disease.

Comment 19: Quiescent leukemia stem cells reinforced by chemotherapy drive early treatment failure in acute myeloid leukemia patients

No evidence was provided supporting the claim that quiescent LSCs drive early treatment failure. Early treatment failure, as defined by refractory disease or early relapse of disease are defined clinical states and to claim causation one must functionally and biologically represent these states. I am concerned with the steps the authors are taking to draw this conclusion, considering refractory or relapse disease states were never modelled or biologically represented in this study. The authors demonstrate using retroactive bioinformatic data that a certain quiescent subset of LSCs are detectable following chemotherapy treatment, which is valuable work. The novelty of this is impaired because this could simply be a selection of quiescent cells, which has been reported prior. Nothing was done to demonstrate these are at the origin of relapse or are functionally related to disease regeneration. Causation studies are required to make the claims stated in the title. Furthermore, these cells in the xenograft modelling system (subcluster 6'), do not demonstrate resistance (are reduced to ~1/10th their original population size) and are not the most proportionally enriched cell type, therefore there are no grounds to suggest these cells are driving refractory disease, and suggesting LSCs are responsible relies on prior and unsubstantiated biases. The authors report functional LSC activity in PDXs after chemotherapy exposure is inconsistent, with some being depleted and some being enriched. Notably, the LSC depletion occurred in a PDX model derived from a patient which relapsed, which goes against the authors hypothesis, but is ignored in the derivation of their conclusions. The authors have shown that the quiescent fraction of AML cells correlated to overall survival and treatment response of patients but have not demonstrated the same for functional LSCs. A more accurate title based on the results would read: "Quiescent LSCs are detectable in post-chemotherapy AML patients and PDX models" but this is very specialized and focussed.

Reviewer #3 (Remarks to the Author):

The manuscript by Naldini describes characterization of early chemotherapy responses in both AML patients undergoing treatment and xenograft models. The authors prospectively identify leukemia stem cells (LSCs) through single cell RNA-seq and functional validation using a miR-126 reporter. The findings show important characteristics of varying cellular subpopulations during the first 2-4 weeks of chemotherapy treatment. In particular, the oxphos-low and miR-126 high LSC population was associated with chemotherapy resistance and survival.

This is an intriguing and unique study that significantly expands our current understanding of how inherently complex AML populations respond to chemotherapy. As noted by the authors, the dynamics of LSCs early in the course of chemotherapy treatment are poorly understood. The present study is much higher resolution and considerably more rigorous than previous reports. Several aspects of the work are particularly noteworthy:

- Serial LSC analysis of a large and relatively homogeneous cohort of patients (Fig 1a) is completely novel and highlights the importance of considering clinical pathology and treatment response in the evaluation of LSC biology.
- Application of temporal scRNA-seq analyses during the early course of therapy in AML patients is also novel and clearly delineates important differences in tumor biology that correlate with clinical response.

- In vivo modeling using the miR-126 reporter virus to map LSC properties in response to chemotherapy in the xenograft model is intriguing and demonstrates the utility of this approach to identify and characterize LSCs during the course of pathogenesis.

Aside from the points above, the paper is very thoughtful, and raises numerous important considerations for future studies. Further, the scope of the work is quite broad with ramifications that are important for both the design of clinical studies and more basic analysis of LSC biology. Lastly, the studies described are very technically demanding and represent a major experimental challenge. Collectively, it's an impressive body of work and will be of great interest to the field.

Minor point:

With regard to the use of IACS to target LSCs (lines 305-312), the rationale is somewhat confusing. The data from Lagadinou et al (Cell Stem Cell, 2013) indicates that it is not the level of oxphos activity that determines efficacy of oxphos inhibition, but rather the unique biology of the LSC population. That is to say, de novo LSCs have intrinsically low oxphos activity, but are highly reliant on oxphos, presumably due to inability to compensate with glycolytic mechanisms. Hence the rationale stated on line 305-306 doesn't really make sense. The authors may appropriately point out that IACS treatment did indeed fail to kill LSCs as they hypothesized, but this is more likely due to mechanism of IACS, which targets ETC complex 1. Several lines of evidence suggest LSCs can bypass complex I to drive oxphos. Hence, failure to kill LSCs in this experiment may be due to the use of IACS rather than the role of oxphos. It would be interesting to repeat the experiment and use venetoclax instead. The reviewer is not suggesting this experiment is required for the present study but would simply request the text to be modified to reflect the points above.

RESPONSE TO REVIEWERS' COMMENTS

Reviewer #2 (Remarks to the Author):

General over/mis-interpretation and lack of data to support claims continues to inflict this study. Furthermore, the authors now have added data on a new genetic subgroup of patients (chromosome 7 deletion... it is unclear why as this was not raised or requested). However, this new data conflicts with the authors original narrative, and instead suggests that for these patients the primitive cells get recruited into cycle transiently similar to Farge, Boyd etc.,. From a conceptual point of view, the variable observations seen across patients makes it difficult for the authors to clearly resolve the controversies in the field that they set out to address.

Although investigated with state-of-the-art technology, the study does not present a new understanding of the underlying biology, and the observations are difficult to clearly interpret, in part due to reliance on molecular descriptions. The authors' main defence of their claims rests heavily on the observations that miR-126high LSC content at diagnosis predicts therapy outcomes in patients, however the observation that LSC signatures predict survival is not surprising or novel.

We respectfully disagree with the view of this reviewer, which continues to be inclined by a single thought on AML response to chemotherapy that may have been advocated by some recent literature but is not fully consistent with the interpretation of the entirety of our data. In all our previous versions of this manuscript, we have adequately and respectfully cited the work by Farge, Boyd etc providing a balanced view on the current thinking of chemotherapy response as a premise for this work.

The data on del(7) AML had already been included in the R1 version of our paper (and not for the appeal). The reason why these data have been added was to provide a broader picture on AML, which is a heterogeneous disease. There is no conflict with our initial narrative, rather an addition of a layer of complexity, which is a well-recognized feature of AML and an added value to our manuscript by broadening the investigation which was initially criticized of being "restricted to NPM1+ subset of AML".

As instructed by the Editor, we have toned down the biological claims of our study to avoid unnecessary controversy, even though we believe in our original conclusions.

This reviewer has clearly stated his view on the novelty of our work already in the R1 review and does not add novel aspects here. Of note, this assessment is not shared by Reviewer 3, who eloquently points out the novel aspects and value of our work.

I have provided very detailed inconsistencies, and concrete comments below. I hope this is helpful to the editors and authors alike.

We thank the reviewer for all the time he has taken to evaluate our work.

Inconsistencies:

- The authors now introduce a second molecular subtype of AML (del7), with a different pattern of chemotherapy response than their initially characterized NPM1+ AML patients. Specifically, they describe a transient proliferative response across all primitive subsets at Day 14 post induction chemotherapy. This challenges the overarching conclusion that cell cycle dormancy represents the central mechanism of therapy failure in AML. It instead suggests that the original conclusions of the study are restricted to NPM1+ subsets of AML, where results were even inconsistent across different patients examined (e.g., Fig 3H, 5F, 5I).

We agree with the reviewer that the addition of different disease entities to the manuscript may add to the complexity of the system, which we acknowledge in the thorough description we outline of blast subpopulations persisting post-chemotherapy in AML patients. We have removed unsubstantiated general conclusions from the manuscript and have maintained a well outlined disease subclass description of the findings.

- The authors also provide new evidence that at relapse, miR126High subsets consistently showed significant cell cycle activity, unlike miR126High cells at diagnosis (Fig 6H). Relapsed AML is notoriously more chemoresistant than initially diagnosed AML (e.g., as seen for Patients PT08, PT10, PT15, and PT19 in Figure 1A), yet Fig 6H suggests that relapsed LSCs are less quiescent than at diagnosis. This observation opposes the central conclusion that LSC quiescence is the main determinant of chemoresistance, but is consistent with the more recent published literature. For some reason, the authors refuse to acknowledge this fact.

The central conclusion of the manuscript is the observation of the persistence of miR-126high LSC throughout induction chemotherapy in patients and PDX. We do not extend such conclusion to the relapse setting, which we separately analyze in detail and clearly highlight both parallelisms and differences with treatment-naïve AML, or AML early after chemotherapy. Furthermore, we also extend our observations to case-reports of relapse AML undergoing re-induction chemotherapy describing, also for this setting, the presence of quiescent miR-126high LSCs in residual AML. Acknowledging that a single case is not sufficient to make broad claims, this case report still shows how return to quiescence may indeed be a major mechanism of chemotherapy resistance in the

relapse setting. We have removed claims & hypotheses on LSC plasticity from the revised version of the manuscript.

- The authors have also incorporated new data showing that miR126High cells can show increased EdU incorporation following chemotherapy treatment (Fig 5I), yet they do not acknowledge this as evidence that LSCs may be recruited into cell cycle by chemotherapy. Why? Early in the manuscript, they firmly assert that miR126 faithfully identifies functional LSCs. However, they avoid using the term LSCs in their interpretation of this cell cycle result, and instead simply address this observation as early proliferation of miR126High “progenitors”. ??? There is no functional or molecular data to support this distinction of LSCs vs transient leukemic progenitors within the cycling miR126High populations, other than the authors’ personal views.

It goes without saying that stem cells may be recruited into cell cycle, some more easily, others requiring stronger signals. We have better highlighted the fact that there is heterogeneity within the miR-126high and the LSC compartment alike, not surprisingly when considering the huge body of literature that is emerging on non-malignant hematopoietic stem cells, of which this reviewer is certainly aware. Regarding functional data distinguishing between the consequences of symmetric/asymmetric division vs dormancy of individual LSCs, we have already clarified in our appeal that, in our view, this would be beyond the scope of this publication.

- There is a general disconnect between molecular descriptions and functional assessments. Some of the strongest claims are derived from scRNAseq data, such as the persistence of LSCs with “enhanced stemness features upon in-vivo chemotherapy”. In contrast, functional assessment of LSCs does not suggest evidence of enhanced stemness, and in fact sometimes suggests reduced stemness (Fig 3G). This is a problem, since functional assays represent the gold standard for studying LSCs (as acknowledged by the authors in their response).

We have trouble understanding the basis on which this reviewer claims a general “disconnect between molecular descriptions and functional assessments”. Our data show good consistency between molecular and “gold-standard” assays and, if this was not the case, we have properly discussed the results.

- Overall, the Title, Abstract, and Discussion oversimplify the complexity of the biology and fail to address the fact that the authors have exclusively examined NPM1+ AML in most of their figures. In the absence of functional intervention experiments in vivo or other sophisticated cell tracking experiments as the authors described, it is not possible to infer what cells are responsible for driving therapy failure.

This comment is inconsistent with the fact that we included the del(7) cases as well as the relapsed AMLs in our paper. We have made significant efforts to adequately represent the complexity of this disease. Not without regret, we have now removed biological interpretations of our data that may generate controversy, as instructed by the Editor.

Specific Scientific Comments:

Comment 1:

Lines 62-64: Knowledge on the cell biology and population dynamics of relapse-promoting cells during chemotherapy is scarce, with the general assumption that their proportion tends to increase with successive cycles of chemotherapy, to which LSCs are resistant.

No citation is provided. As the authors state earlier this is a controversial claim (lines 28-29) and having this statement without a source is setting up a “begging the question” fallacy as it orients their final conclusion. A study in which LSCs have been shown to be consistently preferentially resistant to chemotherapy functionally is required, aside for the dogma and belief. This does not mean to provide a paper that shows they are clinically predictive (Ng 2016) or sometimes retrospectively shown origin of relapse (Shlush 2017) as these are different statements entirely. There are a handful of papers from independent groups that show that LSCs are functionally and phenotypically depleted (Farge, Duy, Boyd) and it is unacceptable to make this statement and speak about the bias for generation of conclusions.

The text has been changed as follows:

Knowledge on the cell biology and population dynamics of relapse-promoting cells during chemotherapy is incomplete. The classical LSC model, whereby chemotherapy-resistant LSCs accumulate during successive cycles of chemotherapy, has recently been challenged.

Comment 2:

Lines 80-82: As part of this score, miR-126 was tightly coupled with LSC and HSC function preserving their quiescent state and promoting chemotherapy resistance by orchestrating a bona fide stem cell program (Lechman 2016).

I once AGAIN suggest the authors critically analyze the work that provides the foundation of their analysis. In Lechman et al 2016, it is apparent that although a conclusion of the paper, there is no data conclusively

supporting that miR-126 induces LSC specific chemotherapy resistance in primary AML. The paper demonstrates that miR-126 cells are enriched post chemotherapy in vitro, but miR-126 is part of general cellular senescence marker biology as the authors state (lines 188-190), and many cells within the hematopoietic/leukemic systems are non-cycling (dc/monocytes/macrophages) and express miR-126, as the authors acknowledge (lines 272-273). With evidence suggesting that monocytic leukemia's demonstrate more chemotherapy resistance (Pei 2020, Canaani 2017), I respectfully urge the authors to consider that enrichment of miR-126 is selecting for non-cycling cells, and not LSCs specifically.

I acknowledge that the miR-126 population is enriched for LSC functional activity, but a maximum 20fold enrichment is not enough to rule out the enrichment of other cell cycle dormant cells within the miR-126 high population, and what the Lechman 2016 Figure 7 conclusion should be is that quiescent cells are resistant to chemotherapy. This is not a novel finding, as we understand the mechanisms of action of cytarabine affects cycling cells. This unfounded statement combined with not citing more conclusive evidence providing the opposite idea, where LSCs are recruited into the cell cycle following a chemotherapy exposure (Boyd et al 2018) in more biologically relevant models (in vivo and from patients), the authors once again attempt to adhere to an unproven ideas, which they hold and use for interpretation, while developing their unsupported claims.

We are accurately citing a published and peer-reviewed paper in the introduction of our work, and, as already stated above, the Boyd paper is also accurately cited in our work, providing a balanced view on the field.

The suggestion of the reviewer to consider that enrichment of miR-126 is selecting for non-cycling cells is a reductionist view not supported by the literature. We now cite in the text another one of our papers (Nucera et al, Cancer Cell 2016) describing how miR-126 orchestrates an oncogenic program in leukemia, by impacting other pathways than the cell cycle.

Moreover, ectopic expression of miR-126 in murine HSC may induce a miR-126 addicted acute leukemia, underlining its oncogenic potential²¹.

Lastly, we believe our experimental investigation of samples coming from patients and PDX undergoing polychemotherapy induction regimens may be viewed as of increased clinical relevance compared to previous reports. Indeed, Boyd et al have used, in their mouse models, a chemotherapy regimen (AraC monotherapy) that should be regarded as clinically-inappropriate for newly-diagnosed AML, unless one wants to model a palliative setting. Their work also provides a less granular picture on chemotherapy response in patients than our work does.

Comment 3:

Lines 144-147 Post-chemotherapy leukemic cells prominently mapped to the progenitor clusters, with underrepresentation of differentiated myelo-/monoblasts, but enrichment in the erythroid-like clusters and accumulation of cells bridging the early and myeloid immature-like progenitor area on day 30 (Fig. 2A,B). One of the final conclusions of the paper is that LSCs are enriched post chemo, which is contrary to this finding and not acknowledged by the authors.

Our conclusion is a persistence of LSC with enriched stemness features during induction chemotherapy in patients and PDX models, and an enrichment of LSCs at relapse. We do not see how this contrasts with the finding cited by the reviewer. At this stage of the article (Fig.2A,B), we describe an enrichment of progenitors (including LSC; we used the umbrella term "progenitor" as we did not delve into the details of this population yet). As we elaborate later in our article, within these enriched progenitors, there is an enrichment of LSC with enhanced stemness features.

We modified the text as follows, not mentioning any more specifically the reduction of mature myeloid cells as it is already implied when we state an enrichment of progenitors.

Post-chemotherapy leukemic cells prominently mapped to the progenitor clusters, with enrichment in the erythroid-like clusters and accumulation of cells bridging the early and myeloid immature-like progenitor area on day 30 (Fig. 2A,B).

Comment 4: 176-178: Due to the limited understanding of the response of LSCs to induction chemotherapy, and how this contributes to early treatment failure.

Once again, there is no evidence besides theoretical that LSCs contribute to early treatment failure

To avoid controversy, we have removed this introductory sentence.

Comment 5: 190-192: Cell cycle analysis on 3 representative diseases confirmed depletion of miR-126high cells in the S/G2/M phase and enrichment in G0 (PT01) or G1 (PT03, PT16) (Supplementary Fig. 3B).

This data supports that quiescent cells are preferentially resistant to cytarabine, which is not a novel finding.

We have added cell cycle profiling, as requested in the previous revision of the manuscript. This data is described objectively in the manuscript, without claims of novelty. Inferences on relationships between

subpopulations prior to and after chemotherapy, as mentioned also by this reviewer, would require complex longitudinal cell tracking assays, which are beyond the scope of this manuscript.

Comment 6: 198 – 201: We analyzed the mouse BM early after chemotherapy (day 8) and showed marked reduction of hCD45+ blasts in the treated mice (Fig. 3E), with a relative increase of immunophenotypically primitive CD34+38- cells indicating that they exhibited chemotherapy resistance (Fig. 3F). The authors must report the total number immunophenotypically primitive AML cells, or alter the statement to say partial chemotherapy resilience. With what appears to be a ~10-100 fold decrease of hCD45+ cells in the grafts, the total number of immunophenotypically primitive cells post likely decreases following treatment. The statements of conclusion do not coincide with the data being reported.

For the sake of simplicity, we have reported only relative proportions of the CD34+38- fraction in the BM of PDX. Absolute counts reveal no changes in CD34+38- cell numbers in 3 out of 5 diseases, while we noted a slight decrease in 2 out of 5 diseases. We have, nonetheless, for convenience changed the terminology in the manuscript to “chemotherapy-resilience”, as we do not regard this terminology as a critical point.

The text now reads:

...with a relative increase of immunophenotypically primitive CD34+38- cells indicating that they exhibited *relative chemotherapy-resilience* (Fig. 3F).

Comment 7: Line 201: “While immunophenotypically-defined LSCs increased”

This statement is incorrect. A statistical increase in cell proportion but on current presented data there is no evidence of an increase in immunophenotypically defined LSCs. See above statement. The embellishment of this data to support their hypothesis is unacceptable.

We have changed the text as follows:

While *the proportion of immunophenotypically-defined LSCs increased*

Comment 8: Line 201-202: Overall miR-126 activity remained stable or decreased after chemotherapy. This statement is in opposition to the conclusion of this paper, and that fact is not acknowledged by the authors. The terminology “remains stable” is misleading, as this actually means the cell population decreased in the same proportion as the rest of the bulk leukemia (~10-100x). Another attempt to support the authors narrative.

We do not find any opposition in the cited statement with regards to the conclusions of the paper, as it refers to the measurement of average miR-126 activity within the bulk blasts and, as we have extensively detailed in the manuscript, bulk observations do not entail sufficient resolution to properly catch changes in rarer LSC-enriched subpopulations which require single-cell readouts (i.e. TGR distribution analysis and scRNAseq analysis).

We have improved the terminology of the text, which now reads:

While *the proportion of immunophenotypically-defined LSCs increased, average miR-126 activity of the total blast population was unchanged or decreased (reflected by an increase in miR-126 reporter vector transgene ratio) after chemotherapy (Supplementary Fig. 3F).*

Comment 9: Figure 5H:

The inconsistent nature of LSC response to chemotherapy demonstrated here (1 responds, 2 don't change, 2 are enriched [one of which has $p = 0.055$ which is not significant]) speaks to the inconsequential nature of LSC persistence following chemotherapy treatment as it relates to relapse. There is not enough evidence here to conclude that LSCs preferentially survive treatment, working against the hypothesis of the paper, and is not acknowledged or considered by the authors while developing the conclusions. When analyzing in an unbiased way, the most reasonable explanation of this data is that functional LSCs are not preferentially spared. This highlights a troubling pattern of selecting pieces of data that supports their ideas and claims vs looking at the data generated in its collective form.

We are firmly opposed to this representation of our data. As already debated in the previous revisions, we sustain that the limiting dilution data presented in Figure 3H is robust, significant and analyzed & presented in its entirety, without bias and in line with scientific standards in the field. In the scope of a more descriptive manuscript, as requested by the editor, we have removed hypothetical text from the conclusion of this paragraph.

~~Taken together, chemotherapy resulted in the enhancement of a miR-126high LSC population in 2 out of 5 AML samples, which correlated with quiescence and higher LIC frequency. Such functional correlation was not observed with the CD34+CD38- immunophenotype, suggesting that miR-126 activity may be a valid LSC marker even after treatment.~~

Comment 10: 265-268: While steady-state control AMLs displayed enrichment for progenitor and cycling populations, chemotherapy radically reshaped AML, with expansion of more differentiated myeloid and, especially, erythroid-like blasts (Fig. 5B,E). On the other hand, the proportion of HSC-like blasts remained stable. This result is in contrast to the authors hypothesis and claims, and is not considered in the conclusions. HSC like blasts are most enriched for miR-126 and are not enriched following treatment and depleted to a similar level as the bulk disease, indicating chemo sensitivity. The term “remains stable” used by the authors needs to be

adjusted as it implies an absence of chemotherapeutic effect. The proportion of cells is similar, which means it reduced just as much as the bulk leukemia (~10-100 fold reduction).

The presumed contradiction has been already addressed in previous replies to comments by the reviewer, where both qualitative and quantitative considerations of the miR-126high LSC-enriched subpopulations have been highlighted.

“remained stable” has been replaced by “unchanged” in the text.

Comment 11: 305-312: If these OxPhoslow LSC were at the origin of relapse, we hypothesized that these diseases would be resistant to pharmacological OxPhos inhibition. Hence, we treated miR-126 reporter+ AML cells from PT16 in vitro with IACS-010759 (IACS)32, a selective mitochondrial electron transport chain complex I-inhibitor, alone or in combination with AraC chemotherapy. IACS-treated cells had higher miR-126 activity (Supplementary Fig. 5E-G), consistent with selection of OxPhoslow LSCs. Strikingly, as compared to the mock-treated group, IACS-treated cells showed strongly enhanced repopulation activity when transplanted into NSG mice, underlining the functional relevance of these miR-126high OxPhoslow LSCs, at least in NPM1mut AML (Fig.5N).

It is unclear how this experiment supports OxPhos low LSCs are the origin of relapse as stated by the authors? This experimental design shows that functional LSCs are selected for by treating with IACS. It speaks nothing to the origin of relapse. Relapse is a clinical state of regenerated disease and not biologically represented or modelled in this experiment. Leukemia initiating activity as represented by LSC engraftment in immune deficient mice is a different biological process than chemotherapy induced disease regeneration resulting in clinical relapse and there is no scientific rationale to compare the two. Drawing this conclusion while the data supports no such claim is misrepresentation of the results.

We have made appropriate modifications to the text as suggested by the reviewer’s comment. Please find the details of the modifications in the response to Reviewer 3’s comment below.

Comment 12: 314: miR-126high LSCs at diagnosis correlate with AML outcome
LSC markers being predictive of AML outcome is not a new.

Emphasis on this statement has been toned down by removing “strikingly” from the text

Comment 13: 315-316: Our xenograft data suggest that a reinforced LSC state, identifiable by high miR-126 activity and persisting through induction chemotherapy, may be responsible for treatment failure
miR-126 activity has not been demonstrated to preferentially persist through chemo treatment in the xenografts. Other cell types appear to exhibit more preferential chemotherapy resistance. The same statement about being “responsible for treatment failure” can be made about any other cell type that persists through chemotherapy, and making this statement exclusive to LSCs is misrepresentative

Following the editorial suggestions, we have employed a strictly descriptive narrative with the removal of this introductory sentence.

Comment 14: 370-371: These data indicate that a 126High LSC subset is selected during induction chemotherapy.

Once again, this isn’t a novel concept. More quiescent cells will preferentially evade chemotherapy. This does not mean they drive treatment failure. A productive of unclear and dogmatic interpretation that should be avoided by the field and not supported.

The statement has been removed from the revised text.

Comment 15: 351- 395: Chemotherapy-refractory miR-126high LSCs are the likely cause of treatment failure in AML patients

The section subtitle has been changed to a purely descriptive statement:

AML cells mapping to miR-126^{high} LSC clusters enrich during chemotherapy and at relapse

This entire section of the results makes leaps in the logic which their data does support. Calling the miR-126 high LSCs refractory is not supported by their data.

First, refractory is a clinical state applied to patients, and cannot appropriately be used to describe a cell population. From this data type, the appropriate descriptor should instead be resistant/resilient. Using a clinically determined term without supporting data is non-scientific.

The word “refractory” has been removed from the section. Nevertheless, it should be noted that the cells, which have been harvested for scRNAseq came from clinically-defined refractory patients, and these show an increase in miR-126high LSCs.

From their presented graph (Figure 5L), it appears as though the Cluster 6' has a $\sim 1.4 \log_2$ FC (~ 2.64) in Treated vs. Control. Considering there is 26.4 median fold decrease in bulk leukemia in the xenografts (provided by the authors), for every 10 miR-126 high LSCs pre chemotherapy, there will be 1 miR-126 high LSC remaining after treatment. This is not chemotherapy resistance. If the authors wish to include this in their findings, they need to use the term "preferentially resistant" (also should mention it is not the most preferentially resistant population, Figure 5E)). The disease burden dynamics are not provided for the patient Day 0, Day 14 and Day 30 scRNA data and nor are the proportions of the primitive cell subclusters and therefore the term resistance/resilience/refractory cannot be used to describe the 0'' and 1'' miR-126 high LSCs here without reporting that data. The use of the clinical term "refractory" without any scientific reasoning is troubling and speaks to a pattern of embellishment of the data and its impact/importance.

We did not find any mention of resistance/resilience/refractory with respect to Clusters 0'' and 1'' in the text. Elsewhere, when appropriate, we have modified the text as suggested by the reviewer.

It is not a novel finding that quiescent cells are more resilient to chemotherapy exposure, so it makes sense that miR-126 high LSCs survive chemotherapy better than miR-126 low LSCs. This, however, implies nothing about treatment failure, or the clinical states of relapse or refractory. Stretching the partial preferential resilience to chemotherapy treatment of subclusters 0'' and 1'' to treatment failure is flawed, especially when it is not the most preferentially resistant cell type in their analysis.

We have edited the text to a descriptive only format, however, the strong correlations of this LSC states with clinical outcome that we and others find speak of a functional relevance of these states for treatment failure. It should be noted that this statement of the reviewer is inconsistent with his argumentation above, where he claims that LSCs are just as chemotherapy-sensitive as the AML bulk.

Comment 16: 383-385: PT08 underwent re-induction chemotherapy (FLAG-IDA) and showed disease persistence, with emergence of a complex karyotype. We performed scRNAseq on NPM1mut blasts collected on day 30 of re-induction chemotherapy.

This patient was the sample that was depleted of LSCs in xenografts. If LSCs drive therapy failure, how do the authors explain this patient having treatment failure while their LSCs are depleted by chemotherapy treatment in a PDX model? This however is consistent with more recent published works by 3 independent groups, but is ignored by the authors. Another example of how the authors are preferentially select data to align with their claims.

PT08 was not tested in the PDX model.

Comment 17: 423: While the link between LSCs and relapse has been well established there is debate within the field on this and shouldn't be stated as fact. The authors cite one study which have used genetic profiling to identify the source of relapse is an LSC in $\frac{1}{2}$ patients. This is not a "well established link".

This statement has been softened:

While some recent data have supported a link between LSCs and relapse⁷, few studies have addressed the molecular mechanisms and dynamics of early therapy resistance^{13-15,47}

Comment 18: Our work exploits high resolution single cell analysis to provide evidence for a "classical" LSC model in NPM1mut AML, where non-response/relapse is driven by persisting, quiescent LSC already present at diagnosis.

No evidence was provided in which LSCs drive chemotherapy induced leukemic regeneration. In fact, this process was never modelled. The term "driven" is inappropriate to use here, as the only thing that was demonstrated is presence of these cells following treatment, with a cytoreduction comparable to the bulk of disease.

We have toned down the conclusions of the paragraph and added considerations on the limits of our study as follows:

Our work exploits high resolution single cell analysis to provide evidence for a "classical" LSC model in NPM1mut AML, where non-response/relapse was strongly correlated to a high proportion of quiescent miR-126high LSC already present at diagnosis. A weakness of our study is the relatively limited number of patients and the lack of longitudinal monitoring of single LSCs, as a direct demonstration that relapse unequivocally derives from a quiescent LSC persisting through induction chemotherapy. Nevertheless, our...

Comment 19: Quiescent leukemia stem cells reinforced by chemotherapy drive early treatment failure in acute myeloid leukemia patients

No evidence was provided supporting the claim that quiescent LSCs drive early treatment failure. Early treatment failure, as defined by refractory disease or early relapse of disease are defined clinical states and to claim causation one must functionally and biologically represent these states. I am concerned with the steps the authors are taking to draw this conclusion, considering refractory or relapse disease states were never modelled or biologically represented in this study. The authors demonstrate using retroactive bioinformatic data that a

certain quiescent subset of LSCs are detectable following chemotherapy treatment, which is valuable work. The novelty of this is impaired because this could simply be a selection of quiescent cells, which has been reported prior. Nothing was done to demonstrate these are at the origin of relapse or are functionally related to disease regeneration. Causation studies are required to make the claims stated in the title. Furthermore, these cells in the xenograft modelling system (subcluster 6'), do not demonstrate resistance (are reduced to ~1/10th their original population size) and are not the most proportionally enriched cell type, therefore there are no grounds to suggest these cells are driving refractory disease, and suggesting LSCs are responsible relies on prior and unsubstantiated biases. The authors report functional LSC activity in PDXs after chemotherapy exposure is inconsistent, with some being depleted and some being enriched. Notably, the LSC depletion occurred in a PDX model derived from a patient which relapsed, which goes against the authors hypothesis, but is ignored in the derivation of their conclusions. The authors have shown that the quiescent fraction of AML cells correlated to overall survival and treatment response of patients but have not demonstrated the same for functional LSCs. A more accurate title based on the results would read: "Quiescent LSCs are detectable in post-chemotherapy AML patients and PDX models" but this is very specialized and focussed.

We have changed the title and text of the manuscript to a "descriptive-only" style.

Data coming from PT15 PDX is very consistent across our analysis. We do not observe an enhanced miR-126high LSC subpopulation after chemotherapy which, indeed, is paralleled by a reduced LIC frequency after chemotherapy (although there still are engrafting mice in the treated AML transplant group). Furthermore, this is the only PDX for which in scRNAseq we do not observe enrichment of the LSC Cluster 6" after chemotherapy. Inconsistencies between PDX modeling and patient outcome can be expected and should not surprise when considering the heterogeneity of AML and the complexity of its treatments. Moreover, ~5mL of a bone marrow aspirate may not sample the entirety of AML found in the patient. Notably, PT15 relapsed also with extramedullary disease after quite a long relapse-free interval post autologous transplantation, for which many clinical, biological and treatment-related variables may be at play.

Reviewer #3 (Remarks to the Author):

The manuscript by Naldini describes characterization of early chemotherapy responses in both AML patients undergoing treatment and xenograft models. The authors prospectively identify leukemia stem cells (LSCs) through single cell RNA-seq and functional validation using a miR-126 reporter. The findings show important characteristics of varying cellular subpopulations during the first 2-4 weeks of chemotherapy treatment. In particular, the oxphos-low and miR-126 high LSC population was associated with chemotherapy resistance and survival.

This is an intriguing and unique study that significantly expands our current understanding of how inherently complex AML populations respond to chemotherapy. As noted by the authors, the dynamics of LSCs early in the course of chemotherapy treatment are poorly understood. The present study is much higher resolution and considerably more rigorous than previous reports. Several aspects of the work are particularly noteworthy:

- Serial LSC analysis of a large and relatively homogeneous cohort of patients (Fig 1a) is completely novel and highlights the importance of considering clinical pathology and treatment response in the evaluation of LSC biology.
- Application of temporal scRNA-seq analyses during the early course of therapy in AML patients is also novel and clearly delineates important differences in tumor biology that correlate with clinical response.
- In vivo modeling using the miR-126 reporter virus to map LSC properties in response to chemotherapy in the xenograft model is intriguing and demonstrates the utility of this approach to identify and characterize LSCs during the course of pathogenesis.

Aside from the points above, the paper is very thoughtful, and raises numerous important considerations for future studies. Further, the scope of the work is quite broad with ramifications that are important for both the design of clinical studies and more basic analysis of LSC biology. Lastly, the studies described are very technically demanding and represent a major experimental challenge. Collectively, it's an impressive body of work and will be of great interest to the field.

Minor point:

With regard to the use of IACS to target LSCs (lines 305-312), the rationale is somewhat confusing. The data from Lagadinou et al (Cell Stem Cell, 2013) indicates that it is not the level of oxphos activity that determines efficacy of oxphos inhibition, but rather the unique biology of the LSC population. That is to say, de novo LSCs have intrinsically low oxphos activity, but are highly reliant on oxphos, presumably due to inability to compensate with glycolytic mechanisms. Hence the rationale stated on line 305-306 doesn't really make sense. The authors may appropriately point out that IACS treatment did indeed fail to kill LSCs as they hypothesized, but this is more likely due to mechanism of IACS, which targets ETC complex 1. Several lines of evidence suggest LSCs can

bypass complex I to drive oxphos. Hence, failure to kill LSCs in this experiment may be due to the use of IACS rather than the role of oxphos. It would be interesting to repeat the experiment and use venetoclax instead. The reviewer is not suggesting this experiment is required for the present study but would simply request the text to be modified to reflect the points above.

We sincerely thank this reviewer for his/her positive feedback and acknowledgment of the efforts involved in the analysis presented within this manuscript. We have revised the results and discussion sections as suggested to better highlight the rationale and setting for which oxidative phosphorylation inhibition with IACS was performed. The text now reads:

Lines 341-347: ~~If these OxPhoslow LSC were at the origin of relapse, we hypothesized that these diseases would be resistant to pharmacological OxPhos inhibition. Hence, We treated miR-126 reporter+ AML cells from PT16 in vitro with IACS-010759 (IACS)35, a selective mitochondrial electron transport chain complex I-inhibitor, alone or in combination with AraC chemotherapy. IACS-treated cells had higher miR-126 activity (Supplementary Fig. 5E-G), consistent with preferential depletion of OxPhoshigh miR-126low progenitors. IACS-treated cells showed strongly enhanced repopulation activity when transplanted into NSG mice as compared to the mock-treated group, underlining the functional relevance of these miR-126high OxPhoslow LSCs, at least in NPM1mut AML (Fig. 5N).~~

Line 537: therapeutic targeting of oxidative phosphorylation, at least at the level of ETC complex I inhibition,